# LucidPPN: Unambiguous Prototypical Parts Network for User-centric Interpretable Computer Vision

**Mateusz Pach**[1]    **Koryna Lewandowska**[2]    **Jacek Tabor**[1]
**Bartosz Zieliński**[1,3]    **Dawid Rymarczyk**[1,4,*]
[1]Faculty of Mathematics and Computer Science, Jagiellonian University
[2]Department of Cognitive Neuroscience and Neuroergonomics, Jagiellonian University
[3]IDEAS NCBR    [4]Ardigen SA    [*]dawid.rymarczyk@uj.edu.pl

## Abstract

Prototypical parts networks combine the power of deep learning with the explainability of case-based reasoning to make accurate, interpretable decisions. They follow the this looks like that reasoning, representing each prototypical part with patches from training images. However, a single image patch comprises multiple visual features, such as color, shape, and texture, making it difficult for users to identify which feature is important to the model. To reduce this ambiguity, we introduce the Lucid Prototypical Parts Network (LucidPPN), a novel prototypical parts network that separates color prototypes from other visual features. Our method employs two reasoning branches: one for non-color visual features, processing grayscale images, and another focusing solely on color information. This separation allows us to clarify whether the model's decisions are based on color, shape, or texture. Additionally, LucidPPN identifies prototypical parts corresponding to semantic parts of classified objects, making comparisons between data classes more intuitive, e.g., when two bird species might differ primarily in belly color. Our experiments demonstrate that the two branches are complementary and together achieve results comparable to baseline methods. More importantly, LucidPPN generates less ambiguous prototypical parts, enhancing user understanding.

## 1 Introduction

Increased adoption of deep neural networks across critical fields, such as healthcare (Rymarczyk et al., 2022b), and autonomous driving (Wu et al., 2017), shows the need to develop models in which decisions are interpretable, ensuring accountability and transparency in decision-making processes (Rudin, 2019; Rudin et al., 2022). One promising approach is based on prototypical parts (Chen et al., 2019; Nauta et al., 2023; Rymarczyk et al., 2021; 2022d), which integrate the power of deep learning with interpretability, particularly in fine-grained image classification tasks. During training, these models learn visual concepts characteristic for each class, called Prototypical Parts (PPs). In inference, predictions are made by identifying the PPs of distinct classes within an image. This way, the user is provided with explanations in the form of "This looks like that".

The primary benefit of PPs-based methods over post hoc approaches is their ability to incorporate explanations into the prediction process (Chen et al., 2019) directly. Nevertheless, a significant challenge with these methods lies in the ambiguity of prototypical parts, visualized using five to ten nearest patches. Each patch encodes a range of visual features, including color[1], shape, texture, and contrast (Nauta et al., 2021a), making it difficult for users to identify which of them are relevant. This issue is compounded by the fact that neural networks are generally biased towards texture (Geirhos et al., 2019) and color (Hosseini et al., 2018), whereas humans are typically biased towards shape (Landau et al., 1988).

---

[1]We follow the color definition from the research of (Berga et al., 2020; Khan et al., 2012)

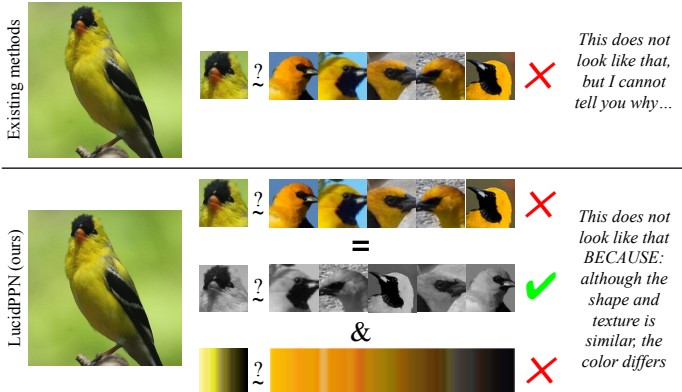

Figure 1: Our novel prototypical parts-based model, LucidPPN, enables the disentangling of color information from the prototypical parts. This capability allows us to examine more closely the differences between an image patch and patches representing a prototypical part. As shown in the image, our model can visualize that the head of a bird, compared to the prototypical part of a bird's head from different classes, shows a high resemblance in shape and texture but differs in color. Such detailed analysis was not possible with previous prototypical parts-based approaches.

Therefore, recent works have attempted to solve this problem using various strategies. Some works propose to reduce the ambiguity of prototypical parts by visualizing them through a larger number of patches (Ma et al., 2024; Nauta et al., 2023). However, it does not solve the problem with various visual features encoded in each patch. Other approaches tend to solve this problem by quantifying the appearance of specific visual features (Nauta et al., 2021a) or concepts (Wan et al., 2024) on prototypical parts. However, they generate ambiguous statements such as "color is important", leading to further questions (e.g. about which color) that complicate understanding (Ma et al., 2024; Xu-Darme et al., 2023).

Motivated by the challenge of decoding the crucial visual attributes of prototypical parts, we introduce the Lucid Prototypical Parts Network (LucidPPN). It uniquely divides the model into two branches: the first focuses on identifying visual features of texture and shape corresponding to specific object parts (e.g. heads, tails, wings for birds), while the second is dedicated solely to color. It allows us to disentangle color features from the prototypical parts and present pairs of a simplified gray prototypical part and corresponding color (see Figure 1). The second advantage of LucidPPN is that the successive prototypes in each class correspond to the same object parts (e.g., the first prototypes are heads, the second prototypes are legs, etc.). Altogether, it enabled us to introduce a novel type of visualization presented in Figure 2, more intuitive and less ambiguous according to our user studies.

Extensive experiments demonstrate that LucidPPN achieves results competitive with current PPs-based models while successfully disentangling and fusing color information. Additionally, using LucidPPN , we can identify tasks where color information is an unimportant feature, as demonstrated on the Stanford Cars dataset (Krause et al., 2013). Finally, a user study showed that participants, guided by LucidPPN explanations, more accurately identified the ground truth compared to those using PIP-Net.

Our contributions can be summarized as follows:

- We introduce LucidPPN, a novel architecture based on PPs, which disentangles color features from the PPs in inference. Consequently, thanks to LucidPPN we know the relevance of the color and shape with texture in the decision process[2].
- We propose a mechanism that ensures successive prototypes within each class consistently correspond to the same object parts.
- We introduce a more intuitive type of visualization incorporating the assumption about the fine-grained classification.
- We conduct a comprehensive examination demonstrating the usability and limitations of LucidPPN. Specifically, we highlight scenarios where color information may not be pivotal or even confuses the model in fine-grained image classification.

---

[2]See the discussion in paragraph Color Impact in Section 5.

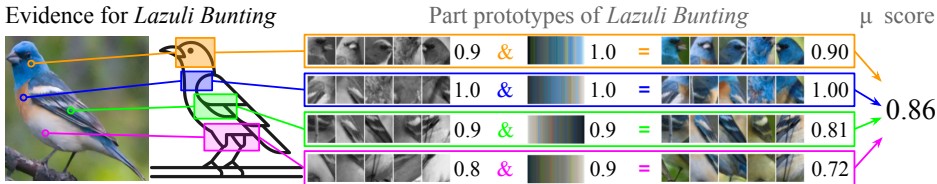

Figure 2: Our novel type of visualization utilizes the fact that the successive prototypes in each class of LucidPPN correspond to the same object parts. That is why we use a schematic drawing of a bird to show the location of the specific prototypical parts. Moreover, LucidPPN disentangles color features from the prototypical parts to present pairs of a simplified gray prototypical part and a corresponding color. The aggregated resemblance is obtained by multiplying the resemblance to the prototypical part and the resemblance to the corresponding color.

## 2 RELATED WORKS

**Ante-hoc methods for XAI.** Self-explainable models (ante-hoc) aim to make the decision process more transparent by providing the explanation together with the prediction, and they have attracted significant attention (Alvarez Melis & Jaakkola, 2018; Böhle et al., 2022; Brendel & Bethge, 2019). Much of this attention has focused on enhancing the concept of prototypical parts introduced in ProtoPNet (Chen et al., 2019) to represent the activation patterns of networks. Several extensions have been proposed, including TesNet (Wang et al., 2021) and Deformable ProtoPNet (Donnelly et al., 2022), which exploit orthogonality in prototype construction. ProtoPShare (Rymarczyk et al., 2021), ProtoTree (Nauta et al., 2021b), ProtKNN (Ukai et al., 2022), and ProtoPool (Rymarczyk et al., 2022d) reduce the number of prototypes used in classification. Other methods consider hierarchical classification with prototypes (Hase et al., 2019), prototypical part transformation (Li et al., 2018), and knowledge distillation techniques from prototypes (Keswani et al., 2022). Prototype-based solutions have been widely adopted in various applications such as medical imaging (Afnan et al., 2021; Barnett et al., 2021; Kim et al., 2021; Rymarczyk et al., 2022b), time-series analysis (Gee et al., 2019), graph classification (Rymarczyk et al., 2023a; Zhang et al., 2022), semantic segmentation (Sacha et al., 2023), and class incremental learning (Rymarczyk et al., 2023b).

However, prototypical parts still need to be improved, especially regarding the understandability and clarity of the underlying features responsible for the prediction (Kim et al., 2022). Issues such as spatial misalignment of prototypical parts (Carmichael et al., 2024; Sacha et al., 2024) and imprecise visualization techniques (Gautam et al., 2023; Xu-Darme et al., 2023) have been identified. There are also post-hoc explainers analyzing visual features such as color, shape, and textures (Nauta et al., 2021a), and approaches using multiple image patches to visualize the prototypical parts (Ma et al., 2024; Nauta et al., 2023). In this work, we address the ambiguity of prototypical parts by presenting a dedicated architecture, LucidPPN, that detects separate sets of prototypes for shapes with textures and another set for colors. This approach aims to enhance the interpretability and clarity of the interpretations.

**Usage of low-level vision features for image classification.** Multiple approaches to extracting features based on texture (Armi & Fekri-Ershad, 2019; Haralick et al., 1973), shape (Khan et al., 2012; Mingqiang et al., 2008), and color (Chen et al., 2010; Kobayashi & Otsu, 2009) have been proposed before the deep learning era, based on the knowledge about human perception (Fan et al., 2017). Similar features are trained by shallow layers of deep networks, which can be visualized with methods such as (Zeiler, 2014; Springenberg et al., 2014). However, while these methods effectively illustrate low-level features, they struggle with deeper layers, where the visualized concepts entangle multiple visual features and lead to ambiguous explanations. Similar behavior can be observed in recent eXplainable AI (XAI) methods (Basaj et al., 2021; Laina et al., 2022; Rymarczyk et al., 2022a; Zieliński & Górszczak, 2021; Nauta et al., 2021a). By using two branches, one for color and one for remaining visual features, our method explicitly disentangles these visual features, reducing ambiguity of explanations based on high-level concepts.

## 3 METHOD

### 3.1 PRELIMINARIES

**Problem formulation.** Our objective is to train a fine-grained classification model based on prototypical parts, which accurately predicts one of $M$ subtly differentiating classes. We use $N$ image-label pairs $\{(x_0, y_0), \ldots, (x_N, y_N)\} \subset I \times \{1, \ldots, M\}$ as a training set to obtain the model returning highly accurate predictions and lucid explanations. For this, we separate color from other visual features at the input and process them through two network branches with separate sets of PPs.

**PDiscoNet.** PDiscoNet (van der Klis et al., 2023) generates segmentation masks of object parts, used in training of LucidPPN to align $K$ successive prototypical parts of each class with $K$ successive object parts. We decided to use it instead of human annotators because it is more efficient and cost-effective. However, it can be replaced with any method of object part segmentation due to the modularity of our approach.

PDiscoNet model $f_{Disco}$ utilizes a convolutional neural network (CNN) to generate a feature map $Z_{Disco} = [z_{ij}]_{i,j} \in (\mathbb{R}^{D_{Disco}})^{H_{Disco} \times W_{Disco}}$ from a given image $x$. Each of $H_{Disco} \times W_{Disco}$ vectors from such feature map is then compared to trainable vectors $q^k \in \mathbb{R}^{D_{Disco}}$ representing $K$ object parts and background, using similarity based on Euclidean distance

$$t_{ij}^k = \frac{\exp(-||z_{ij} - q^k||^2)}{\sum_{k'=1}^{K+1} \exp(-||z_{ij} - q^{k'}||^2)}, \tag{1}$$

for $i = 1, \ldots, W_{Disco}$ and $j = 1, \ldots, H_{Disco}$, and $k \in \{1, \ldots, K+1\}$. This way, we obtain an attention map $T^k = [t_{ij}^k]_{i,j} \in \mathbb{R}^{H_{Disco} \times W_{Disco}}$ for each object part and background. Such attention maps are multiplied by feature map $Z_{Disco}$ and averaged to obtain one vector per object part. Those vectors are passed to the classification part of PDiscoNet, which involves learnable modulation vectors and a linear classifier.

A vital observation is that the maps $T^k$ continuously split the image into regions corresponding to discovered object parts thanks to a well-conceived set of loss functions added to the usual cross-entropy. They assure the distinctiveness, consistency, and presence of the semantic regions. Yet, the only annotations used in training are the class labels.

In the subsequent sections, we ignore the PDiscoNet predictions $P_{Disco}$, using only the attention maps $T^k$, which we will call *segmentation masks* from now on.

### 3.2 LUCIDPPN

#### 3.2.1 ARCHITECTURE

LucidPPN is a deep architecture, presented in Figure 3, consisting of two branches: one for revealing information about shape and texture (*ShapeTexNet*), and the second dedicated to color (*ColorNet*). That is why *ShapeTexNet* operates on grayscaled input, while *ColorNet* uses aggregated information about the color.

**_ShapeTexNet._** A grayscaled version of image $x$ is obtained by converting its channels $x = (r, g, b)$ to $x_S = (w, w, w)$, where $w = 0.299r + 0.587g + 0.114b$. This formula approximates human perception of brightness (Pratt, 2013) and is a default grayscale method used in computational libraries, such as PyTorch (Paszke et al., 2019).

Grayscaled image $x_S$ is fed to a convolutional neural network backbone $f_{S_b}$. For this purpose, we adapt the ConvNeXt-tiny (Liu et al., 2022) without classification head and with increased stride at the two last convolutional layers to increase the resolution of the feature map, like in PIP-Net (Nauta et al., 2023). As an output of $f_{S_b}$, we obtain a matrix of dimension $(D \times H \times W)$, which is projected to dimension $KM \times H \times W$ using $1 \times 1$ convolution layer $f_{S_{cl}}$ (where $K$ is the number of object parts and $M$ is the number of classes) so that each prototype has its channel. Then, it is reshaped to the size of $K \times M \times H \times W$ on which we apply the sigmoid. As a result, we obtain *ShapeTexNet feature map* defined as

$$Z_S = [z_S^{km}]_{k,m} = \sigma(f_{S_{cl}}(f_{S_b}(x_S))) = f_S(x_S) \in (\mathbb{R}^{H \times W})^{K \times M}. \tag{2}$$

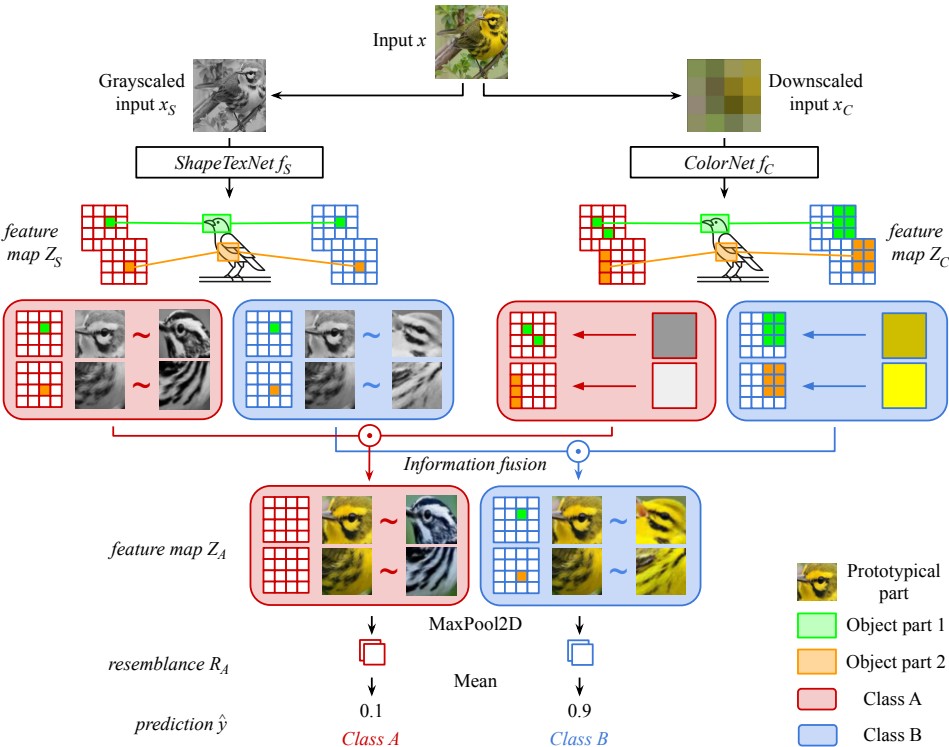

Figure 3: LucidPPN architecture consists of two branches: *ColorNet* and *ShapeTexNet* that encode color and shape with texture in feature maps $Z_C$ and $Z_S$, respectively. Thanks to a special type of training each channel of a feature map corresponds to similarity to a specific object part of a given class. In this image, green and orange correspond to two object parts: head and belly, and red and blue correspond to classes A and B. Therefore, each feature map consists of four channels for head of A, belly of A, head of B, and belly of B. Corresponding channels from both branches are multiplied to obtain feature map $Z_A$, which is then pooled with maximum to obtain the resemblance of prototypical parts fusion and aggregated through mean to obtain final logits.

Thus, we link each map $z_S^{km}$ to a unique *prototypical part* of an object part $k$ for class $m$, from which we compute *ShapeTexNet resemblance* $R_S = [r_S^{km}]_{k,m} \in [0,1]^{K \times M}$, where

$$r_S^{km} = \text{MaxPool2D}(z_S^{km}). \tag{3}$$

Finally, we obtain *ShapeTexNet predictions* $P_S = [p_S^m]_m \in [0,1]^M$ by taking the mean over the resemblance of all parts of a specific class

$$p_S^m = \frac{1}{K} \sum_{k=1}^{K} r_S^{km}. \tag{4}$$

**ColorNet.** To obtain aggregated information about color, as an input of *ColorNet*, image $x$ is downscaled through bilinear interpolation to $H \times W$ resolution, marked as $x_C$. Then, $x_C$ is passed to convolutional neural network $f_C$, composed of six $1 \times 1$ convolutional layers with ReLU activations, except the last layer after which we apply sigmoid. This way, we process each input pixel of $x_C$ separately, taking into account only its color. As a result, we obtain *ColorNet feature map*

$$Z_C = [z_C^{km}]_{k,m} = f_C(x_C) \in (\mathbb{R}^{H \times W})^{K \times M}. \tag{5}$$

Analogously to *ShapeTexNet*, each dimension in the feature map is related to a unique *prototypical part* of an object part $k$ in class $m$. Hence, as before, we calculate *ColorNet resemblance* $R_C = [r_C^{km}]_{k,m} \in [0,1]^{K \times M}$, where

$$r_C^{km} = \text{MaxPool2D}(z_C^{km}). \tag{6}$$

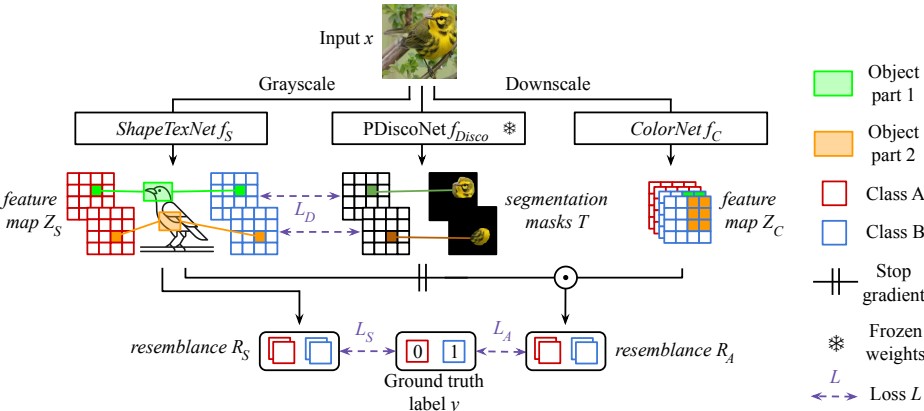

Figure 4: LucidPPN training schema. We use segmentation masks from PDiscoNet to align the activation of prototypical parts with object parts. Additionally, we enforce the *ShapeTexNet* to encode as much predictive information as possible through the usage of $L_S$. Lastly, we learn how to classify images through $L_A$ which is a binary cross-entropy loss.

***Information fusion and prediction.*** To obtain *aggregated feature map* $Z_A = [z_A^{km}]_{k,m} \in (\mathbb{R}^{H \times W})^{K \times M}$ from both branches, we multiply the *ShapeTexNet feature map* with *ColorNet feature map* element-wise

$$z_A^{km} = z_S^{km} \odot z_C^{km}, \qquad (7)$$

and define *aggregated resemblance* $R_A = [r_A^{km}]_{k,m} \in [0,1]^{K \times M}$ as

$$r_A^{km} = \mathrm{MaxPool2D}(z_A^{km}). \qquad (8)$$

The final predictions $\hat{y} = [\hat{y}^m]_m \in [0,1]^M$ for all classes are obtained by averaging $r_A^{km}$ over class-related parts

$$\hat{y}^m = \frac{1}{K} \sum_{k=1}^{K} r_A^{km}. \qquad (9)$$

### 3.2.2 TRAINING

As a result of LucidPPN training (presented in Figure 4), we aim to achieve three primary goals: 1) obtaining a high-accuracy model, 2) ensuring the correspondence of prototypical parts to object parts, 3) and disentangling color information from other visual features. To accomplish these goals, we design three loss functions: 1) prototypical-object part correspondence loss $L_D$, 2) loss disentangling color from shape with texture $L_S$, 3) and classification loss $L_A$ that contribute to the final loss

$$L = \alpha_D L_D + \alpha_S L_S + \alpha_A L_A, \qquad (10)$$

where $\alpha_D, \alpha_S, \alpha_A$ are weighting factors whose values are found through hyperparameter search. The definition of each loss component is presented in the following paragraphs. Please note that we assume that PDiscoNet was already trained, and we denote $\bar{y} \in \mathbb{B}^M$ as a one-hot encoding of $y$.

***Correspondence of prototypical parts to object parts.*** To ensure that each prototypical part assigned to a given class corresponds to distinct object parts, we define the prototypical-object part correspondence loss $L_D$. This function leverages *segmentation masks* $T^k$ from PDiscoNet to align the activations of prototypical parts, represented by the *ShapeTexNet feature map* $Z_S$ with the locations of object parts. Hence, the activations from the *aggregated feature map* $Z_A$ will be aligned with these object parts. It is defined as

$$L_D = \frac{1}{K} \sum_{k=1}^{K} \mathrm{MBCE}\left(Z_S^{ky}, T^k\right), \qquad (11)$$

where $\mathrm{MBCE}(u, v)$ is defined as the mean binary cross-entropy loss between two maps $u, v \in [0,1]^{H \times W}$.

$$\mathrm{MBCE}(u, v) = \frac{1}{HW} \sum_{i=1}^{H} \sum_{j=1}^{W} \mathrm{BCE}(u_{ij}, v_{ij}), \qquad (12)$$

and $y$ is the ground truth class. We align only the maps corresponding to $y$ because the prototypical parts assigned to other classes should not be highly activated.

**Disentangling color from other visual infromation.** To maximize the usage of information about shape and texture during the classification with prototypical parts, we maximize the accuracy of the *ShapeTexNet* through the usage of binary cross-entropy as classification loss function on *ShapeTexNet resemblances* values

$$L_S = \frac{1}{KM} \sum_{m=1}^{M} \sum_{k=1}^{K} \text{BCE}(r_S^{km}, \bar{y}_m). \tag{13}$$

**Classification loss.** Lastly, to ensure the high accuracy of the model and to combine information from *ColorNet* and *ShapeTexNet*, we employ binary cross-entropy on *aggregated resemblances* as our classification loss

$$L_A = \frac{1}{KM} \sum_{m=1}^{M} \sum_{k=1}^{K} \text{BCE}(r_A^{km}, \bar{y}_m). \tag{14}$$

### 3.3 PREDICTION INTERPRETATION

LucidPPN adopts the definition of prototypical parts from PIP-Net (Nauta et al., 2023), where each prototypical part is represented by ten patches, typically activated by ten colored images from the training set. However, in LucidPPN , the visualization must demonstrate how each prototypical part is disentangled into color and shape with texture features. That is why we propose a method to present the disentangled visual features of a prototypical part by combining five grayscale patches, a color bar, and five colored patches. The grayscale and colored patches are selected from the training images with the highest *ShapeTexNet resemblance* and *aggregated resemblance*, respectively. The color bar is created by sampling RGB color values from the ten colored patches with the highest *aggregated resemblance* and projecting them using t-SNE (Van der Maaten & Hinton, 2008). Moreover, in contrast to PIP-Net, LucidPPN creates prototypical parts corresponding to the same object parts in all classes. Therefore, we can use the information about the specific object part location to enrich the explainability.

**Local (prediction) interpretation.** Figure 2 demonstrates how LucidPPN classifies a specific sample $x$ into class $\hat{y}$ by examining the prototypical parts assigned to $\hat{y}$ that are disentangled into color and other visual features. The views are enhanced with pointers to regions of highest *aggregated resemblance*, clearly associated with the object parts.

**Comparison explanation.** Users may wish to inspect and compare local explanations for multiple classes. LucidPPN facilitates this comparison by allowing users to compare prototypical parts of corresponding object parts, making the process intuitive, as shown in pplementary Figure 8.

**Class (global) characteristic.** Disentangled prototypical parts corresponding to object parts reveal the patterns the model uses to classify a given class. This enables the identification of texture and shape features, as well as colors (see Sup. Figure 21), that describe a class without the need to analyze the final-layer connections, unlike other prototypical part-based approaches (Chen et al., 2019; Donnelly et al., 2022; Ma et al., 2024; Nauta et al., 2023; Rymarczyk et al., 2021; 2022c).

## 4 EXPERIMENTAL SETUP

**Datasets.** We train and evaluate our model on four datasets: CUB-200-2011 (CUB) with 200 bird species (Wah et al., 2011), Stanford Cars (CARS) with 196 car models (Krause et al., 2013), Stanford Dogs (DOGS) with 120 breeds of dogs (Khosla et al., 2011), and Oxford 102 Flower (FLOWER) with 102 kinds of flowers (Nilsback & Zisserman, 2008). More details on image preprocessing are in the Supplement.

**Implementation details.** Trainings are repeated 3 times. We made the code public. The size of *ShapeTexNet feature map* is $768 \times 28 \times 28$. The channel number of *ColorNet*'s convolutional layers is $20, 50, 150, 200, 600, K \cdot M$. The values of loss weights are found through hyperparameter search ($\alpha_D = 1.4, \alpha_S = 1.0, \alpha_A = 1.0$). Details are in the Supplement.

**Metrics** During the evaluation, we report the top-1 accuracy classification score. Additionally, we measure the quality of *prototypical parts* alignment with object parts by calculating intersection-over-union (IoU). In PDiscoNet, the segmentation map is the size of

Table 1: Comparison of accuracy of PPs-based models on 4 datasets. LucidPPN achieves competitive results to all methods, and SOTA on 2 datasets. Note that, LucidPPN is trained with $K = 12$, and "–" means that the model did not converge during training when using the code provided by the authors.

|  | CUB | CARS | DOGS | FLOWER |
|---|---|---|---|---|
| ProtoPNet (Chen et al., 2019) | 79.2 | 86.1 | 77.4±0.2 | 92.1±0.3 |
| ProtoTree (Nauta et al., 2021b) | $82.2 \pm 0.7$ | $86.6 \pm 0.2$ | – | – |
| ProtoPShare (Rymarczyk et al., 2021) | 74.7 | 86.4 | 74.1±0.3 | 90.3±0.2 |
| ProtoPool (Rymarczyk et al., 2022c) | $\mathbf{85.5 \pm 0.1}$ | $88.9 \pm 0.1$ | 71.7±0.2 | 92.7±0.1 |
| PIP-Net (Nauta et al., 2023) | $84.3 \pm 0.2$ | $88.2 \pm 0.5$ | $\mathbf{80.8 \pm 0.4}$ | $91.8 \pm 0.5$ |
| LucidPPN | $81.5 \pm 0.4$ | $\mathbf{91.6 \pm 0.2}$ | $79.4 \pm 0.4$ | $\mathbf{95.0 \pm 0.3}$ |

the input image, while our activation map is the size of the latent space. Hence, we are downsizing the segmentation map to $26 \times 26$ resolution to match its dimensions with the activation map before calculating the IoU between the corresponding patches of both maps.

**User studies.** Using ClickWorker System[3], we run two user studies to compare the quality of patch-based prototypes and the influence of disentangled resemblance scores provided by LucidPPN. For the first study, we collect the testing examples from CUB which are correctly classified by PIP-Net, *single branch* CNN[4] and LucidPPN. These are joined with information about the two most probable classes per model and associated prototypical parts. Ninety workers (30 per method) answer the survey, which consists of 10 questions. They are asked to predict the model's decision based on the evidence for the top two output classes without the numerical scores. This approach mimics the user study presented in HIVE (Kim et al., 2022) and is also inspired by the study performed in (Ma et al., 2024). In the second study, we also collect images from CUB. This time we join them with prototypical parts of the correctly predicted class and one other class. Each of the forty workers answers 10 questions in which he/she rates from 1 (least) to 5 (most) to assess the influence of the color features on the model's prediction. The users give ratings based on LucidPPN prototypical parts visualization, with or without included numerical resemblance scores. More details and the survey templates are in the Supplement.

## 5 Results

In this section, we show the effectiveness of LucidPPN, the influence of the color disentanglement in the processing on the model's performance, and the results related to the interpretability of learned prototypical parts based on the user study.

**Comparison to other PPs-based models.** In Table 1 we compare the classification quality of LucidPPN and other PPs-based methods. We present the mean accuracy and standard deviation. We report best performing LucidPPN, which in the case of

Table 2: Comparison of the accuracy of *ShapeTexNet* to LucidPPN. Integrating color with other visual features proves advantageous for datasets containing objects found in nature. However, for the CARS dataset, adding color information does not enhance the model's performance. This is because color is not a significant feature when classifying vehicles, as the same car model can appear in various colors.

|  | CUB | CARS | DOGS | FLOWER |
|---|---|---|---|---|
| *ShapeTexNet* | 80.4 | **91.7** | 78.6 | 93.6 |
| LucidPPN | **81.8** | **91.7** | **78.9** | **95.3** |

all datasets was trained with fixed $K = 12$. Our LucidPPN achieves the highest accuracy for CARS and FLOWER datasets, and competitive results on CUB and DOGS.

**Color impact.** The influence of *ColorNet* on LucidPPN predictions is shown in Table 2. We compare the accuracy of *ShapeTexNet* with the LucidPPN predictions. The *information fusion* enhances the results on the CUB, DOGS, and FLOWER. However, it does not affect performance on the CARS. This can be attributed to the characteristics of the CARS dataset, where vehicles of the same model can differ in colors, indicating that color is not

---

[3]https://www.clickworker.com/

[4]For ablation analysis we also report results of a *single branch* CNN which has the same architecture as *ShapeTexNet* and receives colored images as input. Its local interpretation is visualized similarly to LucidPPN, but without the gray patches and color bar as presented in Sup. Figure 13

critical for this task. This contrasts with the fine-grained classification of natural objects, such as birds and flowers, where color plays a significant role.

In Table 3 we show the results of experiments aiming to analyze how the model is susceptible to the change of the color on the image. We report the accuracy of PIP-Net, *ShapeTexNet*, and LucidPPN on original and hue-perturbed images from the CUB dataset. One can notice that PIP-Net is highly dependent on color information and its score drops by over 37% after perturbation. At the same time *ShapeTexNet* is immune to this transformation, while LucidPPN loses approximately 12.5% accuracy because of it. To alter hue we randomly rotate hue values in the HSV color space. After rotation, we adjust the luminosity of each pixel by proportionately scaling its RGB channels. This step is key to modifying the hue without changing the brightness perceived by humans.

**User studies.** Statistics from the user study assessing the lucidity of explanations generated by LucidPPN, *single branch*, and PIP-Net are in Figure 5 and Supplementary Table 5. We report the mean user accuracy with a standard deviation and *p*-values. Users basing their responses on LucidPPN explanations score significantly better than both PIP-Net and random guess baselines. Additionally, we conclude that most of the accuracy in this user study can be attributed to the PDiscoNet part supervision as *single branch* scores similarly to

Table 3: Robustness of the model to changes in image color. When the hue value is perturbed, the accuracy of PIP-Net drops significantly. In contrast, the accuracy drop for LucidPPN is only half as much, and for *ShapeTexNet* none.

|              | Original | Hue-perturbed |
|--------------|----------|---------------|
| PIP-Net      | 83.9     | 53.0          |
| *ShapeTexNet*| 80.3     | 80.3          |
| LucidPPN     | 81.9     | 71.7          |

LucidPPN, without a statistically significant difference. While both of our explanation variants with prototypical parts corresponding to the same object parts prove to be more intuitive for users, we also want to highlight the advantage of using full LucidPPN over *single branch*. To this end, in Supplementary Figure 7 and Supplementary Table 6 we show the outcomes of the study evaluating the user's ability to recognize the importance of color features in LucidPPN's decisions. Users without information about resemblance values struggle in this task achieving the same performance as if they answered at random. In contrast, users provided with the resemblances in LucidPPN visualizations score 23% better. Note that neither *single branch* nor PIP-Net gives the disentangled resemblance values. In both studies, we perform a one-sided *t*-test and one-sample *t*-test to compare methods against each other and 50% accuracy, respectively. More details can be found in the Supplement.

## 6 Ablation and analysis

In this section, we examine how LucidPPN's performance is impacted by object part supervision and the weights of the loss function components.

**Influence of part supervision on the performance of LucidPPN.** One of the features of LucidPPN is object part supervision based on PDiscoNet. To check its influence on the PPs-based model without disentanglement, in Table 4 we compare the accuracy of a *single branch* to LucidPPN and PIP-Net. The *single branch* scores better than both models. The disentanglement in LucidPPN causes a small (<6%) or negligible drop in accuracy while offering more insights from the model.

**Loss weighting.** In Figure 6 we investigate the impact of the loss weight $\alpha_D$, which is responsible for

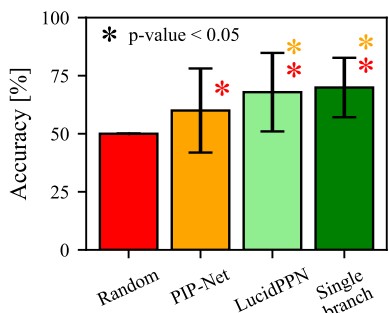

Figure 5: User study results show that users based on LucidPPN explanations outperform those with explanations from PIP-Net to a statistically significant degree.

prototypical-object parts alignment, on training outcomes. In this analysis, the weights of the other losses are fixed at $\alpha_S = \alpha_C = 1$. We evaluate the accuracy and intersection-over-union (IoU) between the highest activated *ShapeTexNet feature map* and PDiscoNet's *segmentation masks* for each object part. The results show that increasing $\alpha_D$ enhances the IoU, but after a certain point, it gradually reduces accuracy. Notably, omitting the

loss $\alpha_D$ (part supervision) significantly diminishes the network's classification performance and makes the learned prototypical parts collapse into a single, most descriptive one as presented in Supplementary Figure 10. While other works address this issue by adding novel regularization losses (Nauta et al., 2023; Wang et al., 2021), these solutions fail to ensure consistency of the considered parts across different classes.

We provide additional results in the Supplementary Materials. They include ablations on the LucidPPN's backbone, the size of a *ColorNet*, type of input color representation, number of object parts $K$, and different training schedules for LucidPPN's branches. Also, we show examples of PIPNet failures mitigated by LucidPPN, and we discuss the reasons for introducing $L_S$.

## 7 CONCLUSIONS

In this work, we propose LucidPPN, an inherently interpretable model that uses prototypical parts to disentangle color from other visual features in its explanations. Our extensive results demonstrate the effectiveness of our method, and user studies confirm that

Table 4: Accuracy of PIP-Net, LucidPPN, and a *single branch* CNN supervised by PDiscoNet.

|  | CUB | CARS | DOGS | FLOWER |
|---|---|---|---|---|
| PIP-Net | 84.3 | 88.2 | 80.8 | 91.8 |
| LucidPPN | 81.5 | 91.6 | 79.4 | 95.0 |
| *single branch* | **86.6** | **91.9** | **82.7** | **95.6** |

our explanations are less ambiguous than those from PIP-Net. In future research, we aim to further refine the model architecture to separately process shape and texture features, as well as analyze different visualization strategies of disentanglement and their recognition by the users. Additionally, we plan to explore the human perception system in greater depth to inform the design of the next generation of interpretable neural network architectures.

**Limitations.** Our work faces a significant constraint: while our designed mechanism adeptly disentangles color information from input images, it cannot currently extract other crucial visual features such as texture, shape, and contrast. This highlights a broader challenge within the field: the absence of a universal mechanism capable of encompassing diverse visual attributes. Furthermore, our approach inherits limitations from other PPs-based architectures, including issues such as spatial misalignment (Sacha et al., 2024), the non-obvious interpretation of PPs (Ma et al., 2024) and those of PIP-Net (Nauta et al., 2023). The latter could be addressed with textual descriptions of concepts discovered by PPs. Lastly, LucidPPN increases the transparency of the decision made by the deep neural networks however it still has a performance gap to black-box models, or even to those offering some

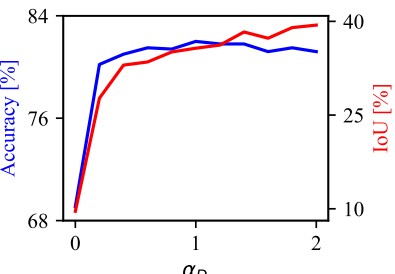

Figure 6: Influence of the weight of prototypical-object part correspondence loss on accuracy and Intersection-over-Union (IoU). An increase of $\alpha_D$ improves IoU but at a certain point gradually reduces accuracy.

insights into the model reasoning process such as PDiscoNet (van der Klis et al., 2023). This shall be under further investigation to fill this performance gap if possible.

**Broader Impact.** Our work advances the field of interpretability, a crucial component for trustworthy AI systems, where users have the right to understand the decisions made by these systems (Kaminski, 2021; Tabassi, 2023). LucidPPN enhances the quality of explanations derived from PPs-based neural networks, which are among the most promising techniques for ante-hoc interpretability methods. Consequently, it can facilitate the derivation of scientific insights and the creation of better human-AI interfaces for complex, high-stakes applications.

Additionally, LucidPPN provides visual characteristics for PPs, which are especially beneficial in domains lacking standardized semantic textual descriptions of concepts. This is particularly useful in fields such as medicine, where it aids in analyzing radiology and histopathology images.

## Reproducibility Statement

We ensured that our experiments are reproducible by thoroughly describing them in Section 4 and the Supplement. Additionally, the Supplementary Materials include the code used to perform the experiments, along with a `README.md` file providing further instructions.

## Acknowledgements

The work of M. Pach, K. Lewandowska and B. Zieliński work was funded by National Centre of Science (Poland) grant no. 2022/47/B/ST6/03397. The work of D. Rymarczyk was funded by National Centre of Science (Poland) grant no. 2022/45/N/ST6/04147. The work of J. Tabor was funded by National Centre of Science (Poland) grant no. 2023/49/B/ST6/01137.

We gratefully acknowledge Polish high-performance computing infrastructure PLGrid (HPC Centers: ACK Cyfronet AGH) for providing computer facilities and support within computational grant no. PLG/2023/016555. Some experiments were performed on servers purchased with funds from the Priority Research Area (Artificial Intelligence Computing Center Core Facility) under the Strategic Programme Excellence Initiative at Jagiellonian University.

We are grateful to Jakub Pach and Tomasz Pach for their assistance in composing images for the survey according to our developed template.

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

# SUPPLEMENT FOR LUCIDPPN: UNAMBIGUOUS PROTOTYPICAL PARTS NETWORK FOR USER-CENTRIC INTERPRETABLE COMPUTER VISION

## MORE DETAILS ON DATA PREPROCESSING

In training, we apply transformations as follows: `Resize(size=224+8)`, `TAWideNoColor()`, `RandomHorizontalFlip()`, `RandomResizedCrop(size=(224, 224), scale=(0.95, 1.)`, where `TAWideNoColor()` is the same variation of TrivialAugment augmentation as in PIP-Net. Additionally, the image entering the *ShapeTexNet* is normalized with `Normalize(mean=0.445, std=0.269)` after being converted to grayscale. At test time and when finding the prototypical parts patches, we only apply `Resize(size=224)` followed by grayscaling and normalization in case of *ShapeTexNet* input. The CUB images used for training and evaluation are first cropped to the bounding boxes similarly to other PP-based methods.

We do not modify any parameters in PDiscoNet. CUB settings are used for datasets not trained in the PDiscoNet paper. For efficiency, we generated and saved the segmentation masks to avoid inferencing PDiscoNet during LucidPPN' training.

## MORE DETAILS ON EXPERIMENTAL SETUP

The networks (*ShapeTexNet* and *ColorNet*) are optimized together in minibatches of size 64 for 40 epochs using AdamW (Loshchilov & Hutter, 2017) optimizer with beta values of 0.9 and 0.999, epsilon of $10^{-8}$, and weight decay of 0. The learning rate of *ShapeTexNet* parameters is initialized to 0.002 and lowered to 0.0002 after 15 epochs. The learning rate of the *ColorNet* is fixed at 0.002. We freeze the weights of *ShapeTexNet* backbone for the first 15 epochs as a warm-up stage similar to other PPs-based approaches (Chen et al., 2019; Nauta et al., 2023; Rymarczyk et al., 2022c).

## MORE DETAILS ON COMPUTING RESOURCES

We ran our experiments on an internal cluster and a local cloud provider, a single GPU, it was either NVIDIA A100 40GB or NVIDIA H100 80GB. The node we ran the experiments on has 40GB of RAM and an 8-core CPU. The model on average trains for 3 hours.

## MORE DETAILS ON USER STUDIES WITH EXEMPLARY SURVEYS.

Each worker answering a short 10-question survey was paid 1.50 euros. Questions between users may differ as they are randomly composed. Participants are gender-balanced and have ages from 18 to 60.

**User study on quality of prototypical parts.** For PIP-Net, we randomly select samples with $K' = 4, 3, 2, 1$ in the proportion of $5 : 3 : 2 : 1$ based on the frequency of occurrence as PIP-Net doesn't have the same number of prototypical parts assigned to data classes. The LucidPPN pieces of evidence for classes in the same samples always show four prototypical parts as we use a model trained with $K = 4$ here.

Example surveys for LucidPPN, PIP-Net, and *single branch* are presented in Figures 25 to 37, 38 to 50, and 51 to 63, respectively.

**User study on the importance of disentangled visual features.** Because we focus on the influence of the color features, we use visualizations with a random single object and prototypical part to let the user focus on the influence of the color. When gathering samples for the survey, we make sure that for nearly half of them color was important for the correct prediction, and for half of them, it was not. We define that the color was important, when LucidPPN was correct, but *ShapeTexNet* was wrong. And, we define that color was unimportant if both outputs were correct.

Example surveys for LucidPPN with color feature scores and without them are presented in Figures 64 to 76, and 77 to 89, respectively.

Detailed results of the user study

In Tables 5 and 6, we present detailed results of the user studies. We also visualize the results of the user study on the importance of scores in the Figure 7.

Table 5: User study results indicate that users based on LucidPPN explanations outperform those with explanations from PIP-Net to a statistically significant degree.

|  | Mean Acc. [%] ± Std. | random | $p$-value PIP-Net | LucidPPN |
|---|---|---|---|---|
| PIP-Net | $60.0 \pm 18.1$ | 0.002 | – | – |
| LucidPPN | $67.9 \pm 16.9$ | $2.13 \cdot 10^{-6}$ | 0.044 | – |
| *single branch* | $69.9 \pm 12.8$ | $1.11 \cdot 10^{-9}$ | 0.008 | 0.299 |

Table 6: Details of the user study about assesing the importance of color.

|  | Mean Acc. [%] ± Std. | random | $p$-value without resemblances |
|---|---|---|---|
| without resemblances | $49.50 \pm 11.3$ | 0.577 | – |
| with resemblances (LucidPPN) | $60.87 \pm 19.9$ | 0.012 | 0.016 |

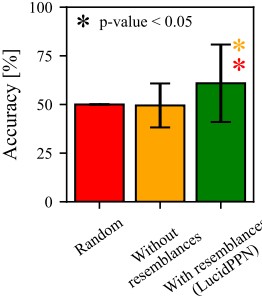

Figure 7: User study shows that disentangled resemblance scores enable users to better understand the relevance of color in model's decisions.

Comparison explanation example

We show how our model can generate explanations by comparison of two potential classes in Figure 8.

Color representation

We have performed an ablation study to evaluate how different color representations of the input $x_c$ influence the model's results. Instead of directly downsizing the RGB image, we first transformed it into HSV space, replaced the S and V values with the Hue value, and then downsized the image. In other words, the input to the network was an image composed of the Hue channel repeated three times. The results demonstrate that LucidPPN with this input still outperforms *ShapeTexNet*, with performance similar to the basic LucidPPN as presented in the Table 7.

ColorNet size

Since the architecture of *ColorNet* may significantly impact LucidPPN's performance, we conducted an ablation study on the architecture's size. Table 8 presents the accuracy comparison for CUB. All layers are $1 \times 1$ convolutions followed by ReLU, except the last layer, which is followed by a sigmoid activation. The results indicate that using at least two layers

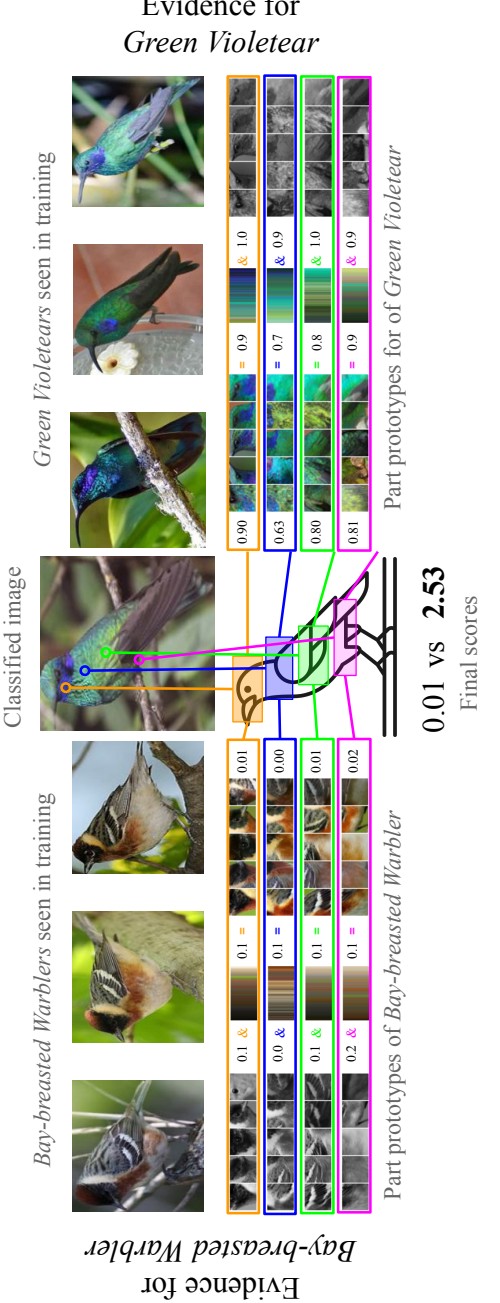

Figure 8: Comparison explanation example. Best viewed in landscape orientation.

to introduce non-linearity is beneficial. Additional layers have a smaller impact but can be added to ensure sufficient expressiveness of the network.

## QUALITATIVE EXAMPLES OF FAILURE CASES WITHOUT DISENTANGLEMENT THAT ARE IMPROVED THROUGH LUCIDPPN

The main goal of the disentanglement is not to improve the accuracy but to provide a better understanding of the model's reasoning based on color and shape with texture information. The explanations containing *ShapeTexNet*, *ColorNet*, and *aggregated resemblances* intro-

Table 7: Accuracy of LucidPPN when ColorNet receives RGB values vs. only hue value. *ShapeTexNet* added for comparison.

| | ColorNet input | CUB | CARS | DOGS | FLOWER |
|---|---|---|---|---|---|
| *ShapeTexNet* | - | $80.1 \pm 0.2$ | $91.7 \pm 0.1$ | $79.0 \pm 0.3$ | $93.6 \pm 0.3$ |
| LucidPPN | RGB | $81.5 \pm 0.4$ | $91.6 \pm 0.2$ | $79.4 \pm 0.4$ | $95.0 \pm 0.3$ |
| | Only Hue | $81.1 \pm 0.4$ | $91.6 \pm 0.2$ | $79.5 \pm 0.2$ | $94.1 \pm 0.5$ |

Table 8: LucidPPN's accuracy for CUB vs the size of *ColorNet*.

| Number of layers | Hidden dimensions | Accuracy [%] |
|---|---|---|
| 1 | - | 80.3 |
| 2 | $(600)$ | 81.4 |
| 4 | $(50, 200, 600)$ | 81.2 |
| 6 | $(20, 50, 150, 200, 600)$ | **81.5** |

duced in this work offer this additional information. Such an insight was missing in the previous prototypical parts models (Chen et al., 2019; Nauta et al., 2023).

Nevertheless, disentanglement can enhance accuracy in scenarios where shape and texture are the primary decision factors, with color serving to refine decisions that are difficult to make based on other features alone. Figure 9 illustrates examples from CARS and CUB where LucidPPN adheres to this principle, whereas the *single branch* CNN does not. Table 9 shows how often *single branch* CNN with colored input misclassifies color-altered images compared to the LucidPPN, and vice versa, for two specific data classes scenarios:

1. For test images of the typically red *Lamborghini Aventador*, which were converted to green and yellow via hue rotation, the *single branch* CNN mistakenly classified these altered images as either the typically green *Lamborghini Gallardo* or the usually yellow *Lamborghini Diablo* 24 times, despite the noticeable differences in shape (e.g., headlights and bumpers). In contrast, LucidPPN correctly classified these altered images. LucidPPN only made such a mistake 3 times, while the *single branch* CNN did not.

2. For test images of the red *Cardinal*, which were similarly converted to yellow and indigo, the *single branch* CNN misclassified the *Cardinal* as any other bird 34 times. LucidPPN made this mistake only once, while the *single branch* CNN was correct in all other instances.

Table 9: Number of found examples for which LucidPPN and *single branch* outperformed each other when asked to predict class in the color altered images.

| | Lamborghini Avendator | Cardinal |
|---|---|---|
| LucidPPN correct but *single branch* wrong | 24 | 34 |
| LucidPPN wrong but *single branch* correct | 3 | 1 |

PROTOTYPICAL PARTS EXAMPLES TRAINED WITHOUT PART SUPERVISION

are presented in Figure 10.

| Original image | Altered images | Predictions | Wrong class guessed by *single branch* |
|---|---|---|---|
| **Correct label:** *Lamborghini Aventador* |  | *single branch*: *Lamborghini Gallardo* LucidPPN: *Lamborghini Aventador* *single branch*: *Lamborghini Diablo* LucidPPN: *Lamborghini Aventador* |  |
| **Correct label:** *Cardinal* |  | *single branch*: *Blue-winged Warbler* LucidPPN: *Cardinal* *single branch*: *Indigo Bunting* LucidPPN: *Cardinal* |  |

Figure 9: Examples of images with altered colors that change the prediction of a single branch CNN include a Lamborghini Aventador (top) and a Cardinal (bottom). Both are incorrectly classified by the single branch CNN, while LucidPPN with color disentangling classifies them correctly.

NUMBER OF PARTS

In Figure 11, we show the impact of choosing a different number of parts $K$. LucidPPN achives high results for all tested $K$, however it is noticeable that increasing $K$ improves classification. Especially on CARS, our method seems to strongly benefit from choosing $K \geq 4$. The reasonably high scores for all $K$ allow for a choice between sparse explanations and higher accuracy.

NEED FOR $L_S$

Many prototypical-parts-based models, such as ProtoPNet (Chen et al., 2019), ProtoPool (Rymarczyk et al., 2022d), and PIP-Net (Nauta et al., 2023), involve complex training schemes with warm-up and pretraining phases. Initially, we believed that *ShapeTexNet* should be pretrained before training *ColorNet*, given that *ShapeTexNet* processes more complex data. However, the ablation study presented in Figure 12 shows that warming up *ShapeTexNet* (or delaying the training of *ColorNet*) is either unnecessary or may even negatively influence color-based explanations.

During the initial development of LucidPPN, we used $L_S$ to guide the learning of *ShapeTexNet* during its warm-up phase. Once we observed that *ShapeTexNet* did not require a warm-up, we switched to jointly using $L_S$ and $L_A$ in training. We found that removing $L_S$ negatively impacted LucidPPN's performance which is presented in Figure 6 in the manuscript. Consequently, we retained $L_S$, as it provides essential guidance for *ShapeTexNet* to effectively extract important features from its more complex input.

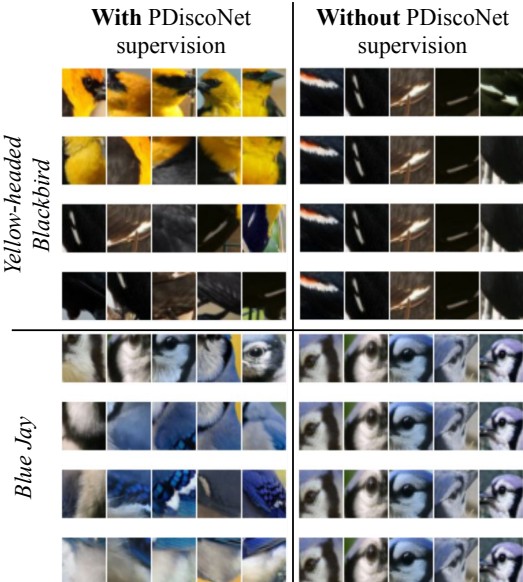

Figure 10: Examples from two classes demonstrate that prototypes learned without PDiscoNet supervision focus on a single object part, in contrast to those more diverse learned by LucidPPN.

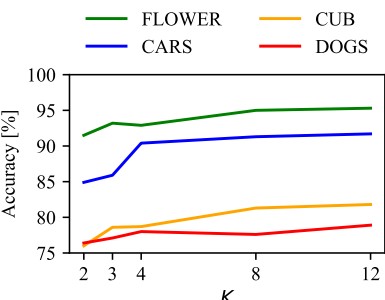

Figure 11: Influence of the number of object parts $K$ on LucidPPN accuracy. Increasing the number of parts improves the accuracy of the model. Note that each dataset is plotted in a unique color.

This need for stronger guidance aligns with observations in multimodal learning, where separate learning of different modality branches maximizes the information extracted from each modality (Wu et al., 2022). Here, we can think of each branch as different modalities. Alternatively, using a weighted average could yield similar accuracy, but it would complicate the final prediction. This approach would necessitate analyzing the contribution of each logit vector separately and understanding their aggregation, with potentially different weights for each dataset, making the process less transparent for the user.

Start of color network training

It is natural to ask whether delaying the start of *ColorNet* optimization could improve LucidPPN. In Figure 12, we report the accuracy and color sparsity after delaying the training of *ColorNet*. The change in classification quality is negligible. However, we observe a drop in color sparsity, indicating that *ColorNet* is less focused on relevant colors. It is important to note that despite the delay, the number of training epochs for *ColorNet* remains constant for comparability.

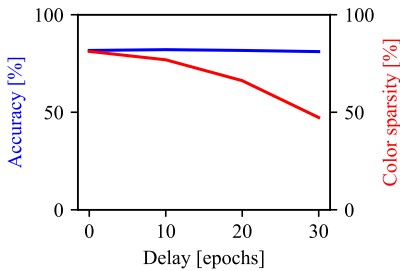

Figure 12: Influence of a delay when *ColorNet* starts to train on LucidPPN 's accuracy and color sparsity. While this delay does not negatively affect accuracy, it results in lower color sparsity. This means that the network is not concentrating on a single color when processing the PP.

LUCIDPPN WITH DIFFERENT BACKBONES

We evaluate LucidPPN on an additional ResNet50 backbone and compare the results to other models and backbones in Table 10. LucidPPN with ResNet50 backbone performs worse than the one with ConvNeXt-tiny, which is similar to PIP-Net. When comparing the results, note that iNaturalist-pretrained backbones have an advantage over ImageNet resulting in a few points higher accuracy.

Table 10: Accuracy for different PP-based methods and backbones. The asterisk means that the used backbone was pretrained on the iNaturalist dataset instead of ImageNet.

|  |  | CUB | CARS | DOGS | FLOWER |
|---|---|---|---|---|---|
| ProtoPNet | ResNet34 | 79.2 | 86.1 | - | - |
| ProtoPShare | ResNet34 | 74.7 | 86.4 | - | - |
| ProtoTree | ResNet50 | $82.2 \pm 0.7^*$ | $86.6 \pm 0.2$ | - | - |
| ProtoPool | ResNet50 | $\mathbf{85.5 \pm 0.1}^*$ | $88.9 \pm 0.1$ | - | - |
| PIP-Net | ResNet50 | $82.0 \pm 0.3^*$ | $86.5 \pm 0.3$ | - | - |
|  | ConvNeXt-tiny | $84.3 \pm 0.2$ | $88.2 \pm 0.5$ | $\mathbf{80.8 \pm 0.4}$ | $91.8 \pm 0.5$ |
| LucidPPN | ResNet50 | $75.5 \pm 1.1$ | $89.0 \pm 0.3$ | $70.8 \pm 0.2$ | $89.5 \pm 0.4$ |
|  | ConvNeXt-tiny | $81.5 \pm 0.4$ | $\mathbf{91.6 \pm 0.2}$ | $79.4 \pm 0.4$ | $\mathbf{95.0 \pm 0.3}$ |

DISCUSSION ON PATCHES OF THE COLOR INFO AND THE PATCH OF GRAY-SCALED INPUT ALIGNED IN THE LATENT SPACE

Using a convolutional backbone, we assume a spatial correspondence between the latent map from *ShapeTexNet* and the input, similar to the approach in ProtoPNet (Chen et al., 2019). As we downsize the colorful image to match the height and width of the activation map, the input dimensions for *ColorNet* are maintained consistently. *ColorNet* employs $1 \times 1$ convolutions to encode color information, ensuring the latent map has the same dimensions as both the downsized input and the *ShapeTexNet* activation map.

Given the use of $1 \times 1$ convolutions on the downsized image and a convolutional backbone for the full-resolution image, we can assume that the $(i, j)$ position on one map corresponds to the $(i, j)$ position on the other. Finally, we extract color information from the latent representation at the same location where the prototypical part is most active, ensuring alignment between the color and shape features.

LOCAL INTERPRETATION OF *SINGLE BRANCH*

An example of prediction interpretation of *single branch* is presented in Figure 13.

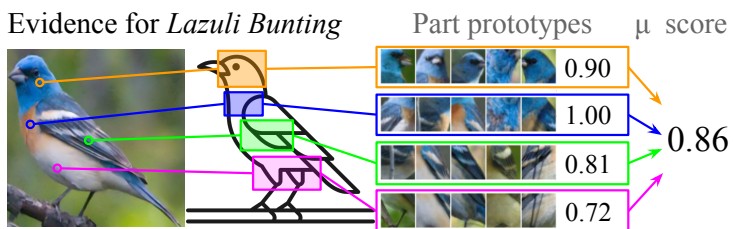

Figure 13: An example local interpretation of *single branch* for *Lazuli Bunting*.

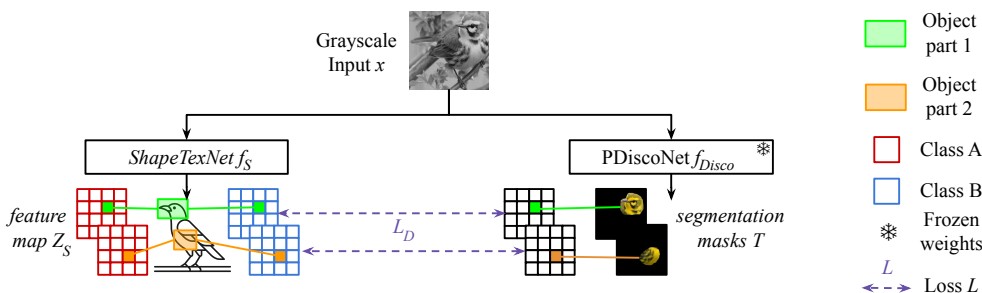

Figure 14: The image illustrates the object-part correspondence loss, which is applied solely to the outputs of *ShapeTexNet* and PDiscoNet. First, we identify object parts through PDiscoNet (e.g., first on the head, second on the wing). Next, we align the corresponding prototypical parts with the object parts identified by the segmentation results through $L_D$ loss.

OBJECT-PART CORRESPONDENCE

On Figure 14 we present how object-part correspondence $L_D$ loss works.

CONSISTENCY AND STABILITY OF PROTOTYPICAL PARTS

One way to evaluate the quality of prototypical parts is to measure their consistency and stability Huang et al. (2023). In Table 11. we present the results of those metrics. The results show that LucidPPN achieves state-of-the-art results on stability without any additional loss components, and is comparable to other metrics when it comes to stability. This improvement is likely due to the enhanced object-part correspondence enabled by its prototypical parts.

COMPUTATIONAL COSTS

In Table 12, we provide computational costs of training LucidPPN when compared to other prototypical-parts-based architectures.

GENERALIZATION TO NOT FINE-GRAINED DATASET

To assess whether LucidPPN generalizes to broader classification tasks (beyond fine-grained datasets), we present results on PartImageNet He et al. (2022). On this dataset, LucidPPN achieves an accuracy of 84.1%, outperforming PIPNet, which achieves 82.8%.

COMPARISON OF EXPLANATION VISUALIZATIONS

In Figures 15, 16, 17, 18, and 19 we compare the decision explanations generated by different methods.

Table 11: Results of LucidPPN on consistency and stability metrics from the work of Huang et al. (2023). The results indicate that LucidPPN is more robust than other prototypical-parts-based approaches and achieves state-of-the-art results for Consistency while still remaining competitive in Stability.

| Method | Consistency | Stability |
|---|---|---|
| ProtoPNet | 28.3 | 56.7 |
| ProtoTree | 16.4 | 23.2 |
| ProtoPool | 35.7 | 58.4 |
| TesNet | 48.6 | 60.0 |
| Deformable ProtoPNet | 44.2 | 53.5 |
| Huang et al. (2023) | 70.6 | **72.1** |
| LucidPPN (our) | **71.2** | 66.3 |

Table 12: Computational costs of prototypical-parts-based methods. One can observe that training of LucidPPN requires fewer hours and less RAM memory than PIP-Net, but more GFLOPs. Generally, LucidPPN and PIPNet require more RAM memory than ProtoPNet and ProtoPool, however they converge faster.

| Method | Training time | GFLOPs for 1 batch of data | Avg. Training Memory Usage |
|---|---|---|---|
| ProtoPNet | 3h | 586 | 4.9GB |
| ProtoPool | 18h | 658 | 14.4GB |
| PIP-Net | 3h | 354 | 41.5GB |
| LucidPPN (our) | 2h | 475 | 22.9GB |

FAITHFULLNESS OF PATCH VISUALIZATIONS

LucidPPN introduces a key difference in the definition of prototypical parts compared to PIPNet. While PIPNet employs Softmax across channels in the latent feature map, LucidPPN uses the sigmoid activation function. The sigmoid function allows each channel's activation to be learned independently, not influenced by the relative activations of other channels. At the same time, Softmax normalization can distort activations by emphasizing values that are only relatively high compared to others, even if they are low in absolute terms. Therefore, using the sigmoid function instead of Softmax, one can easily verify if the image patches selected for visualization are faithful because such patches should have a resemblance score close to 1. In Figure 20, we provide a distribution of the sigmoid function values obtained for patches used in prototype visualization. For LucidPPN trained on the CUB dataset (blue curve), 61.04% of those patches have values above 0.9, which indicates that prototype visualizations are relatively faithful. Moreover, higher faithfulness can be obtained when training with an additional loss component $L_C$ that punishes the model if the sigmoid function value for a given prototype is smaller than 1 for all samples in the batch:

$$L_C = \frac{1}{KM} \sum_{k=1}^{K} \sum_{m=1}^{M} \max_{b \in B} (1 - r_{A,b}^{km}),$$

where $B$ is the number of samples in a batch and $r_{A,b}^{km}$ is the value of $r_A^{km}$ for sample $b$ in the batch. As we observe in Figure 20 (green and yellow curves), the distribution of sigmoid function values moves right with increasing weight $\alpha_C$ of $L_C$. However, it also comes with a small decrease in accuracy.

PRUNING PROTOTYPES WITH LESS FAITHFUL VISUALIZATIONS

To increase the faithfulness of LucidPPN, we analyze the effects of pruning the prototypes with less faithful representation (those with resemblance scores $< 0.9$). As shown in Table 13, LucidPPN accuracy after pruning drops only by around 2% (from 81.6% to 79.3%). However, interestingly, the accuracy stays the same for $L_C = 0.05$. It suggests that combi-

nation of using loss $L_C$ and applying the pruning allows to enforce high resemblance scores
($>0.9$) of visualized patches without sacrificing on the accuracy.

### REASON BEHIND USING THE BINARY CROSS ENTROPY WITH SIGMOID INSTEAD OF THE CROSS ENTROPY WITH SOFTMAX

The intuition behind Binary Cross Entropy (BCE) usage is rooted from multilabel classification. To some degree, ShapeTexNet operates in a multilabel setting from the prototypical parts perspective, as they may match multiple classes. Hence, to enable multiple classes having high similarity to the same prototypical parts, we use sigmoid instead of softmax when computing the feature maps. This necessitates a shift from Cross-Entropy (CE) to Binary Cross-Entropy (BCE) because CE would then solely maximize the activation of the correct class while ignoring crucial signals from negative classes. Another reason behind our choice is to make it easier to verify the faithfulness of visualizations, as the sigmoid function allows each channel's activation to be learned independently, not influenced by the relative activations of other channels. While, Softmax normalization used with CE can distort activations by emphasizing values that are only relatively high compared to others, even if they are low in absolute terms.

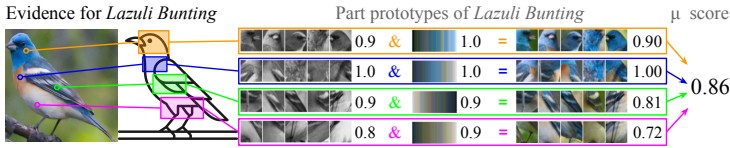

Figure 15: Local interpretation visualization in LucidPPN

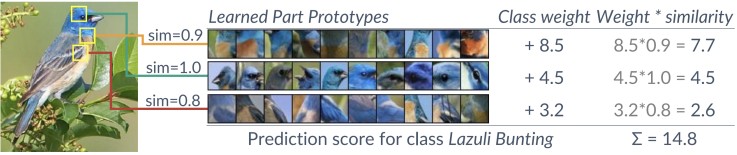

Figure 16: Local interpretation visualization in PIP-Net

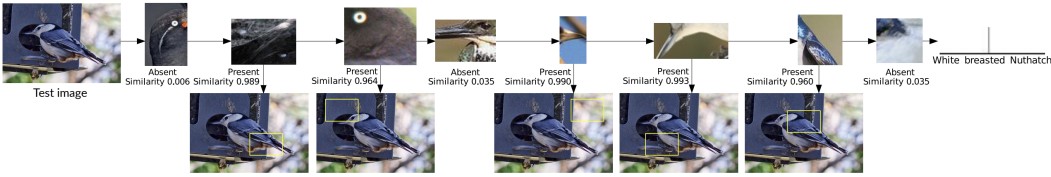

Figure 17: Local interpretation visualization in ProtoTree

### GLOBAL CHARACTERISTICS EXAMPLES

We present global characteristics for different datasets in Figures 21, 22, 23, 24, 25.

Why is that *Ford Freestar Minivan 2007*?

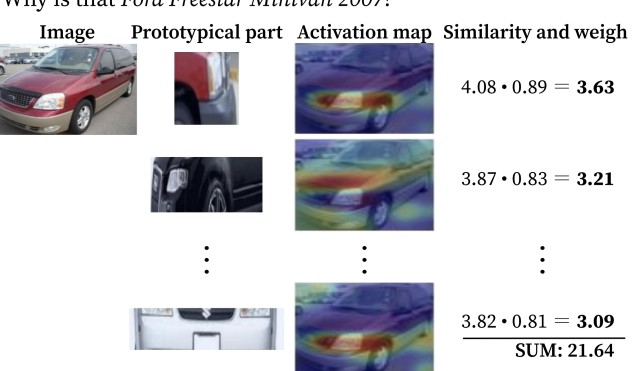

Figure 18: Local interpretation visualization in ProtoPool

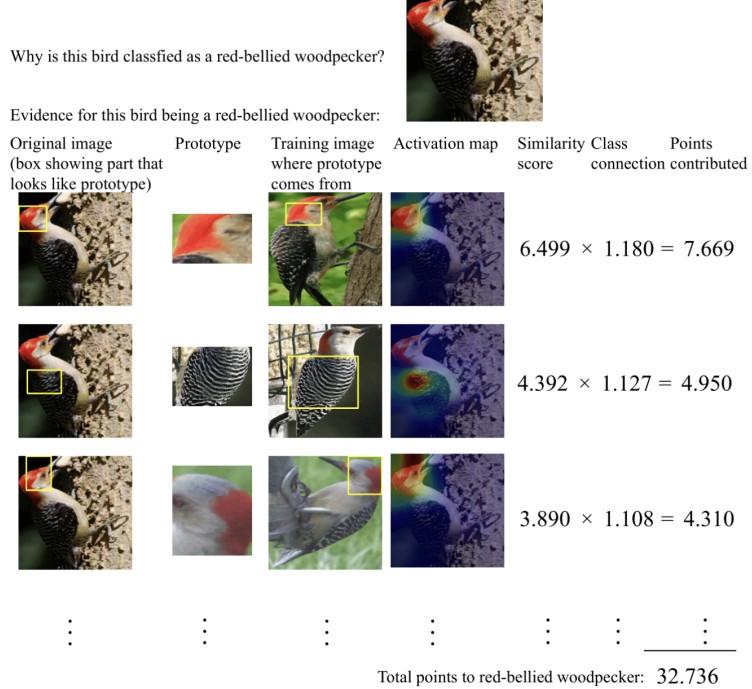

Figure 19: Local interpretation visualization in ProtoPNet

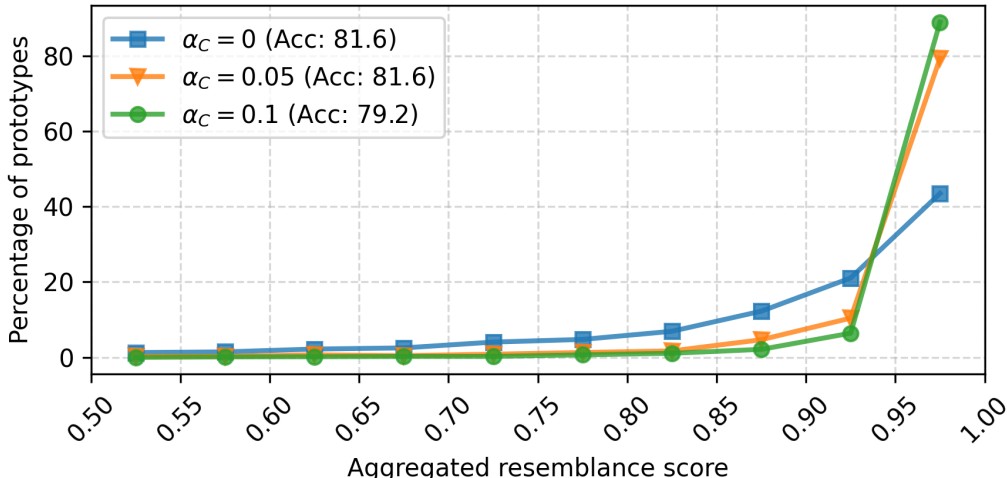

Figure 20: Distribution of the aggregated resemblance scores for different weights of cluster loss $L_C$. For LucidPPN ($\alpha_C = 0$) trained on CUB dataset, 61.04% of patches representing prototypes have aggregated resemblance score above 0.9, which indicates that prototype visualizations are relatively faithful. Moreover, when training with additional loss function $L_C$, we obtain 95.33% patches with values above 0.9, with only a small drop in accuracy.

Table 13: Accuracy before and after pruning the prototypes with less faithful visualizations. The results show that the combination of training with loss $L_C$ and pruning can enforce faithfullness of visualizations without the loss in performance.

| $\alpha_C$ | Before pruning | After pruning |
|---|---|---|
| 0 | 81.6 | 79.3 |
| 0.05 | 81.6 | 81.6 |
| 0.1 | 79.2 | 79.2 |

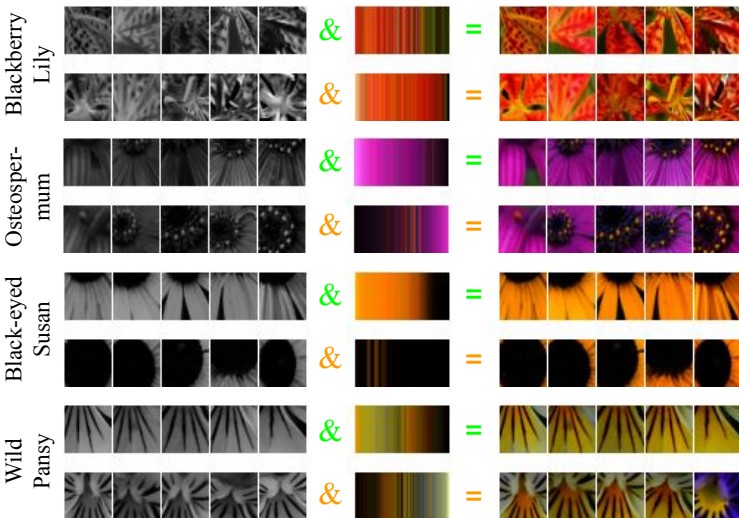

Figure 21: An example showcasing global characteristics of four classes in the FLOWER dataset, using prototypical parts from LucidPPN trained with $K = 2$. This visualization demonstrates the ability to detect differences between data classes. For instance, the *osteospermum* and *black-eyed susan* exhibit more variation in color, while the *blackberry lilly* and *wild pansay* classes differ in texture and shape.

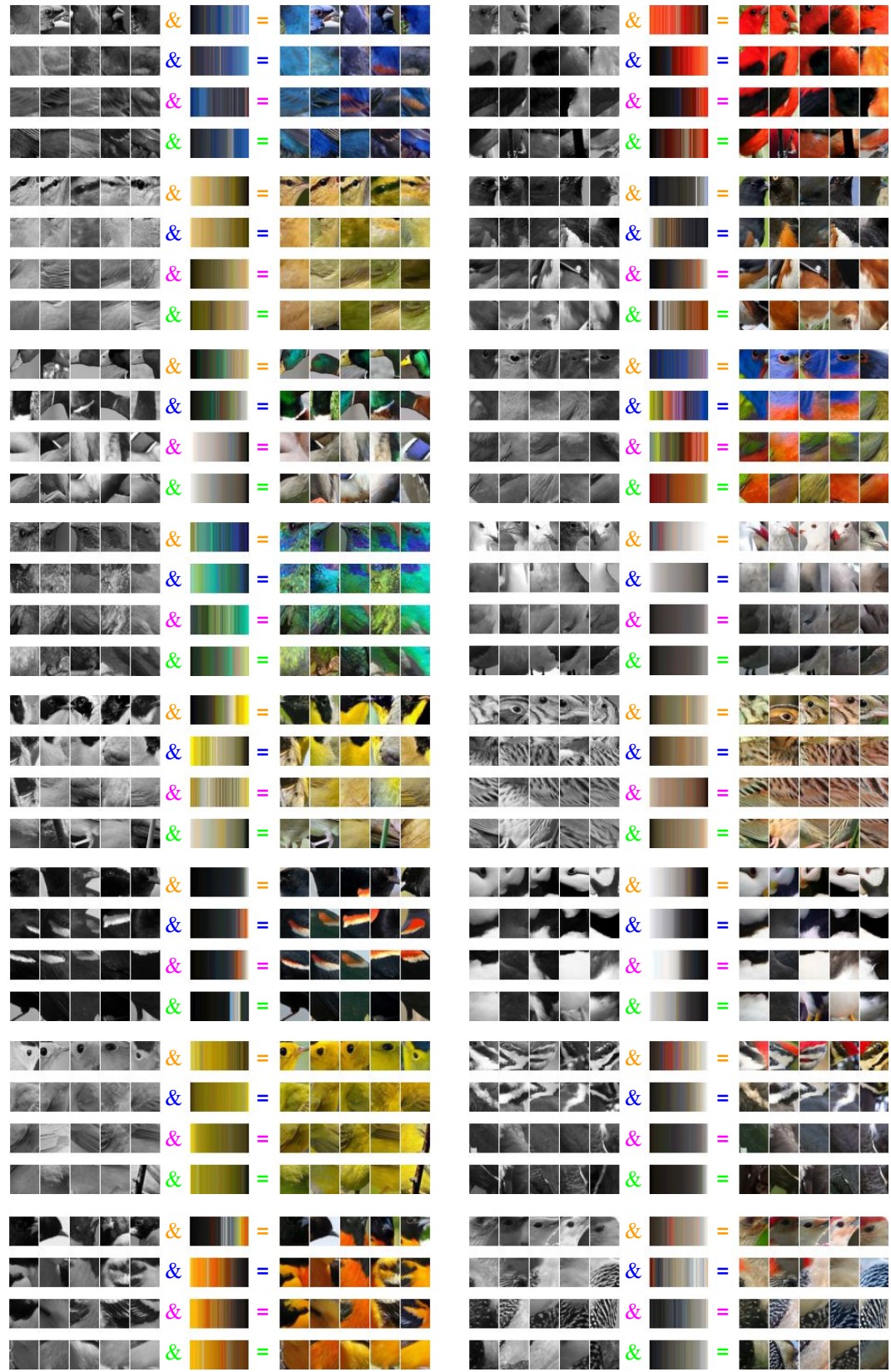

Figure 22: Selected global characteristics for LucidPPN trained on CUB with $K = 4$

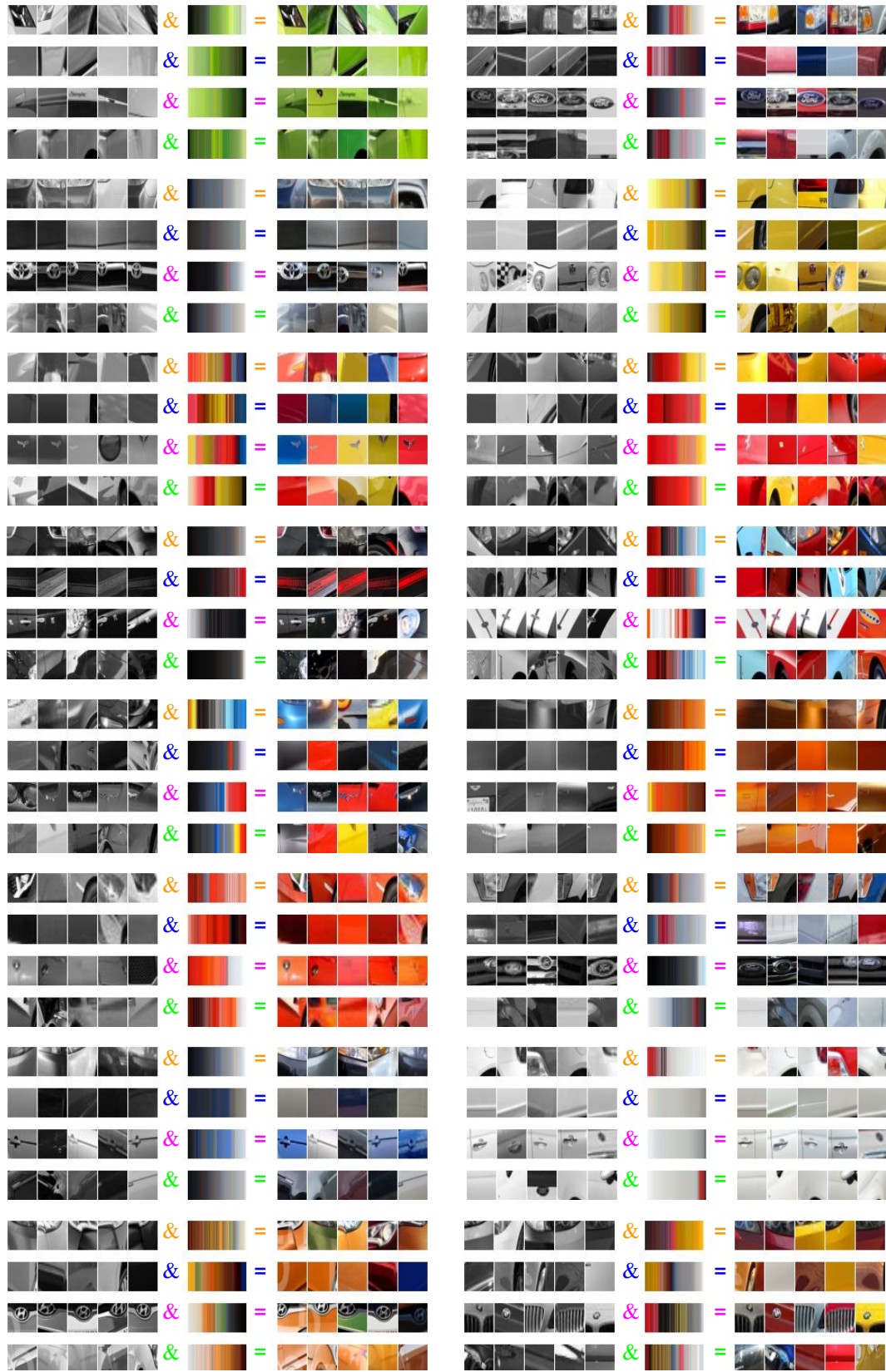

Figure 23: Selected global characteristics for LucidPPN trained on CARS with $K = 4$

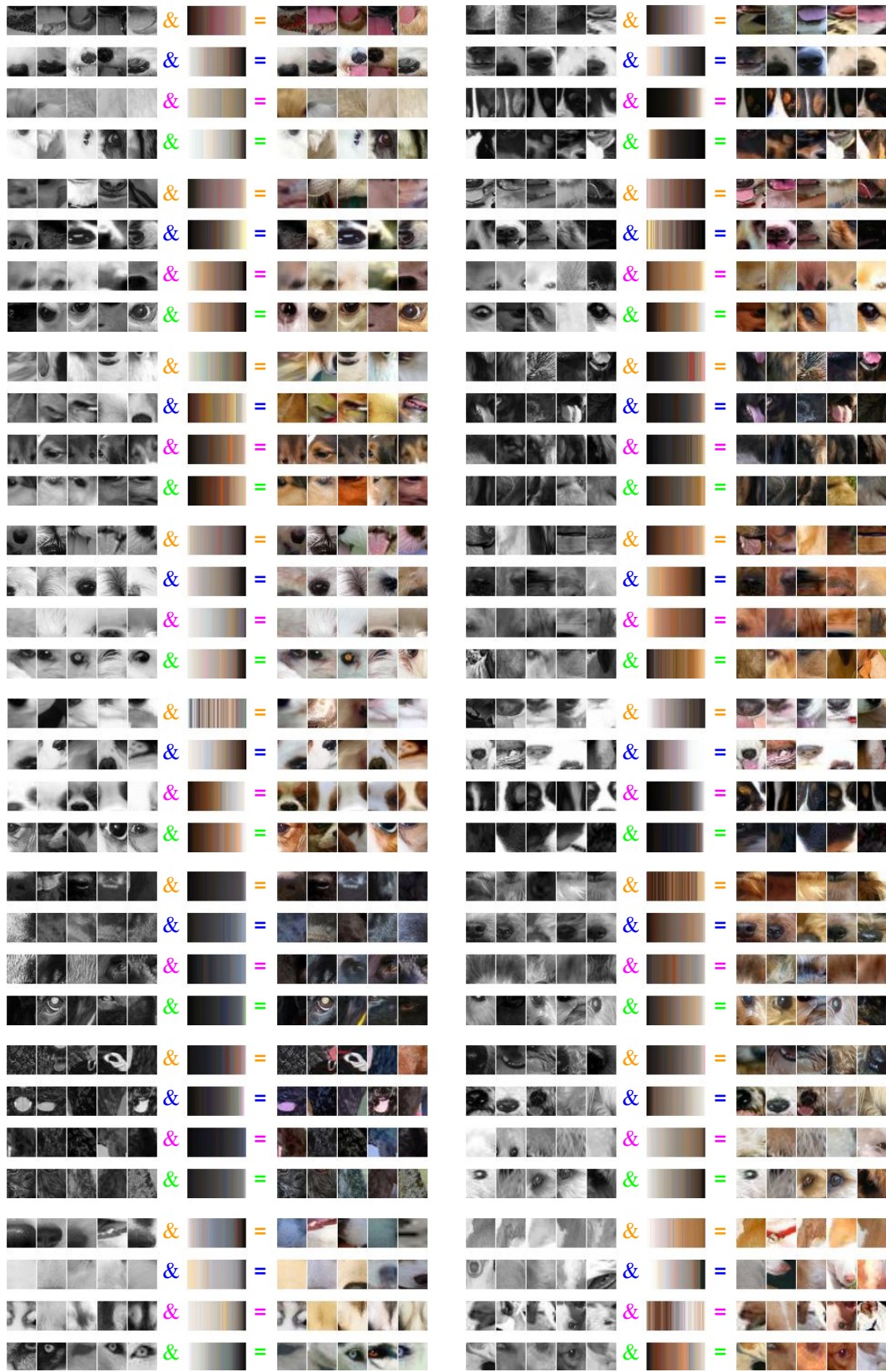

Figure 24: Selected global characteristics for LucidPPN trained on DOGS with $K = 4$

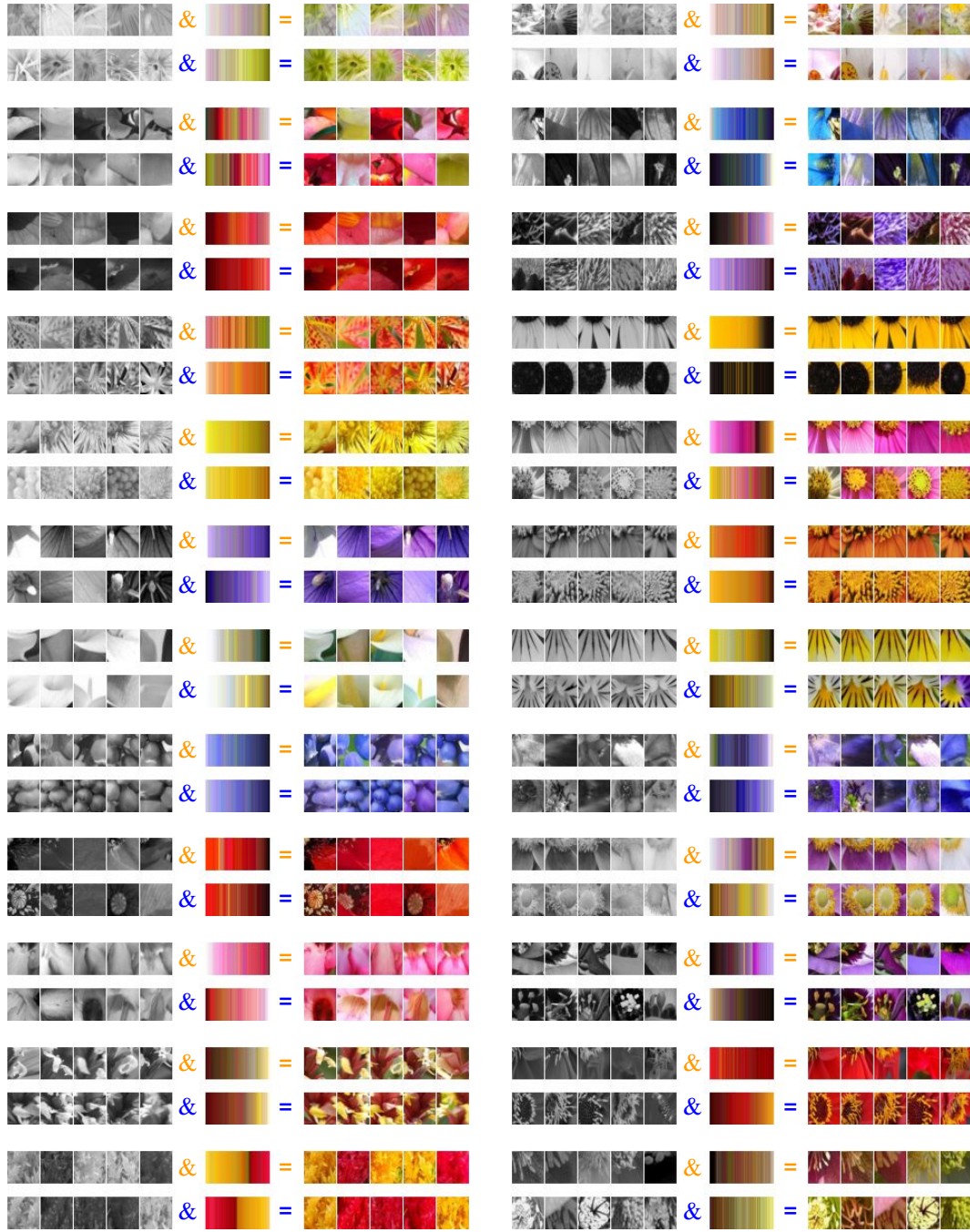

Figure 25: Selected global characteristics for LucidPPN trained on FLOWER with $K = 2$

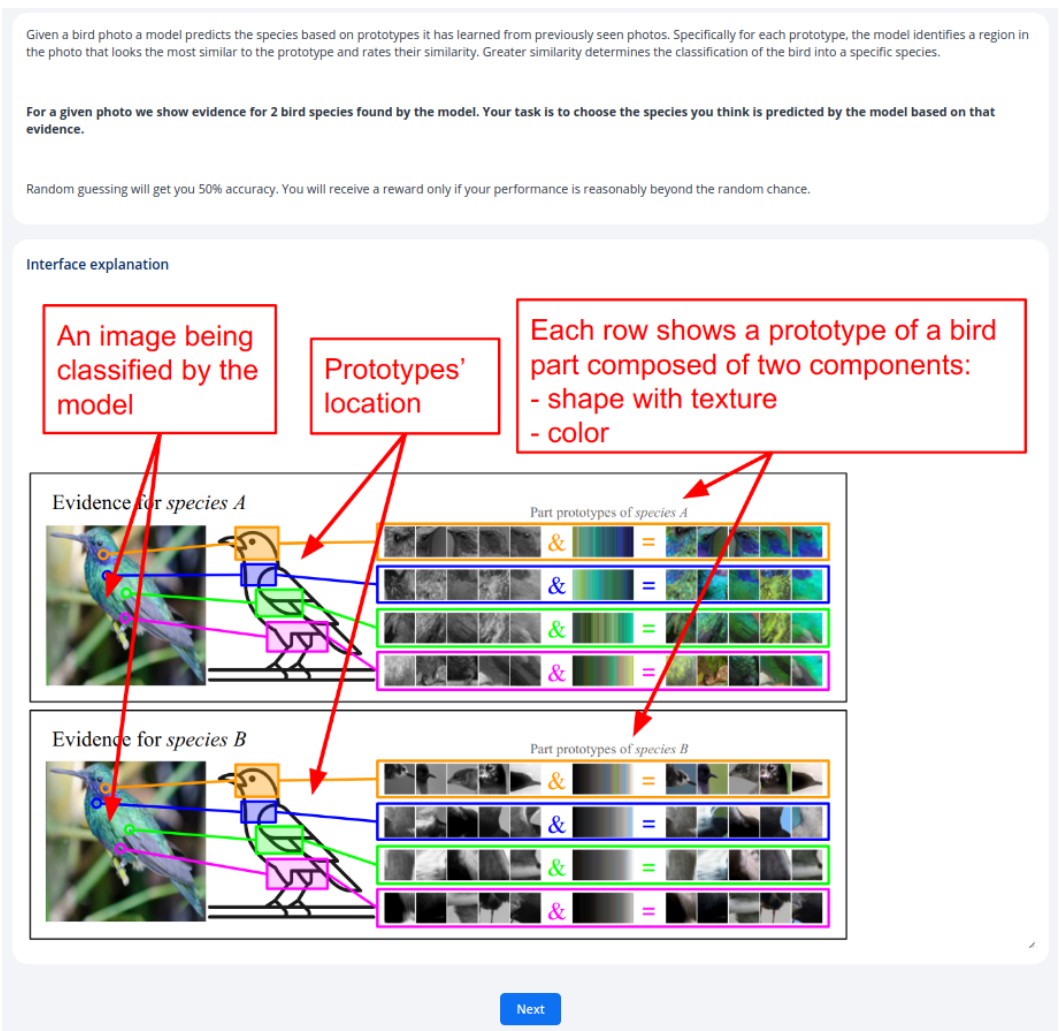

Figure 26: Page 1 of survey for LucidPPN

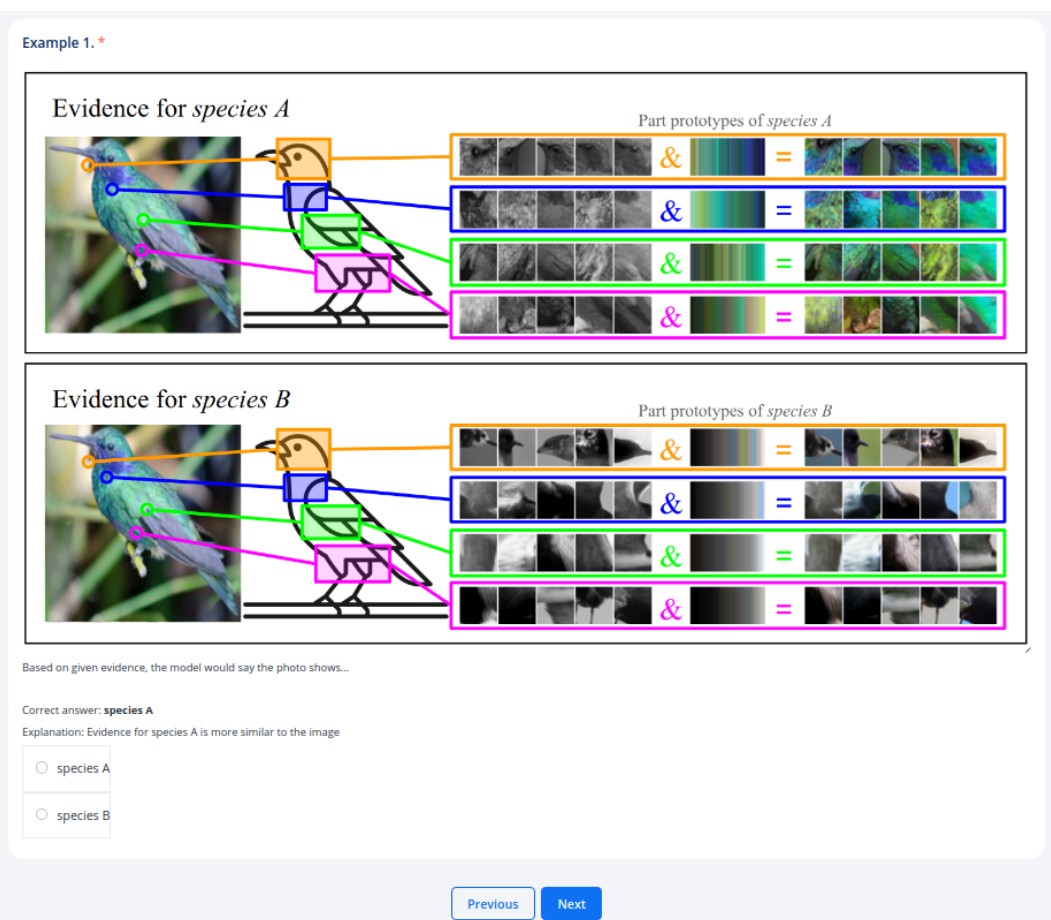

Figure 27: Page 2 of survey for LucidPPN

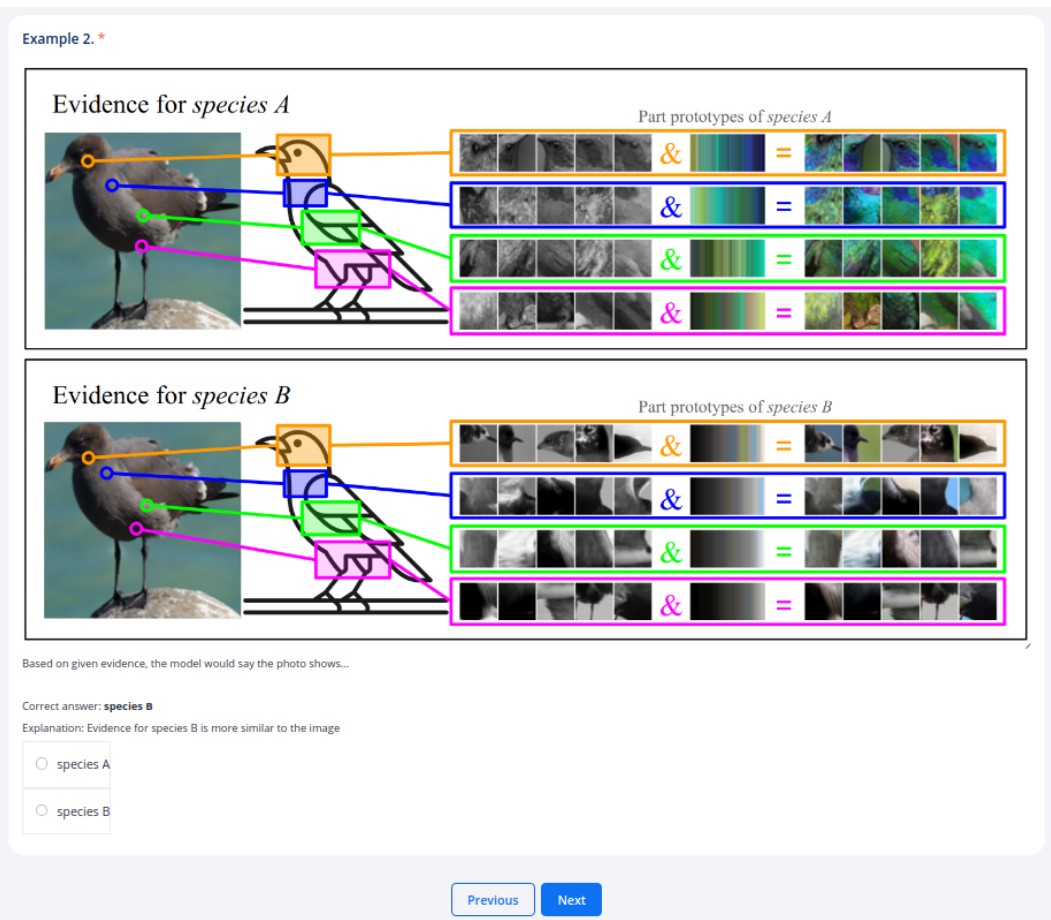

Figure 28: Page 3 of survey for LucidPPN

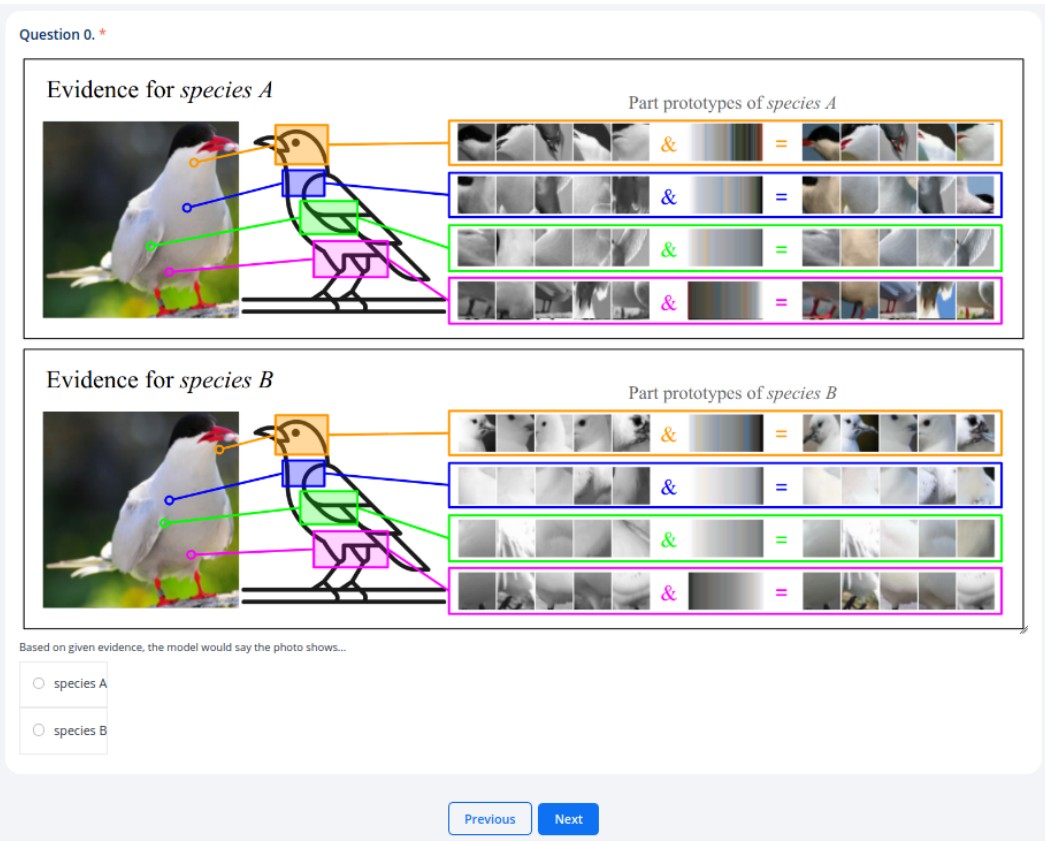

Figure 29: Page 4 of survey for LucidPPN

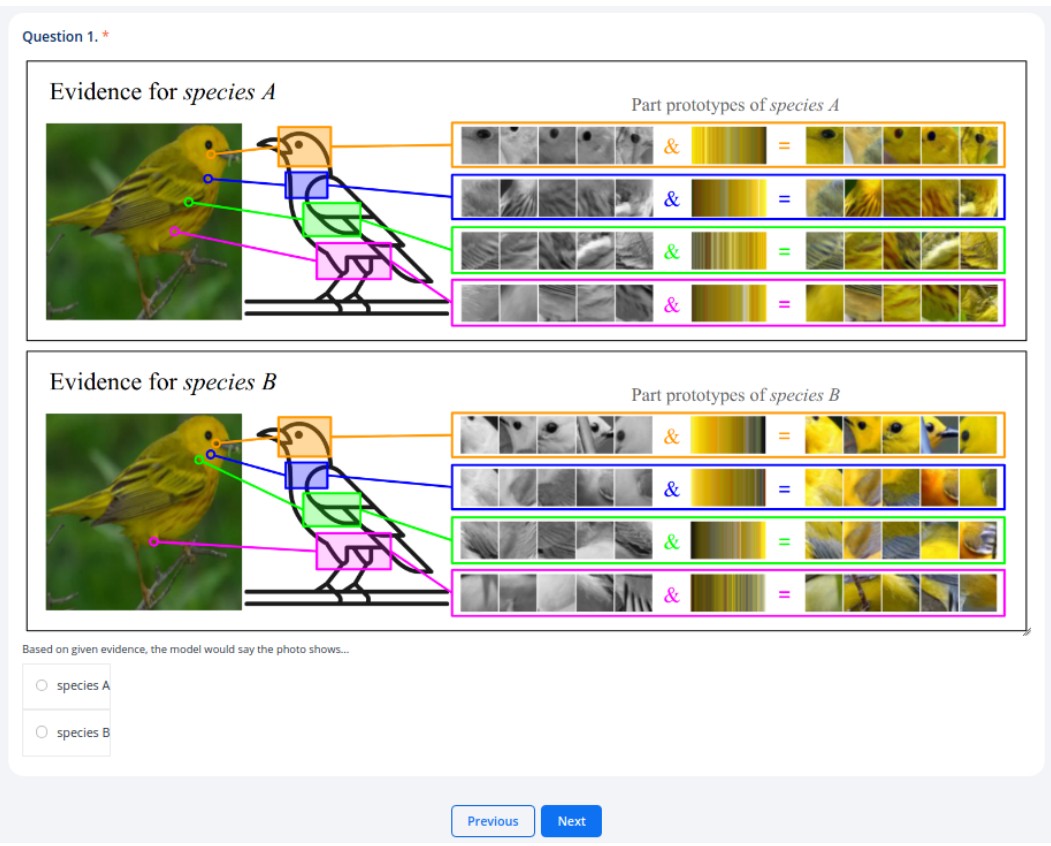

Figure 30: Page 5 of survey for LucidPPN

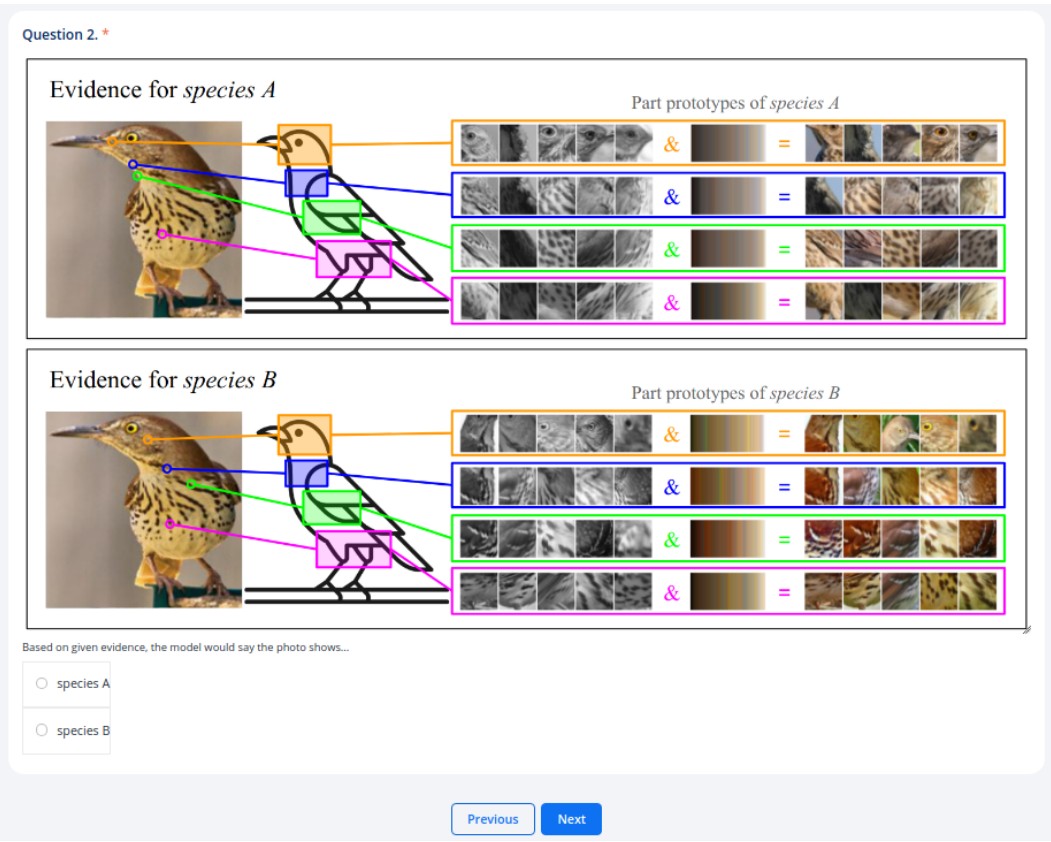

Figure 31: Page 6 of survey for LucidPPN

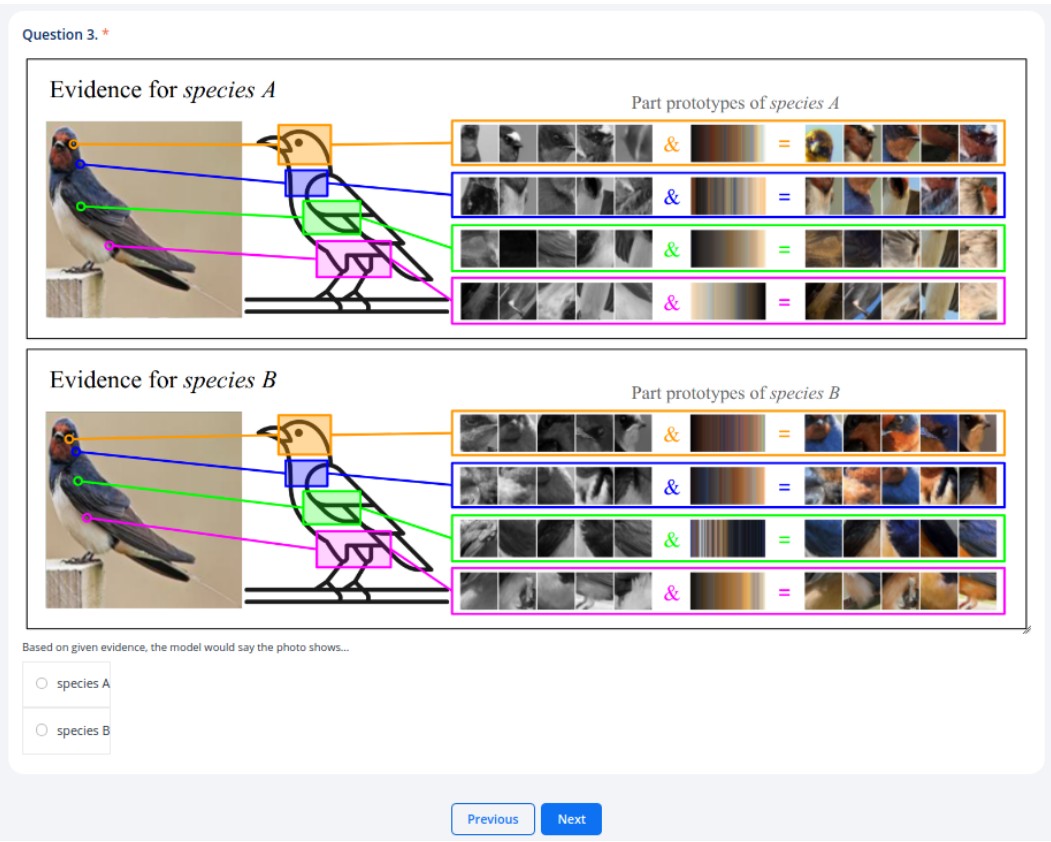

Figure 32: Page 7 of survey for LucidPPN

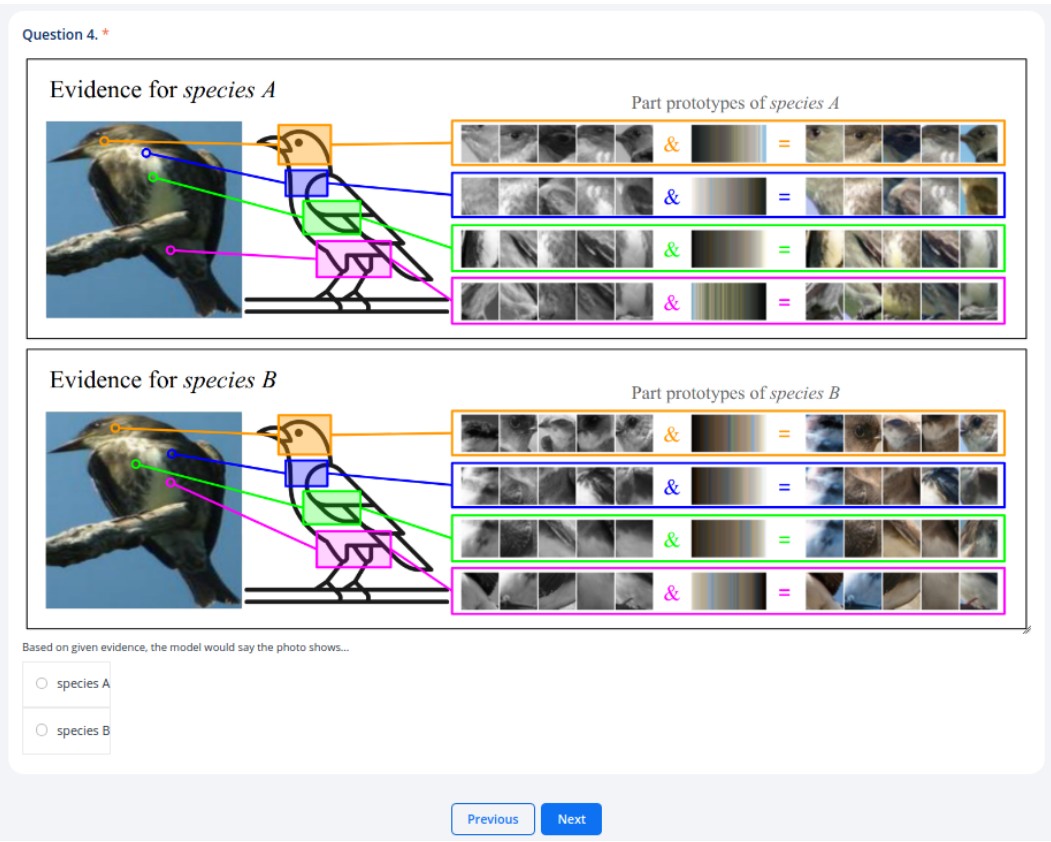

Figure 33: Page 8 of survey for LucidPPN

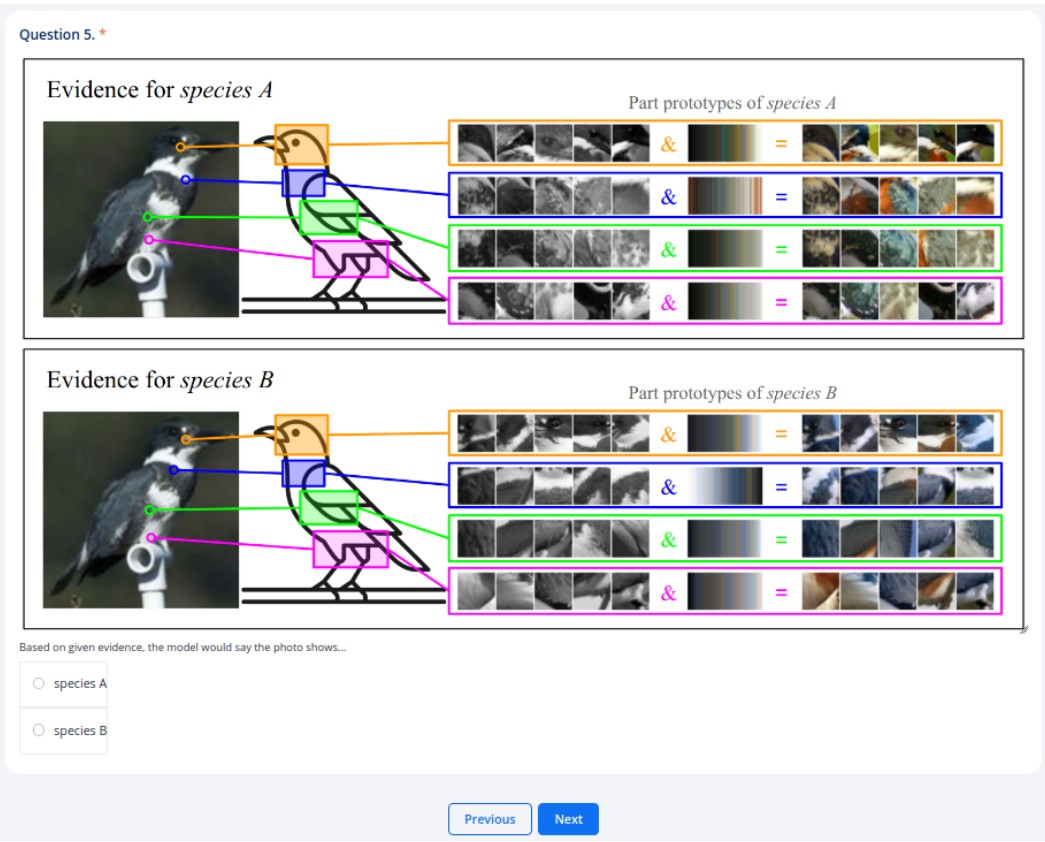

Figure 34: Page 9 of survey for LucidPPN

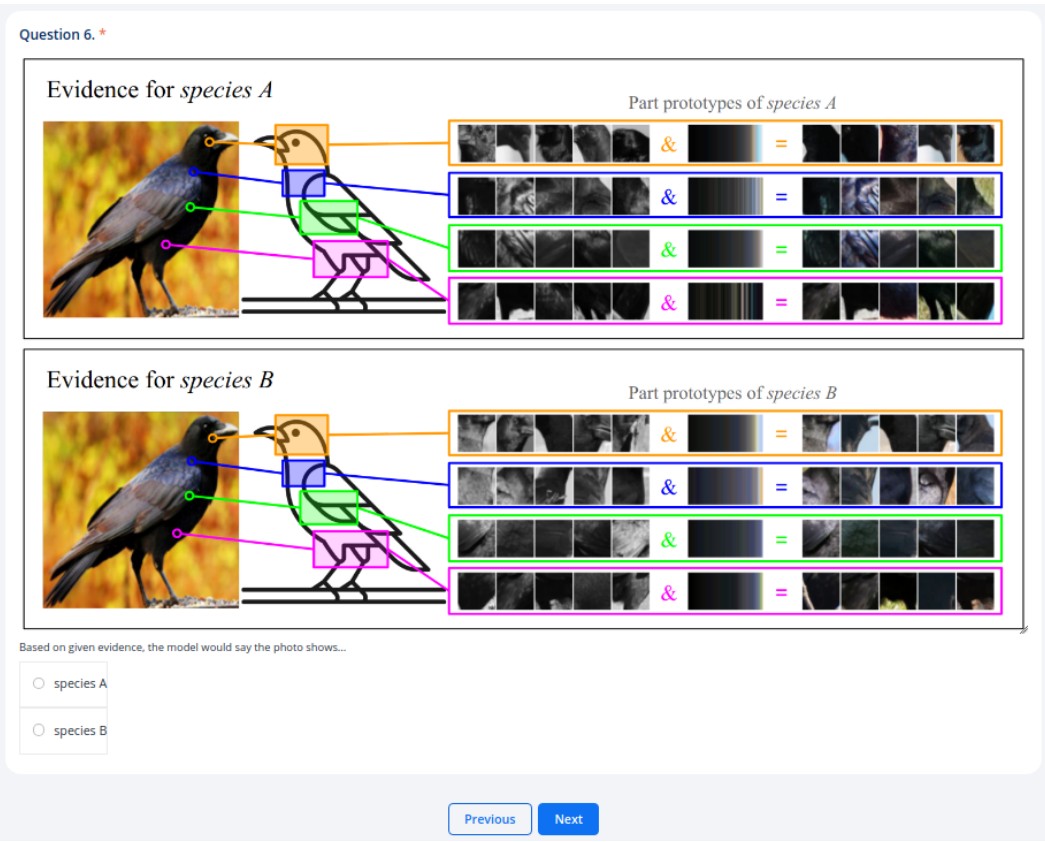

Figure 35: Page 10 of survey for LucidPPN

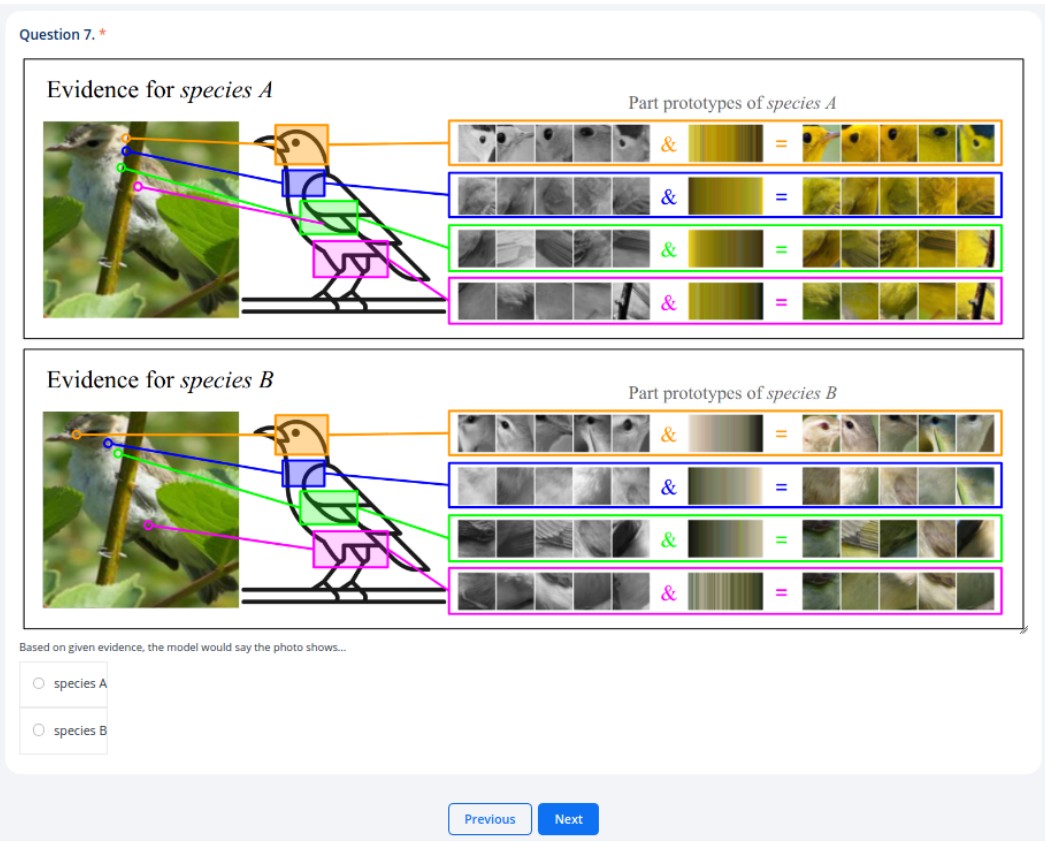

Figure 36: Page 11 of survey for LucidPPN

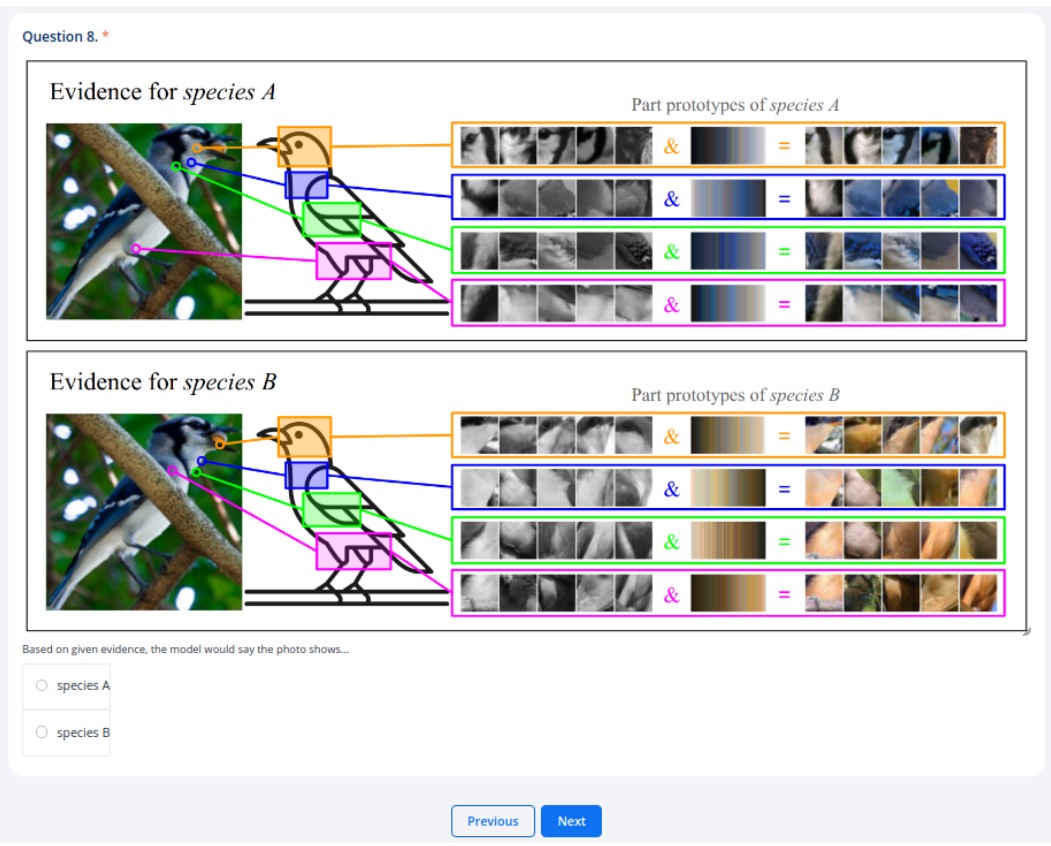

Figure 37: Page 12 of survey for LucidPPN

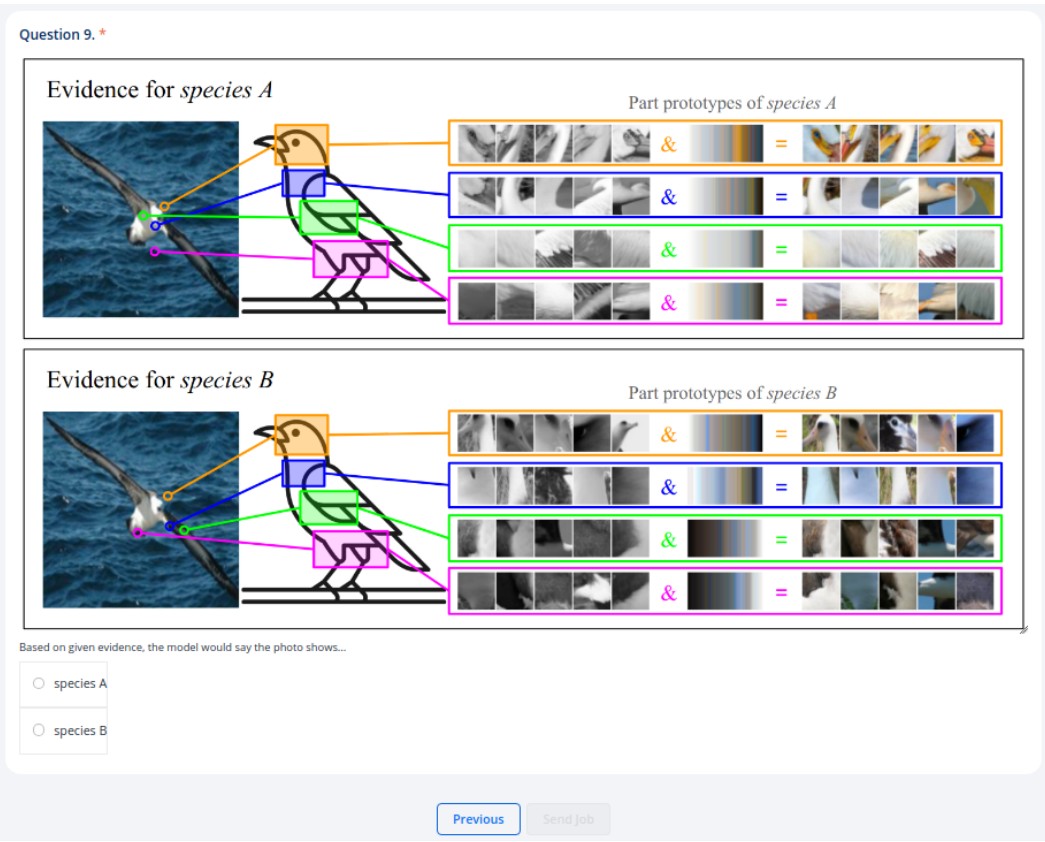

Figure 38: Page 13 of survey for LucidPPN

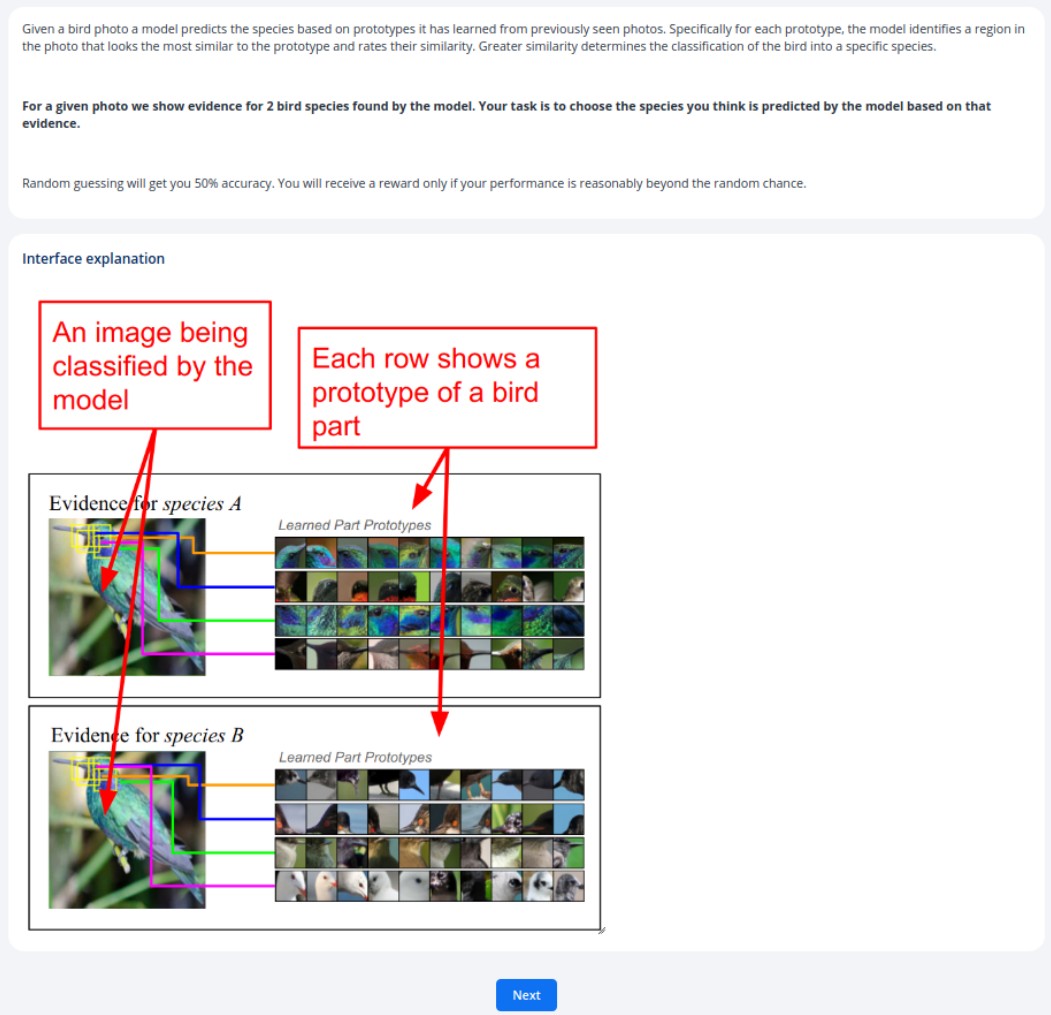

Figure 39: Page 1 of survey for PIP-Net

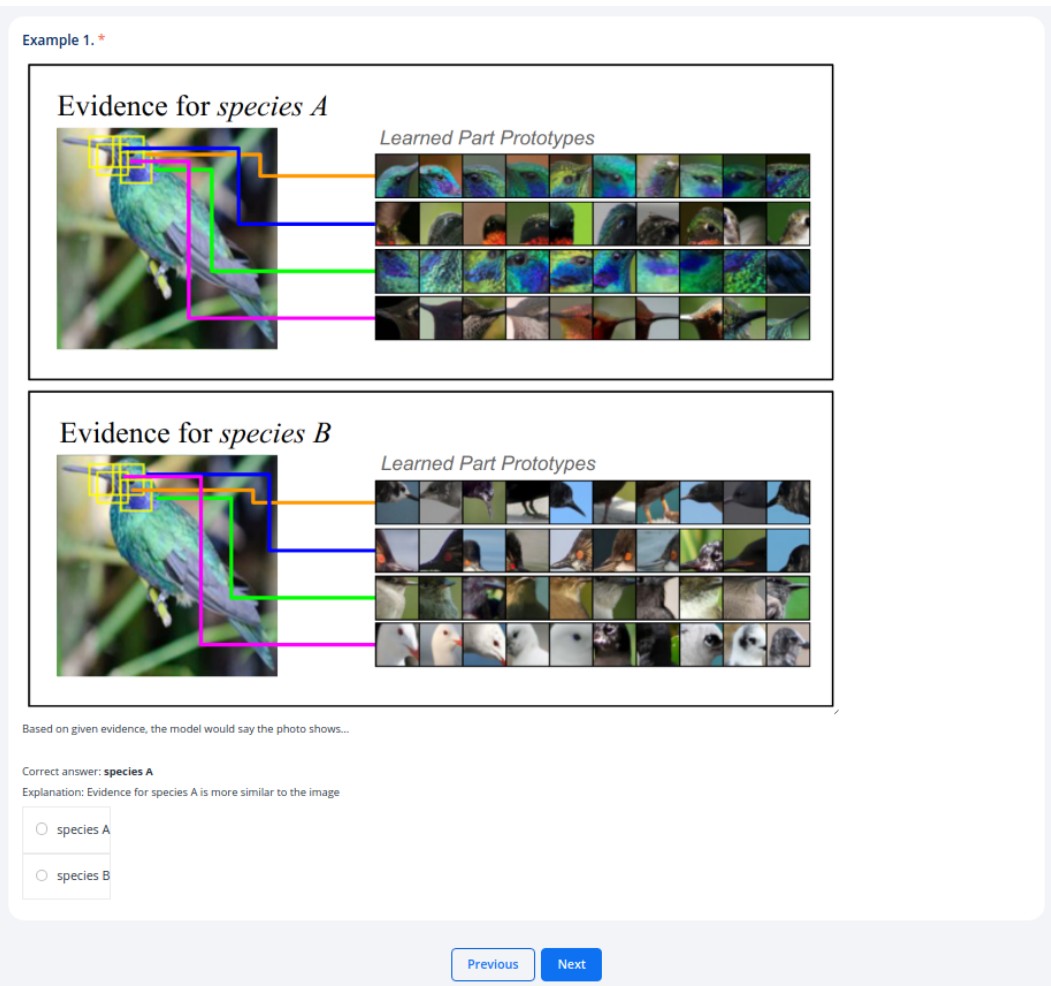

Figure 40: Page 2 of survey for PIP-Net

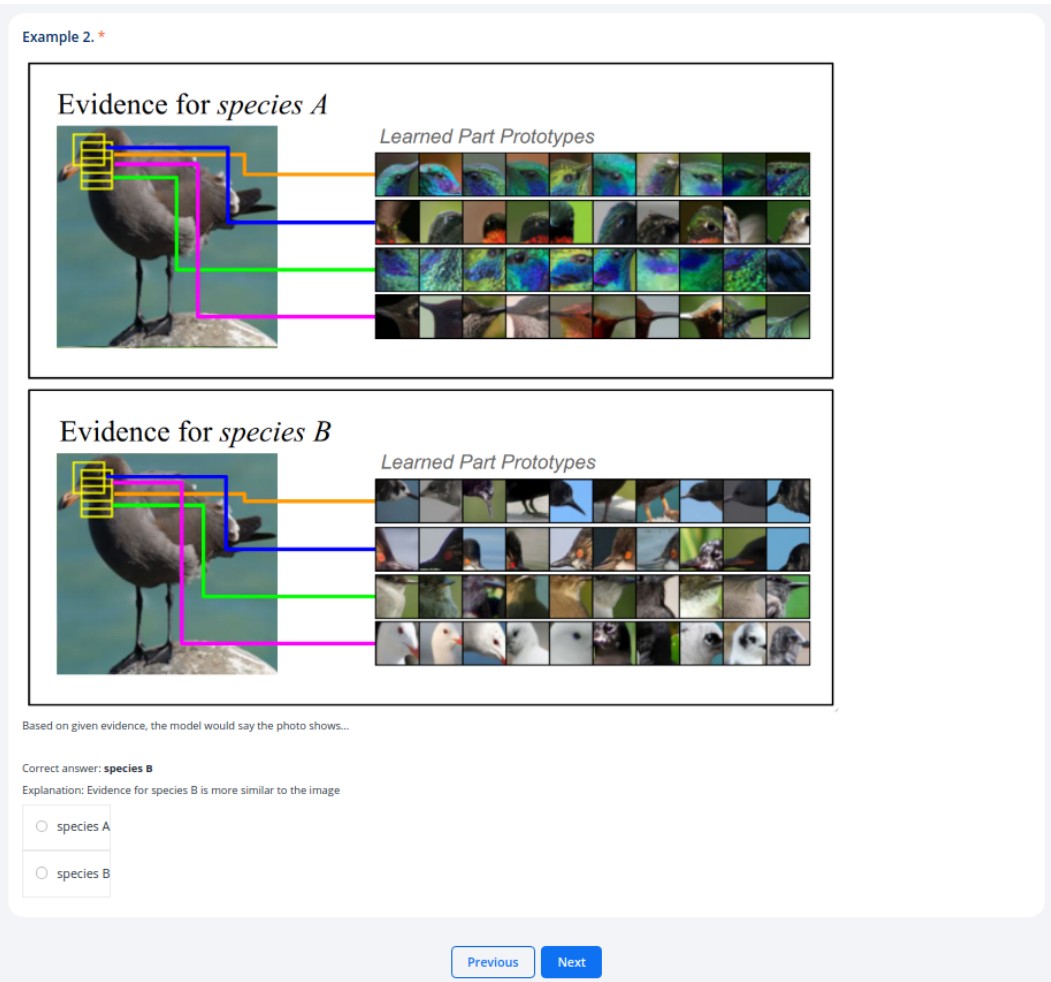

Figure 41: Page 3 of survey for PIP-Net

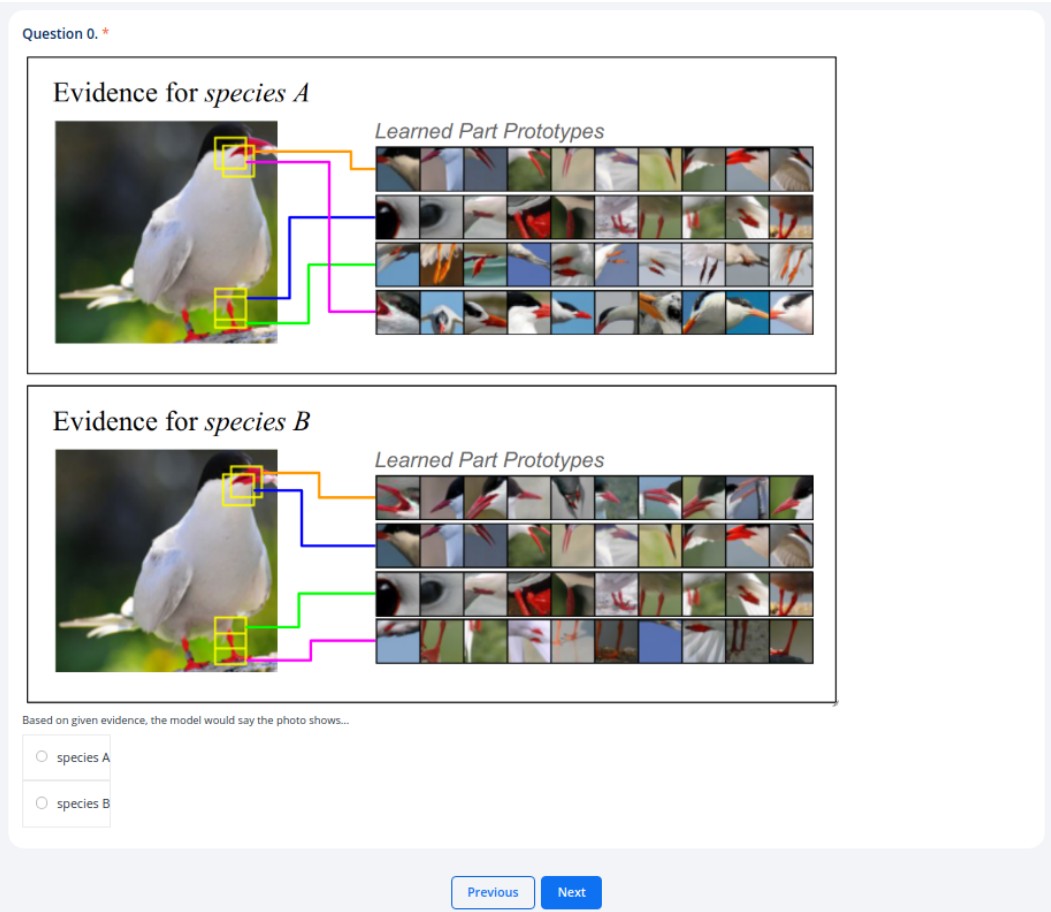

Figure 42: Page 4 of survey for PIP-Net

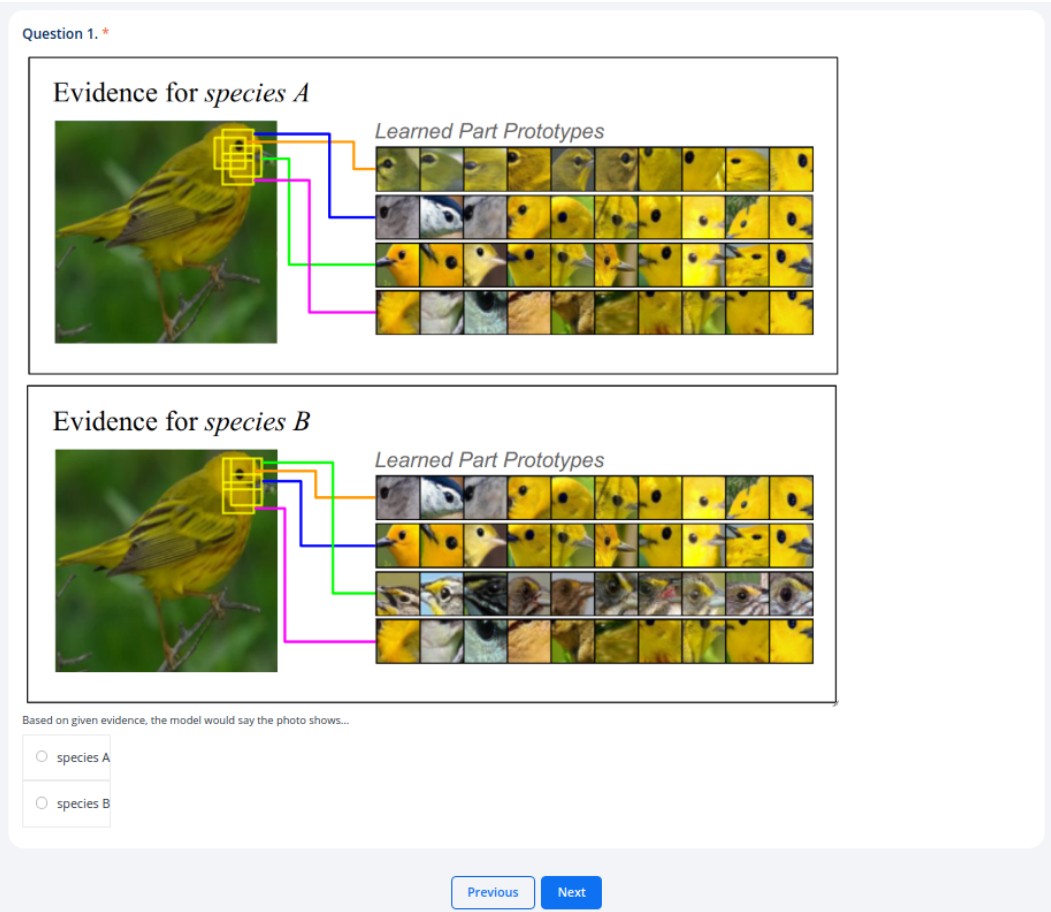

Figure 43: Page 5 of survey for PIP-Net

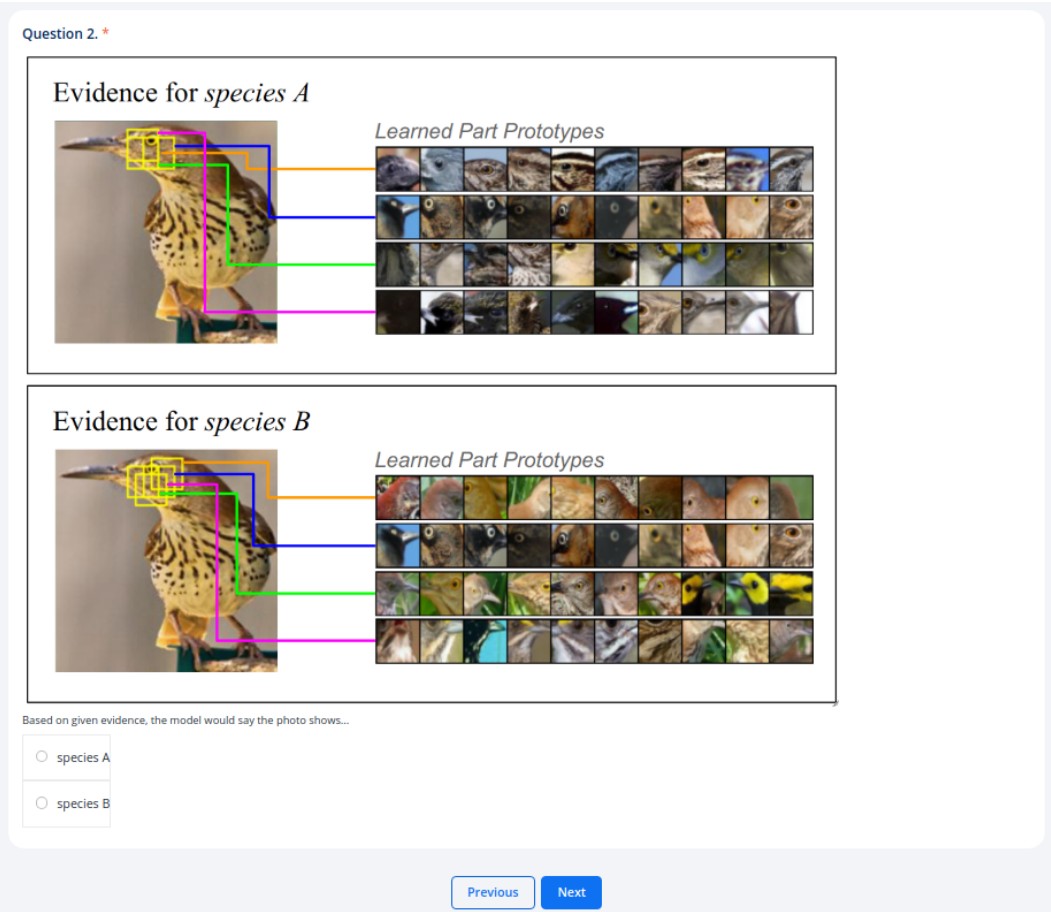

Figure 44: Page 6 of survey for PIP-Net

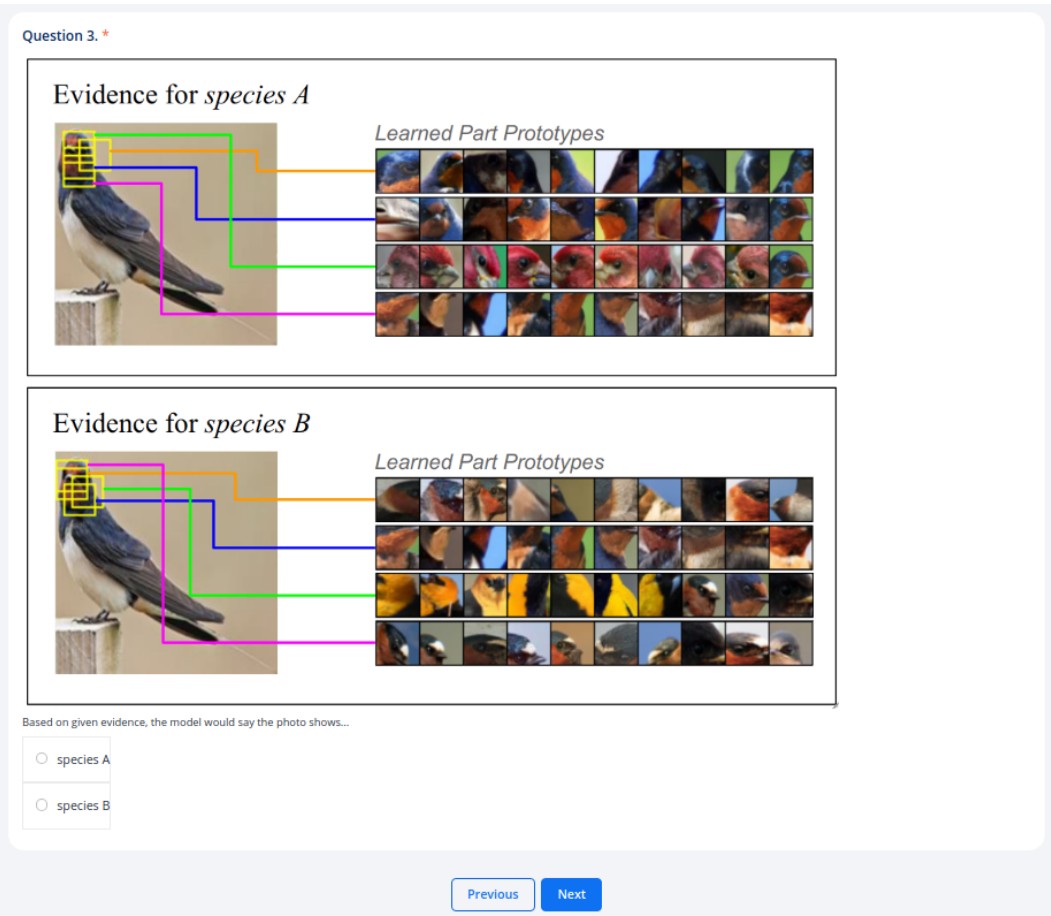

Figure 45: Page 7 of survey for PIP-Net

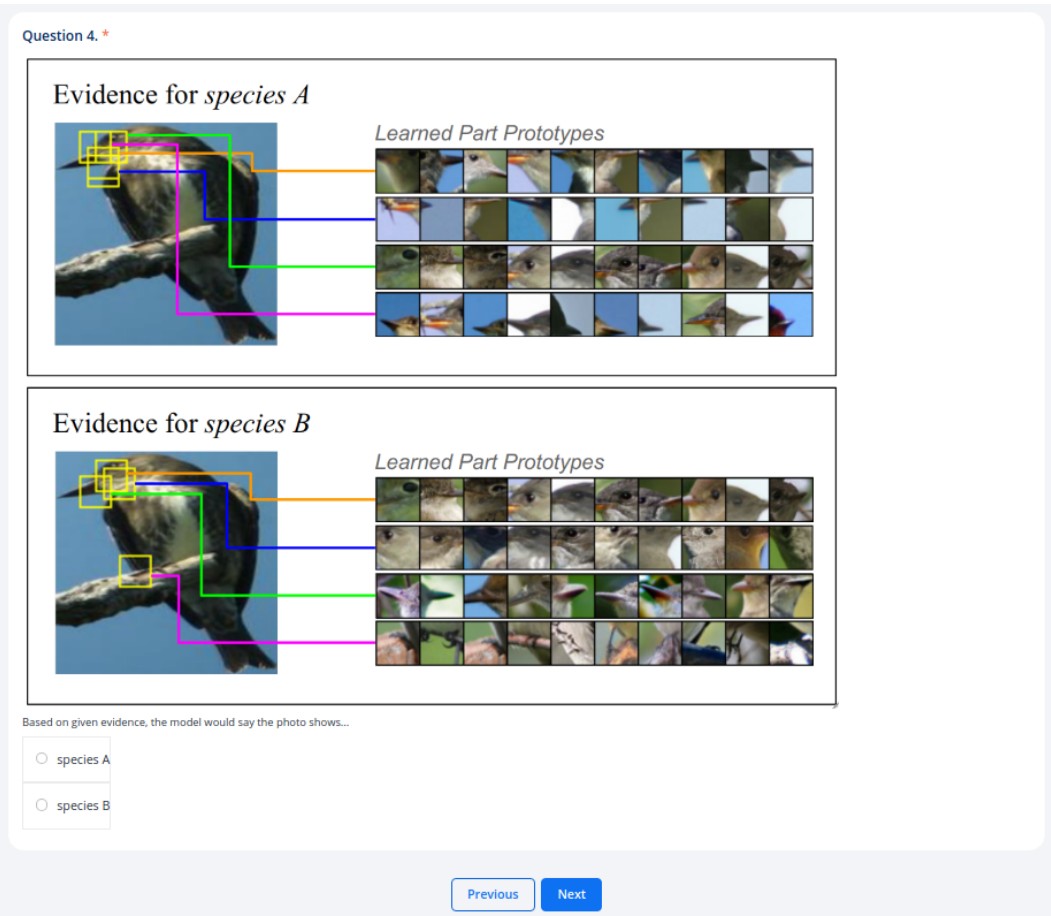

Figure 46: Page 8 of survey for PIP-Net

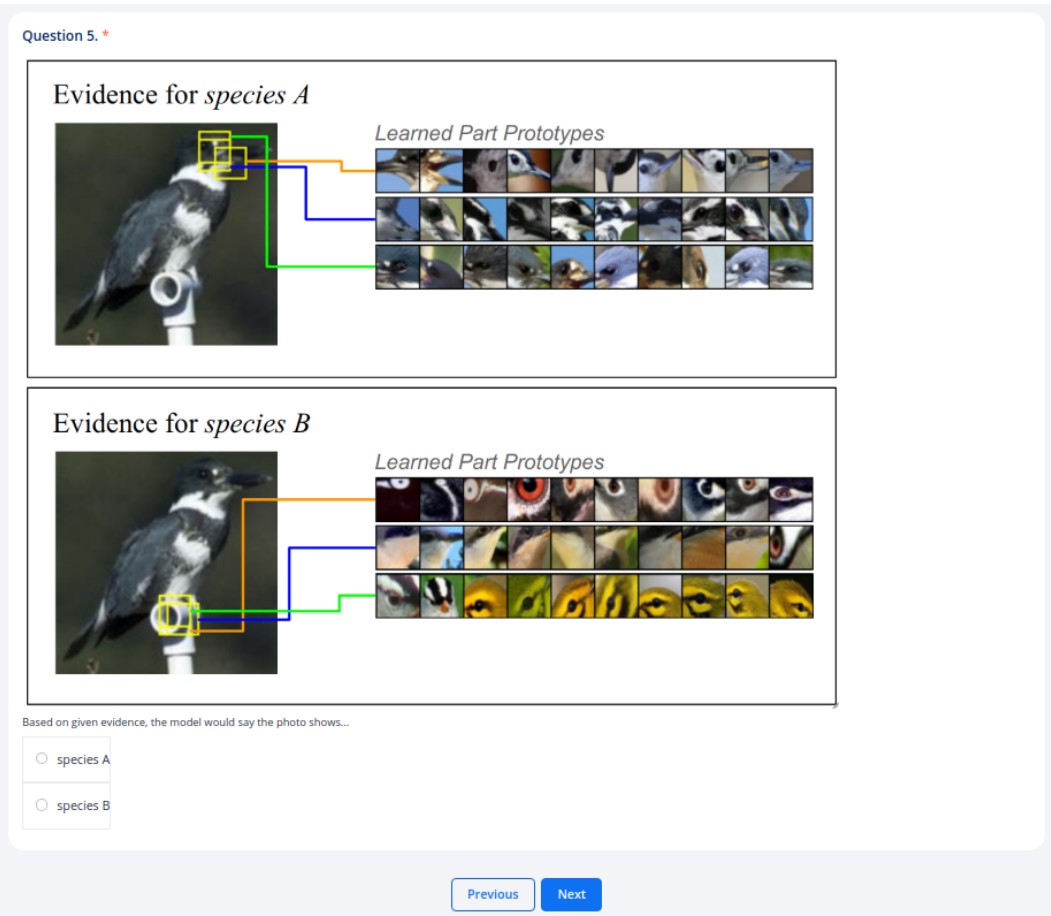

Figure 47: Page 9 of survey for PIP-Net

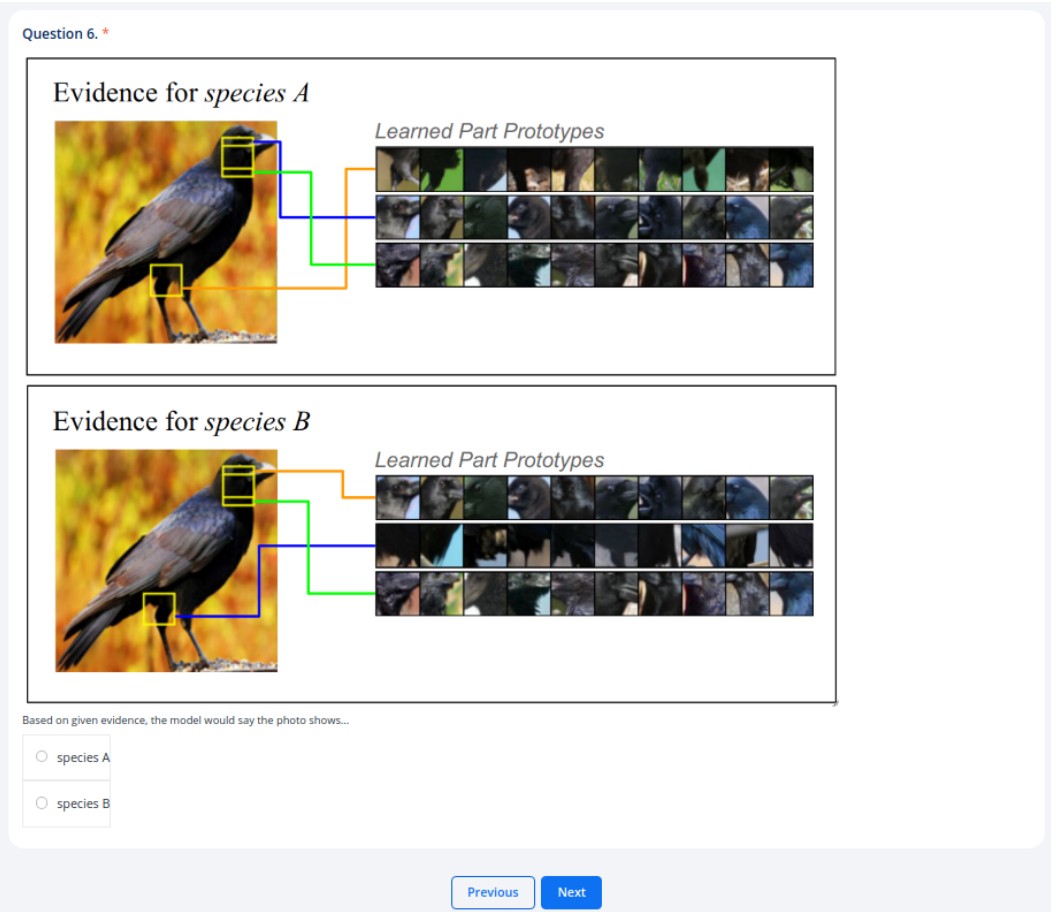

Figure 48: Page 10 of survey for PIP-Net

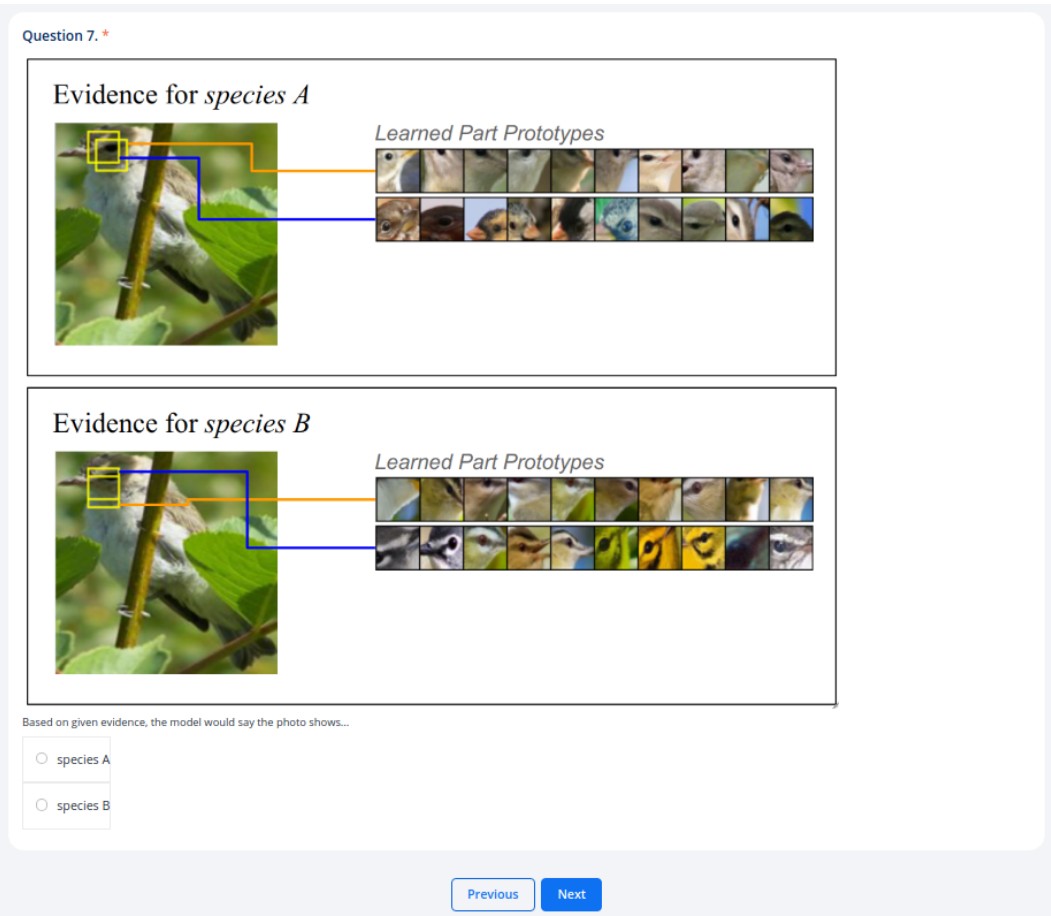

Figure 49: Page 11 of survey for PIP-Net

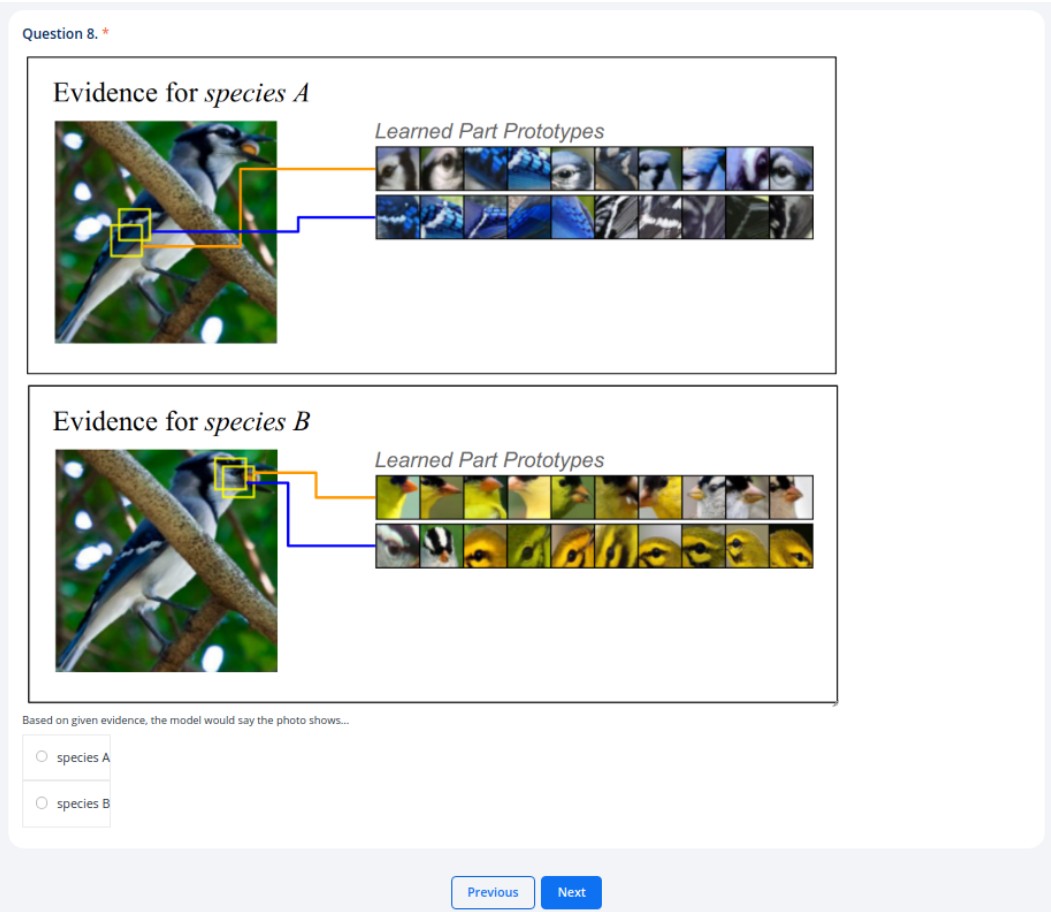

Figure 50: Page 12 of survey for PIP-Net

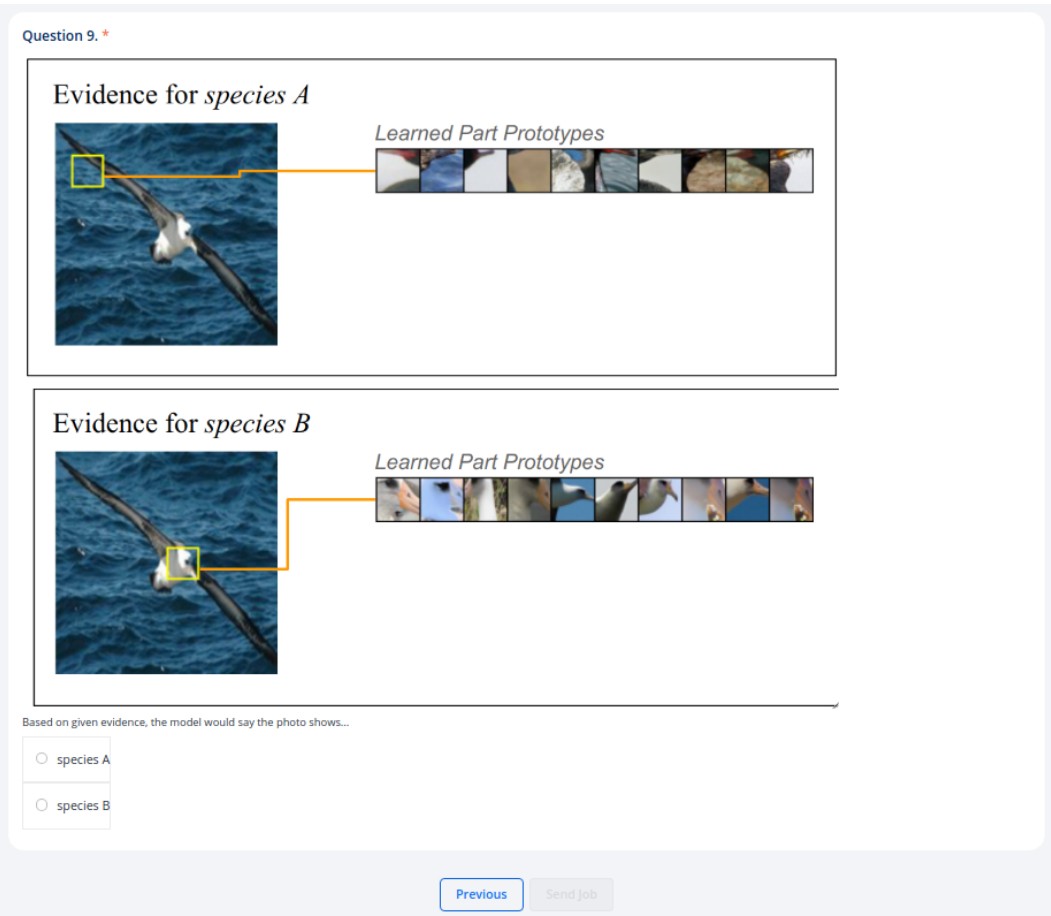

Figure 51: Page 13 of survey for PIP-Net

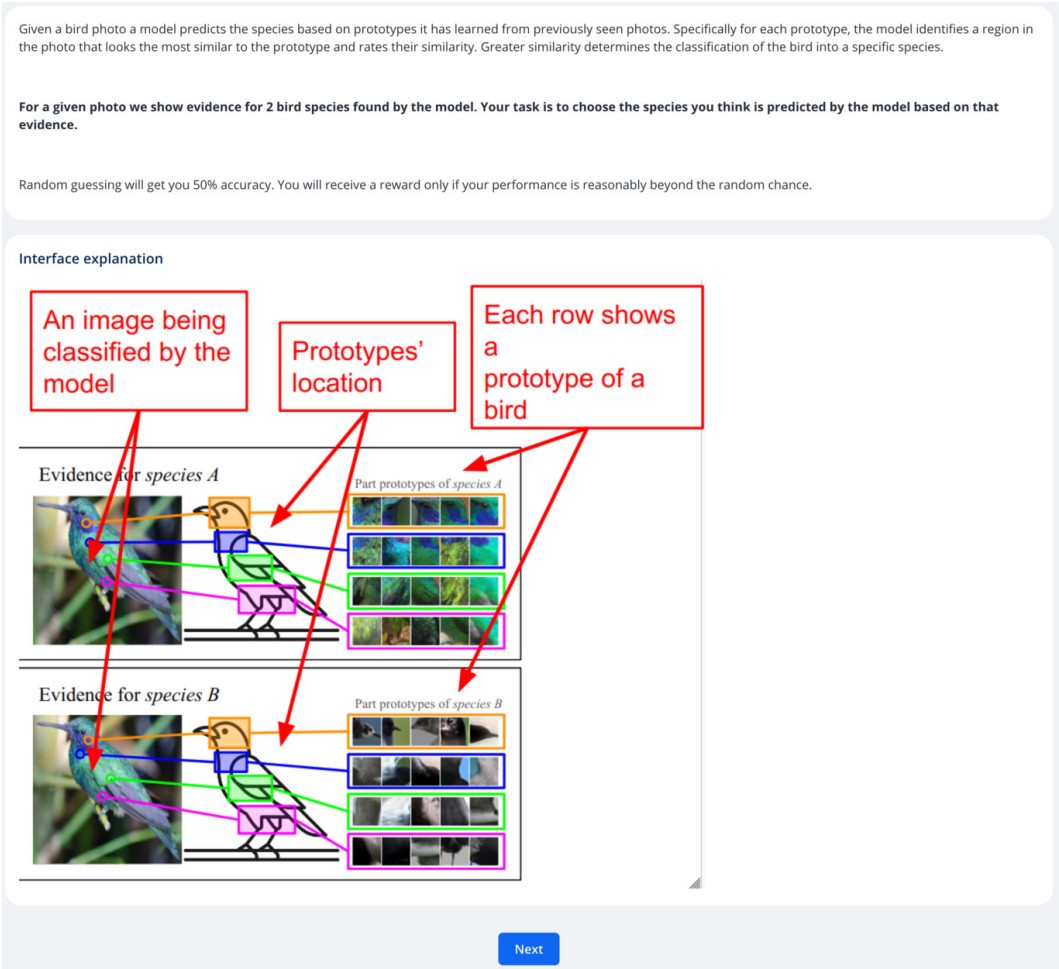

Given a bird photo a model predicts the species based on prototypes it has learned from previously seen photos. Specifically for each prototype, the model identifies a region in the photo that looks the most similar to the prototype and rates their similarity. Greater similarity determines the classification of the bird into a specific species.

**For a given photo we show evidence for 2 bird species found by the model. Your task is to choose the species you think is predicted by the model based on that evidence.**

Random guessing will get you 50% accuracy. You will receive a reward only if your performance is reasonably beyond the random chance.

Figure 52: Page 1 of survey for *single branch*

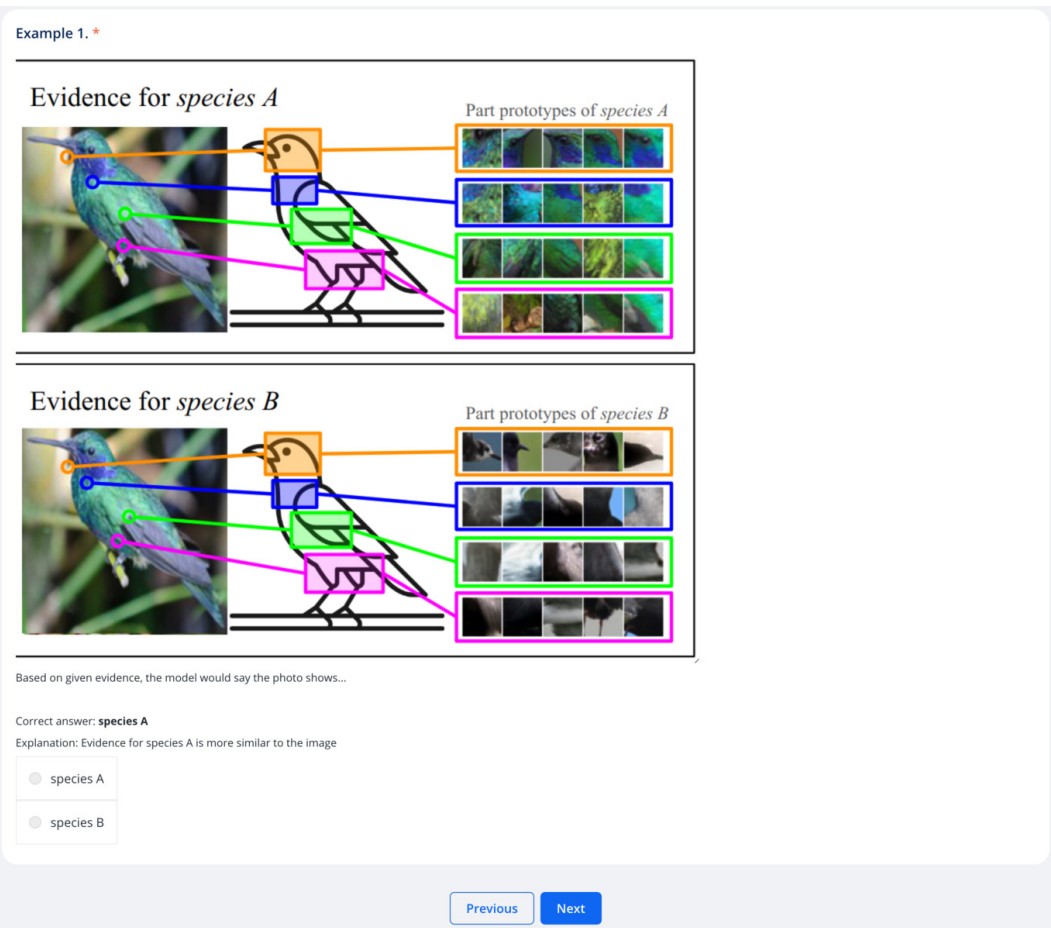

Figure 53: Page 2 of survey for *single branch*

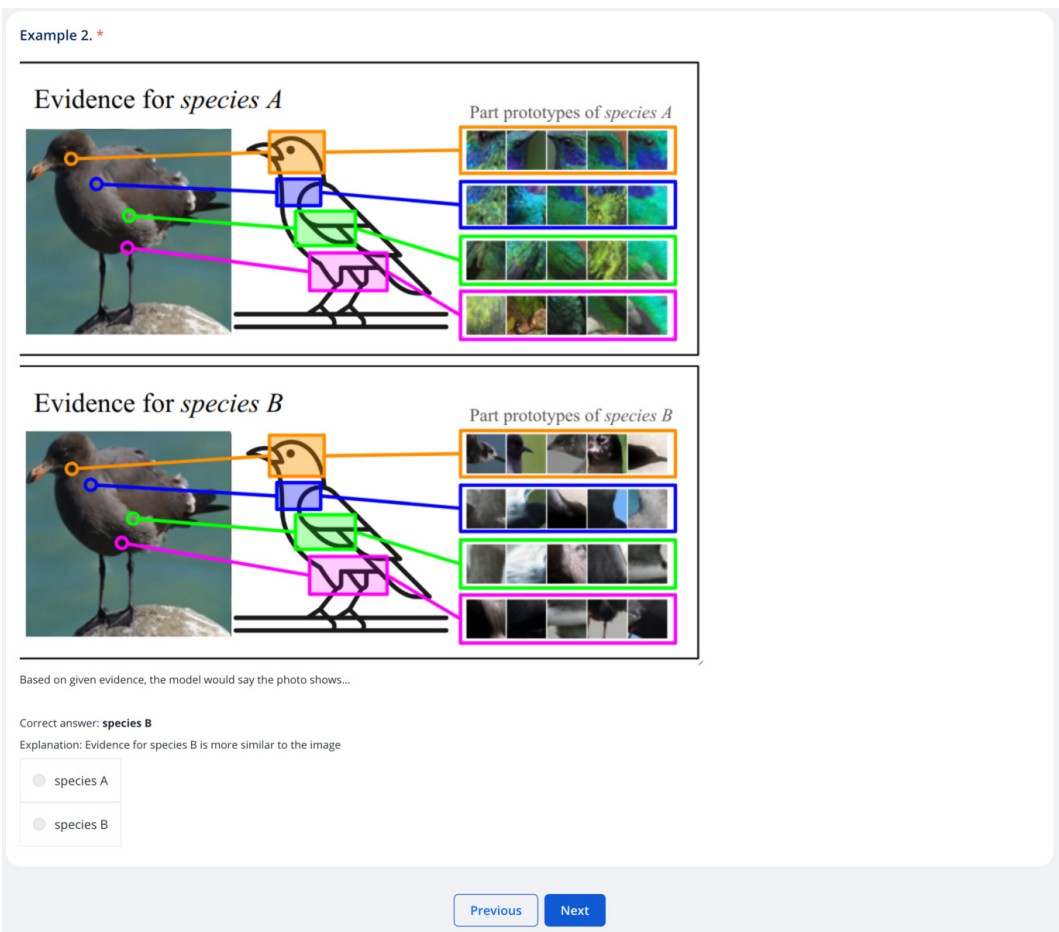

Figure 54: Page 3 of survey for *single branch*

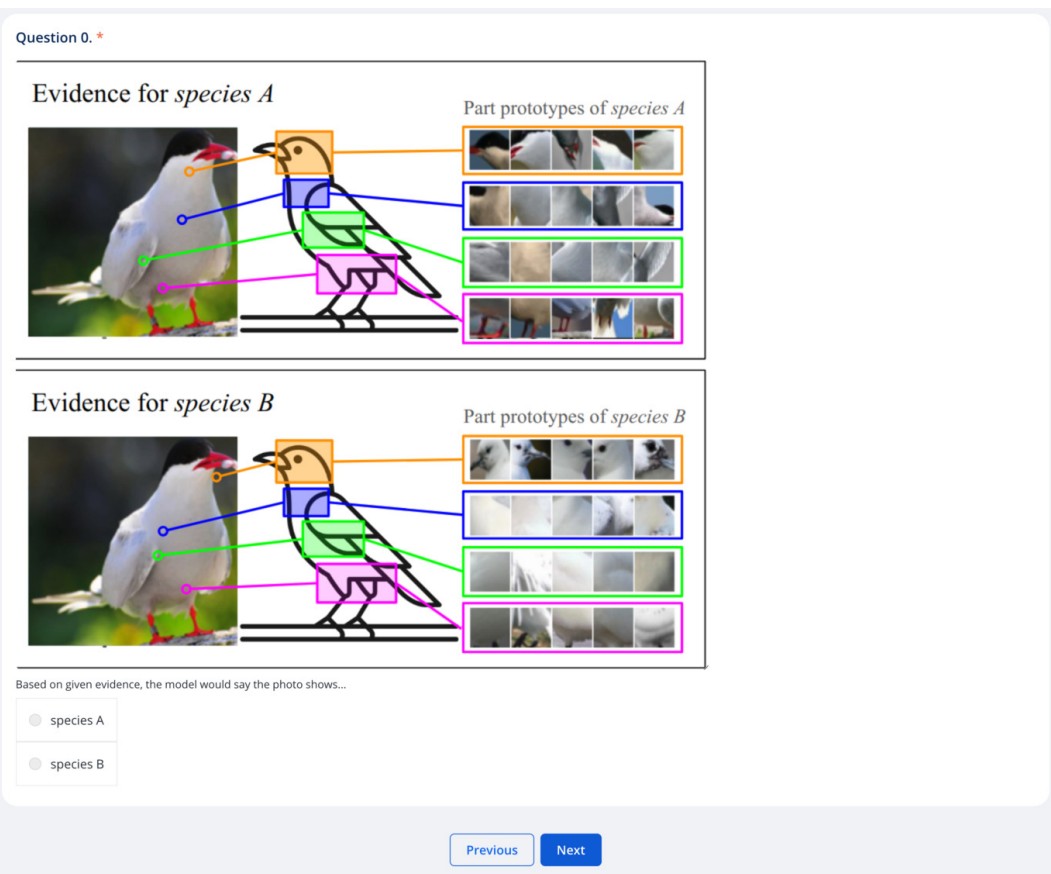

Figure 55: Page 4 of survey for *single branch*

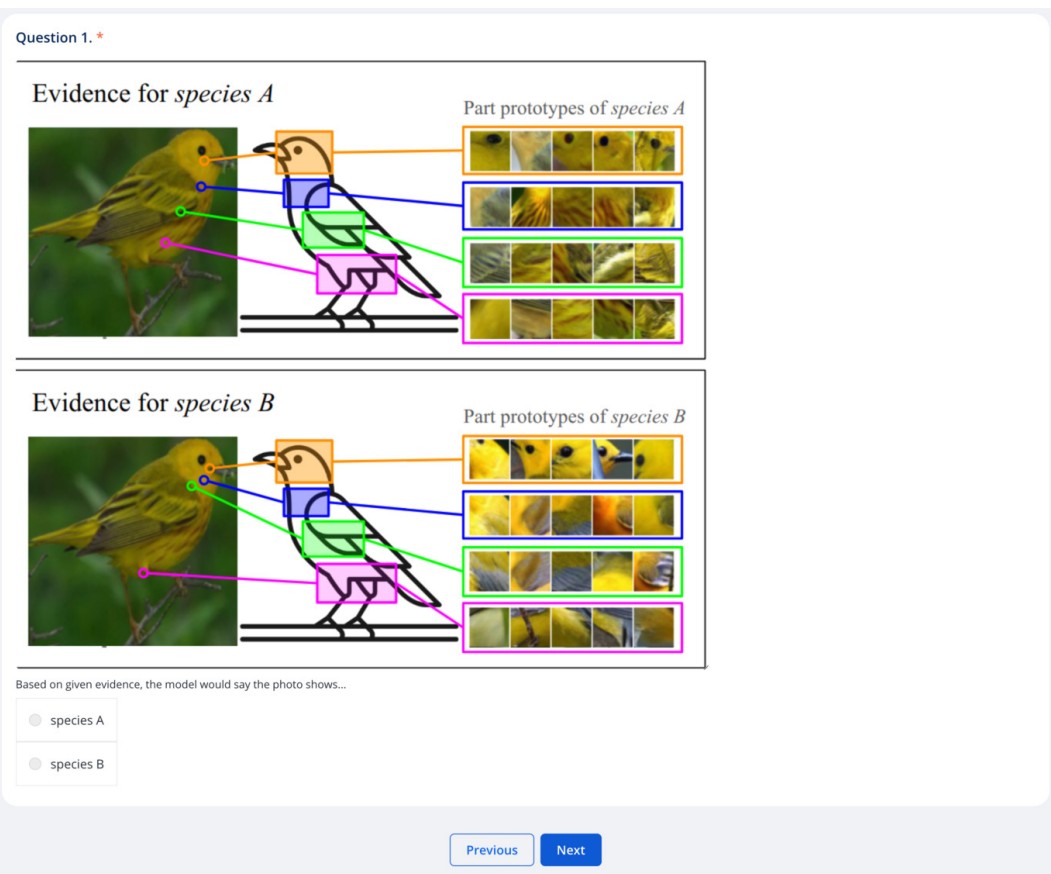

Figure 56: Page 5 of survey for *single branch*

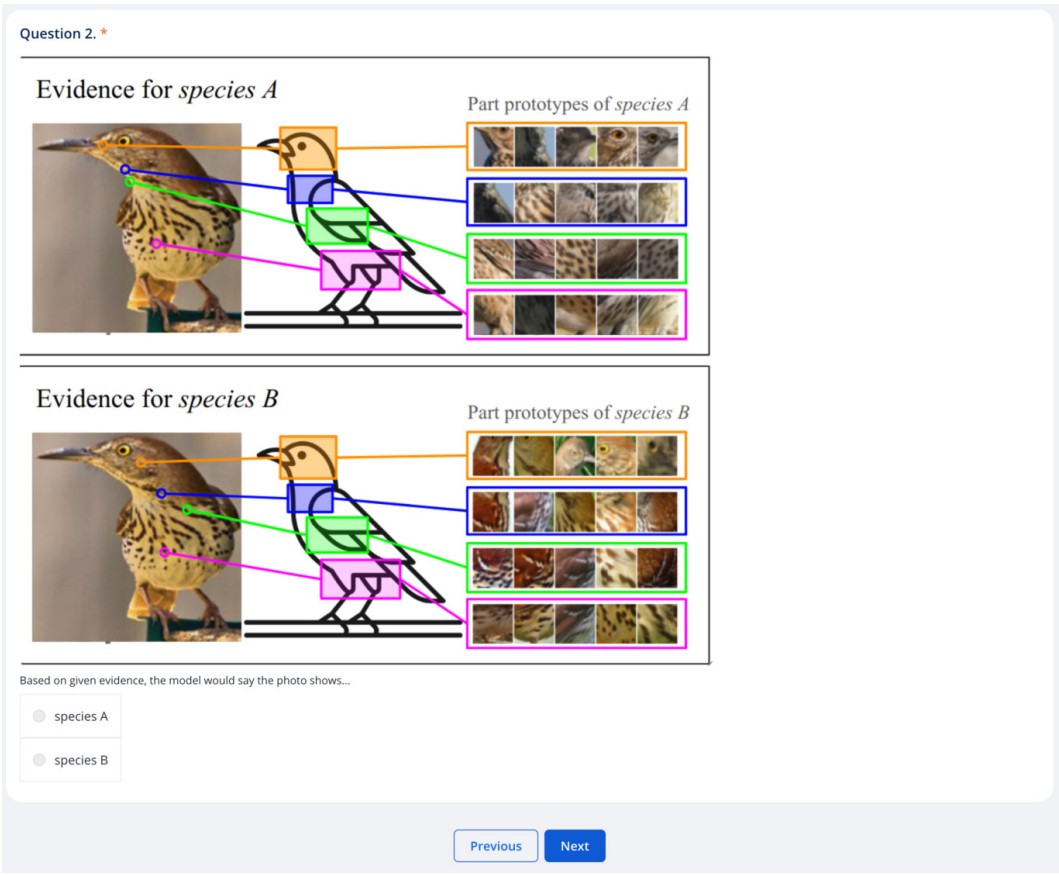

Figure 57: Page 6 of survey for *single branch*

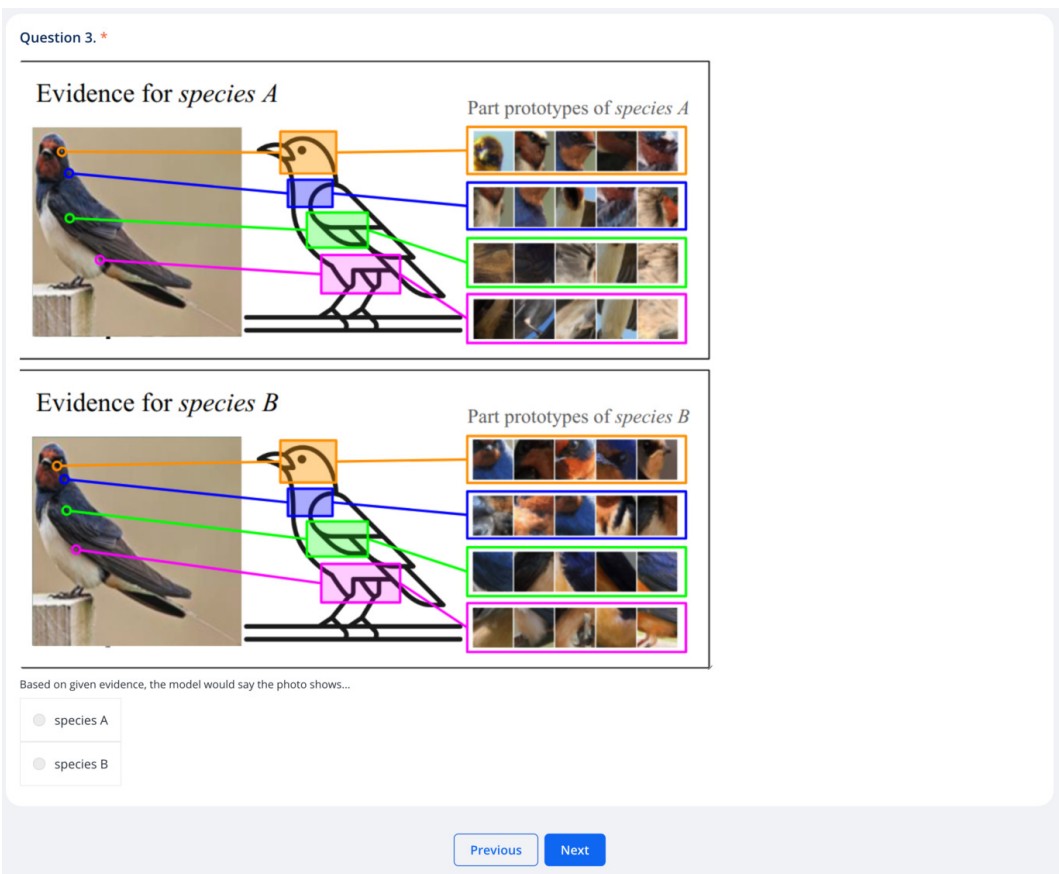

Figure 58: Page 7 of survey for *single branch*

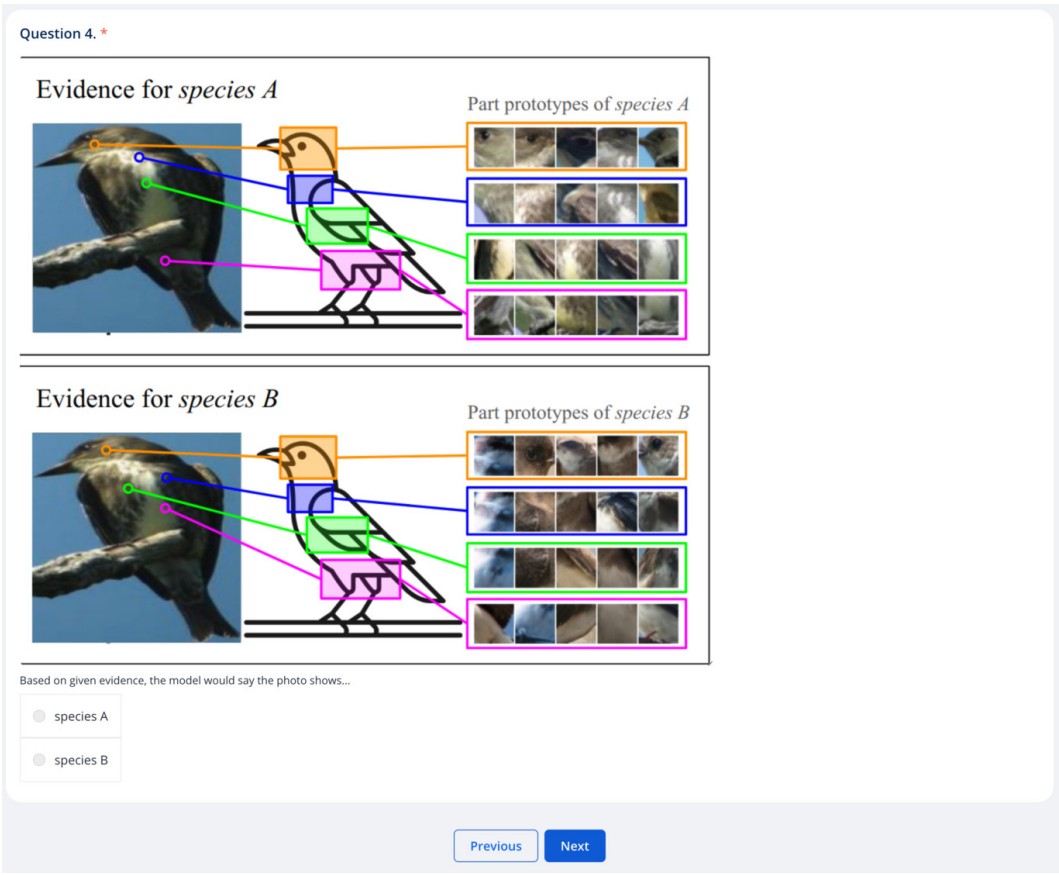

Figure 59: Page 8 of survey for *single branch*

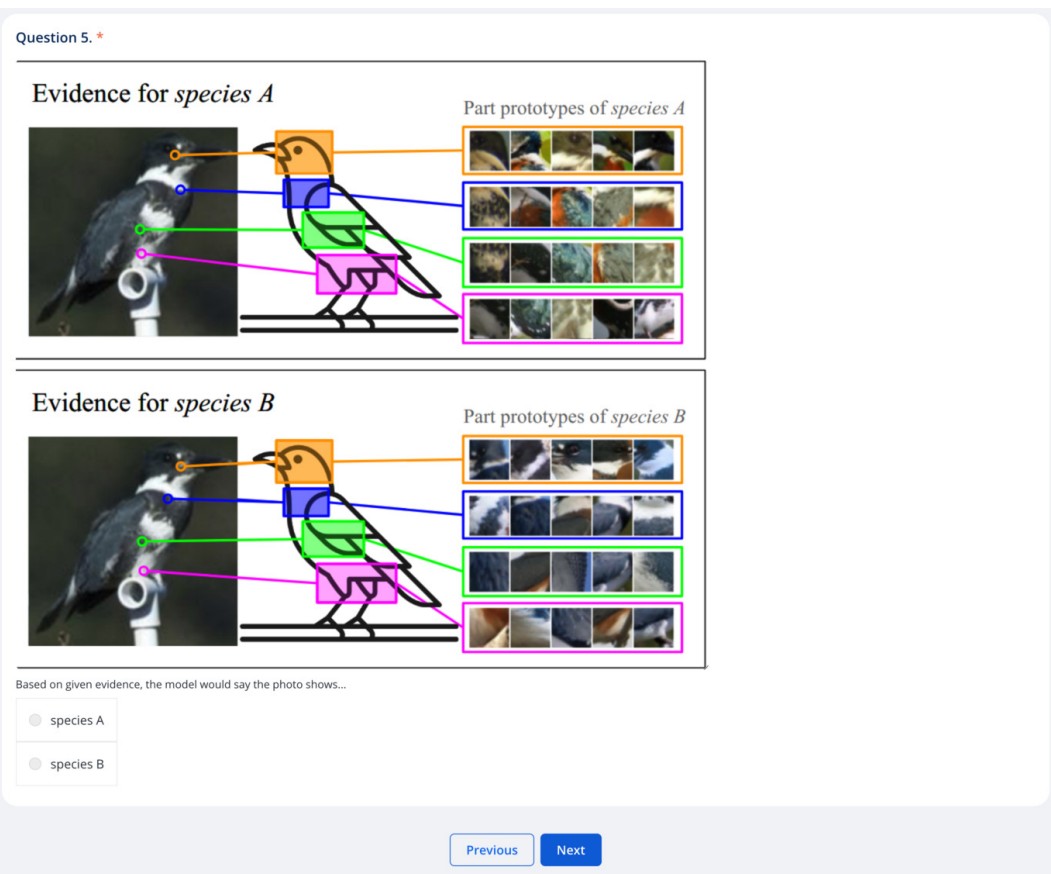

Figure 60: Page 9 of survey for *single branch*

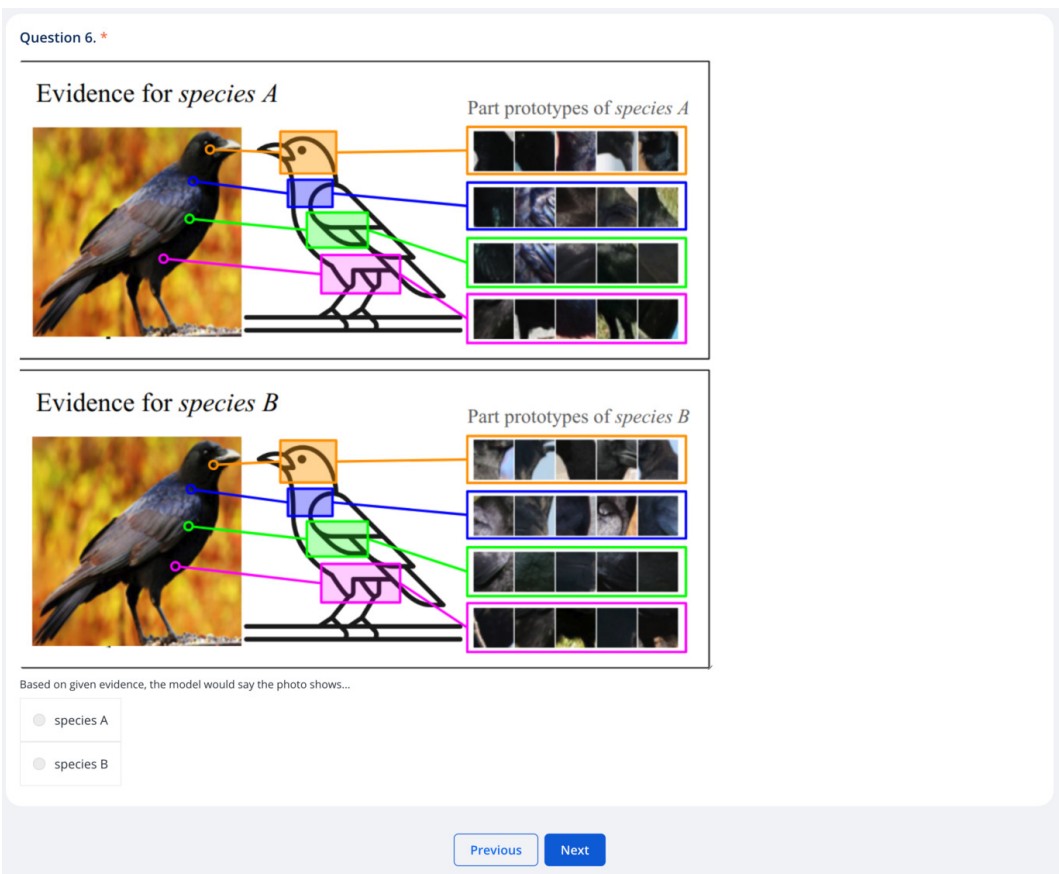

Figure 61: Page 10 of survey for *single branch*

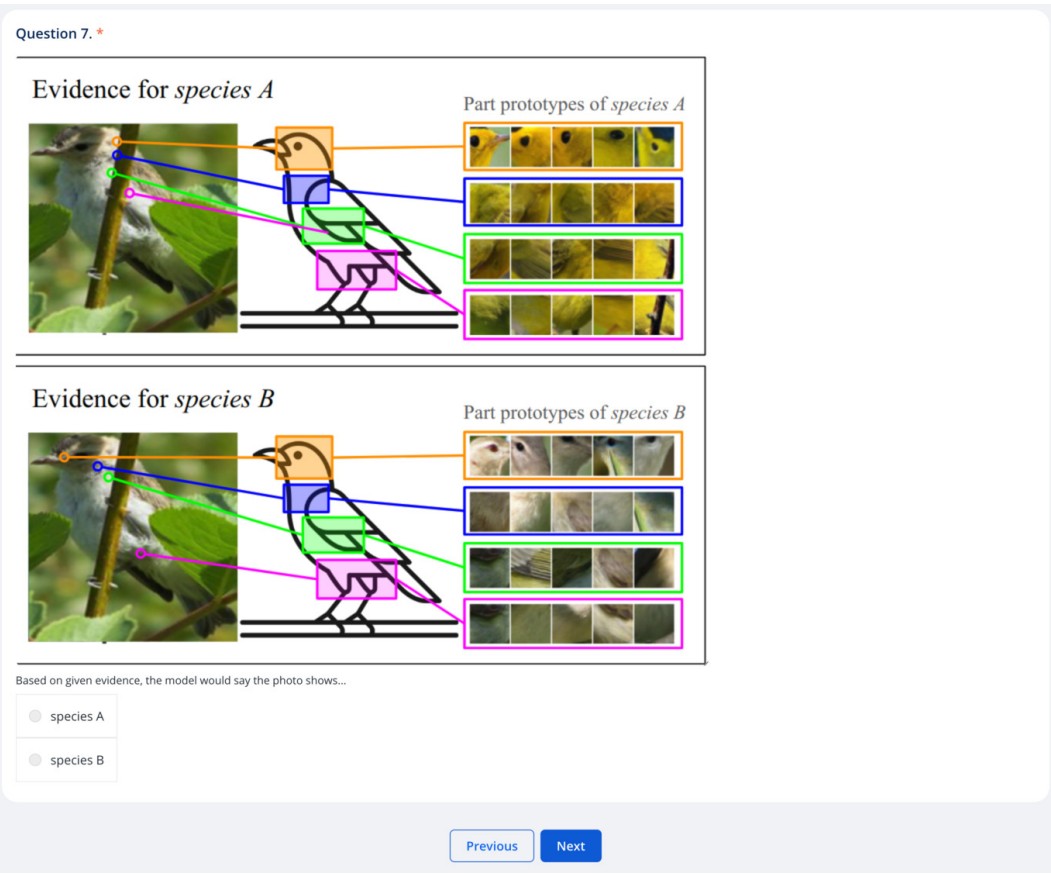

Figure 62: Page 11 of survey for *single branch*

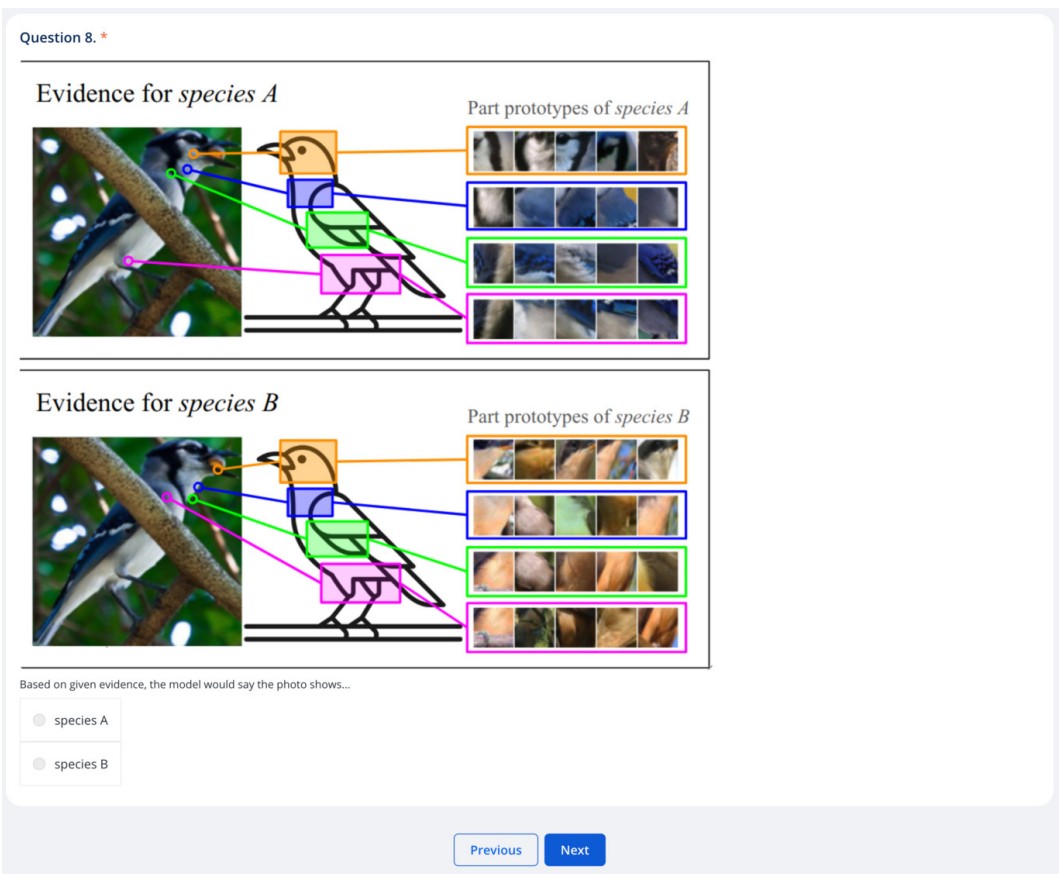

Figure 63: Page 12 of survey for *single branch*

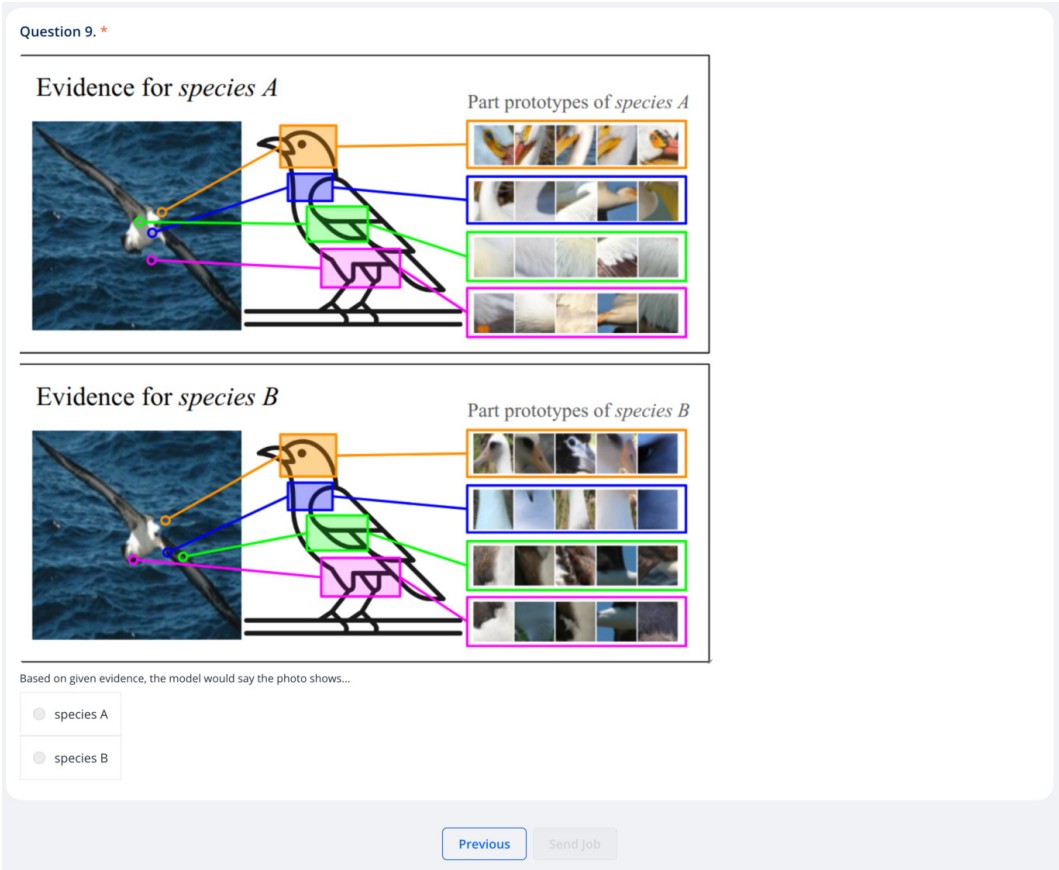

Figure 64: Page 13 of survey for *single branch*

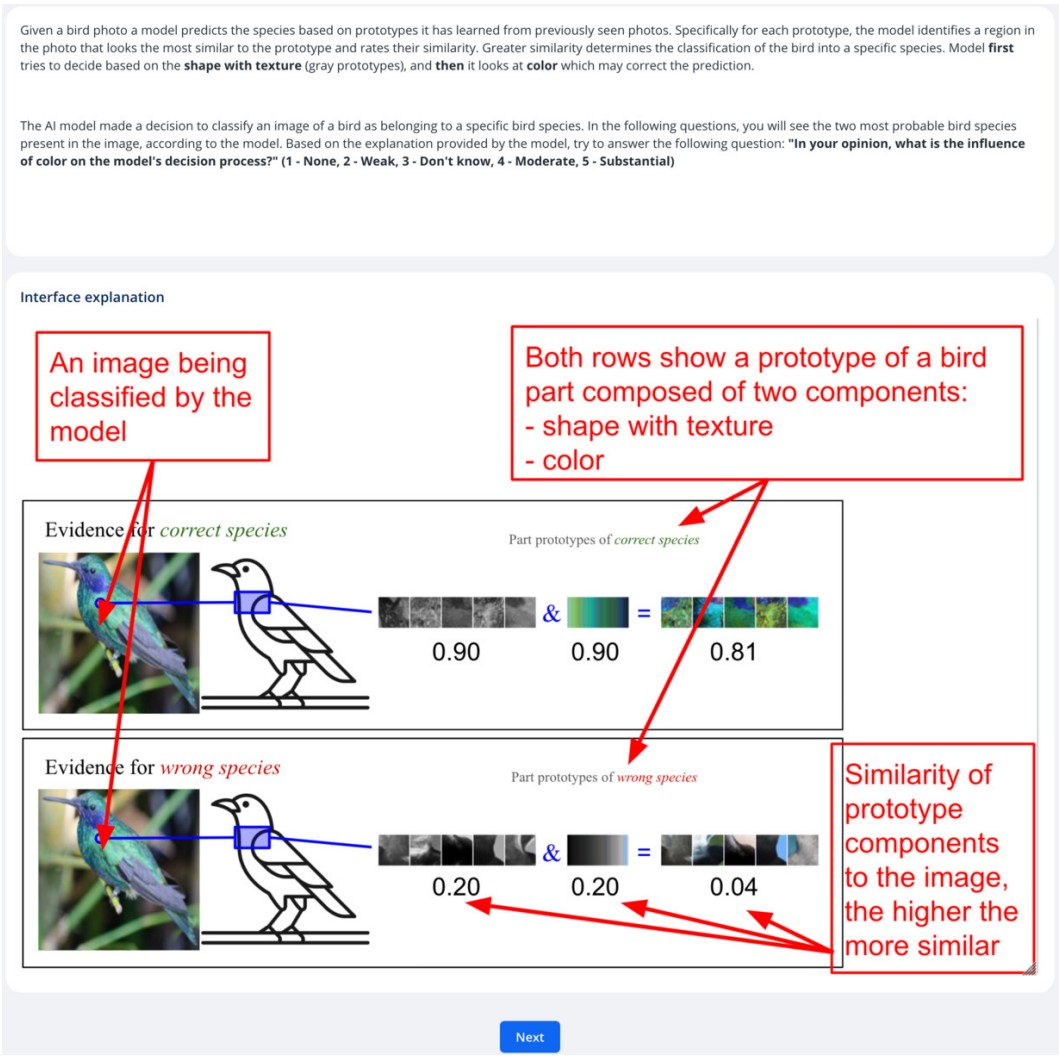

Figure 65: Page 1 of survey for full LucidPPN (with scores)

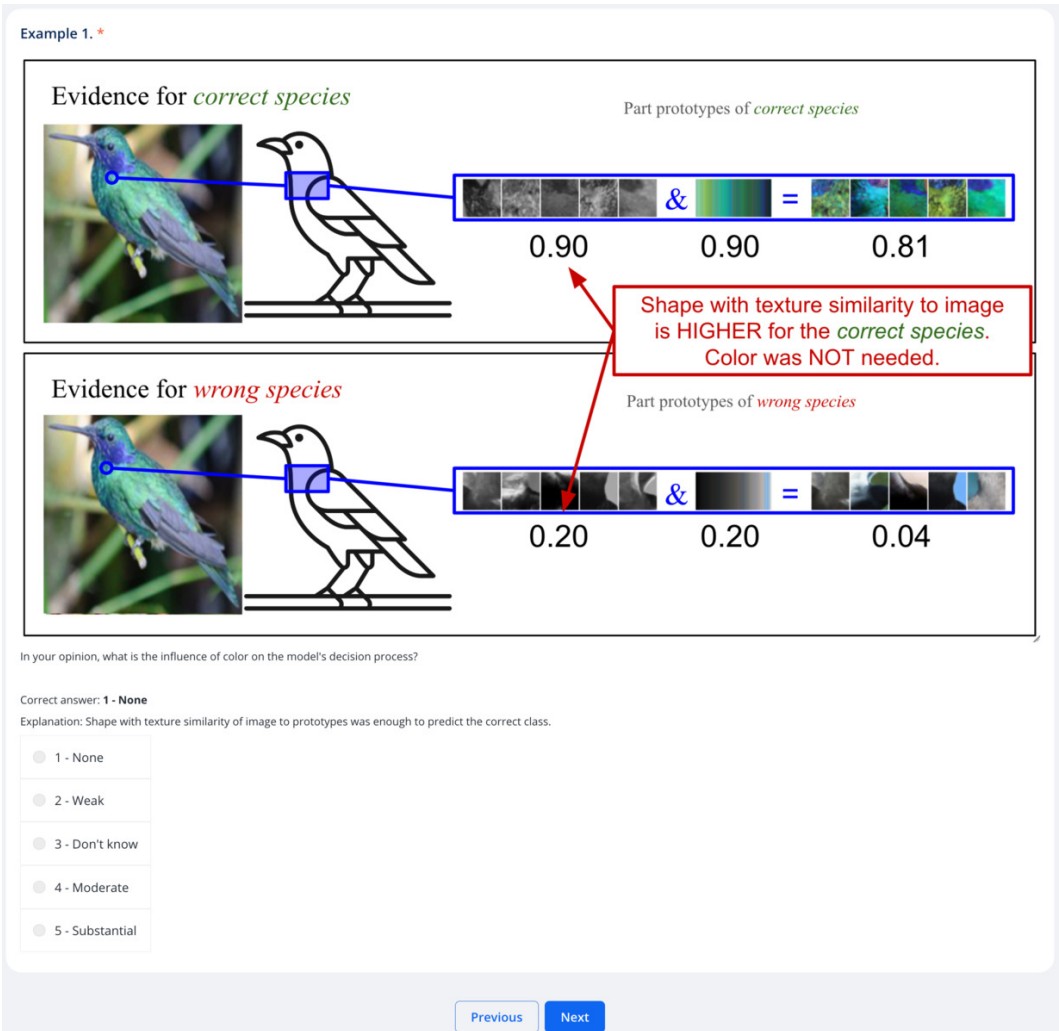

Figure 66: Page 2 of survey for full LucidPPN (with scores)

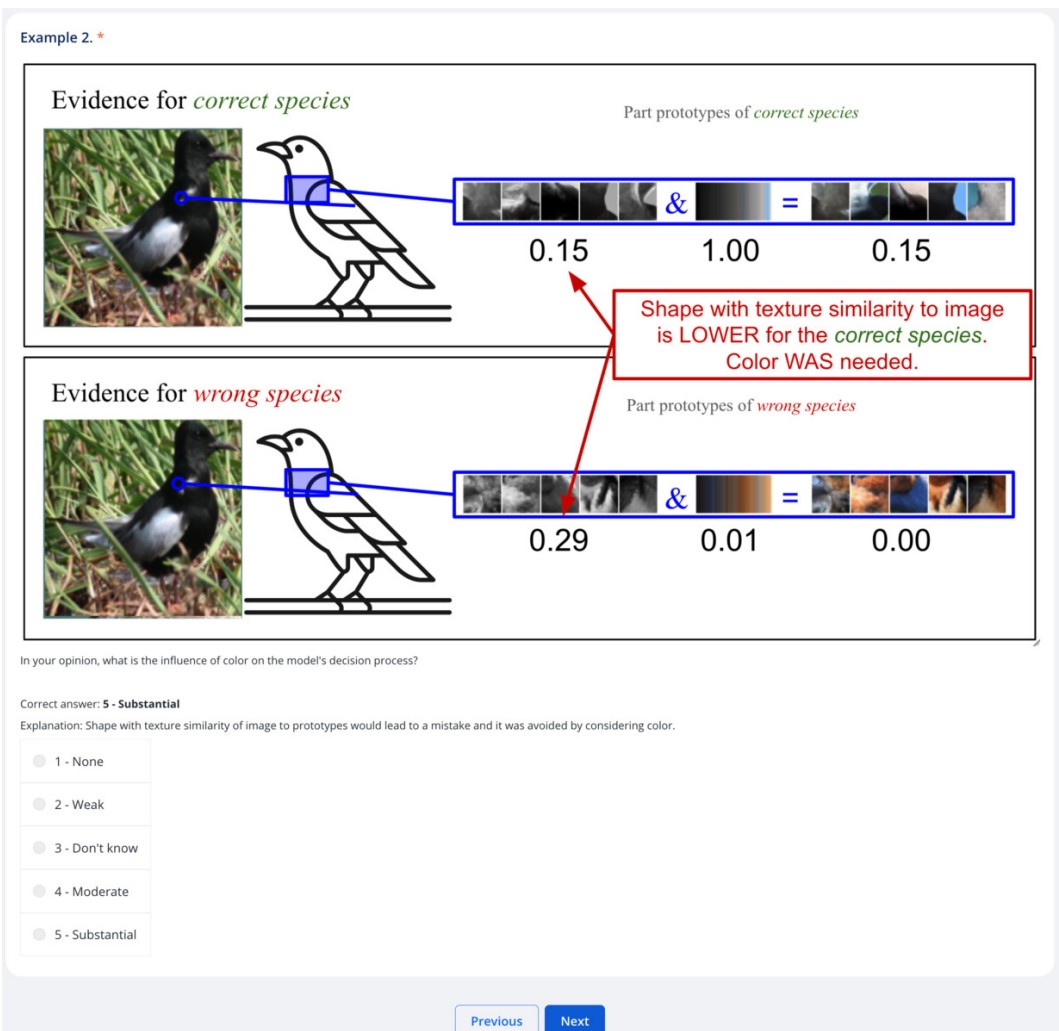

Figure 67: Page 3 of survey for full LucidPPN (with scores)

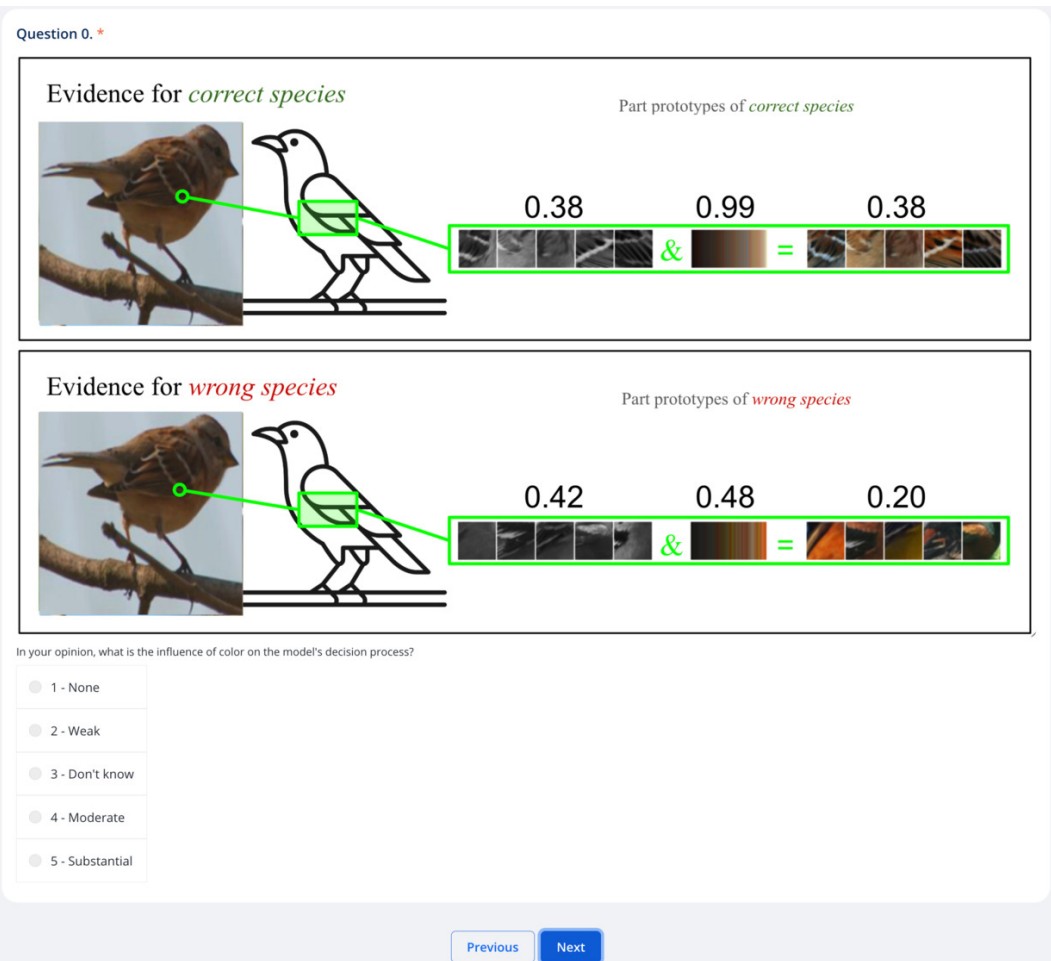

Figure 68: Page 4 of survey for full LucidPPN (with scores)

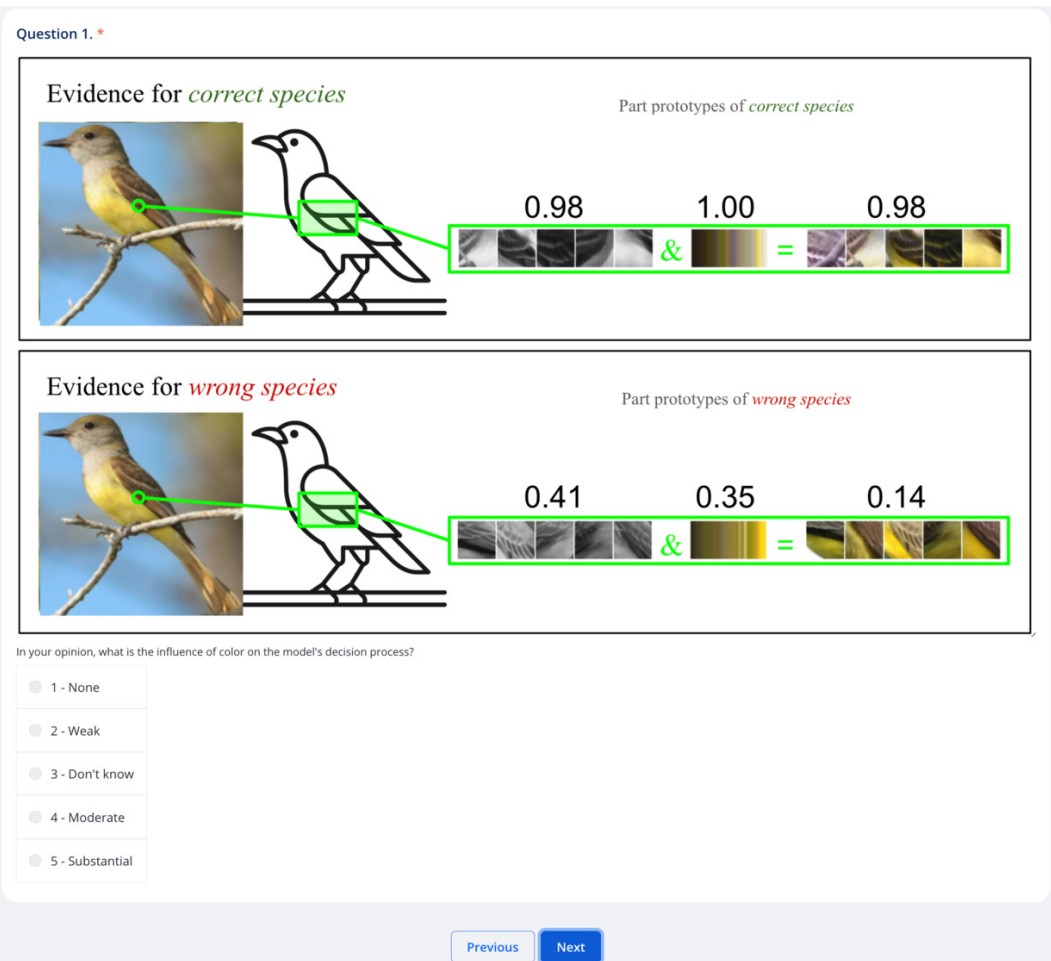

Figure 69: Page 5 of survey for full LucidPPN (with scores)

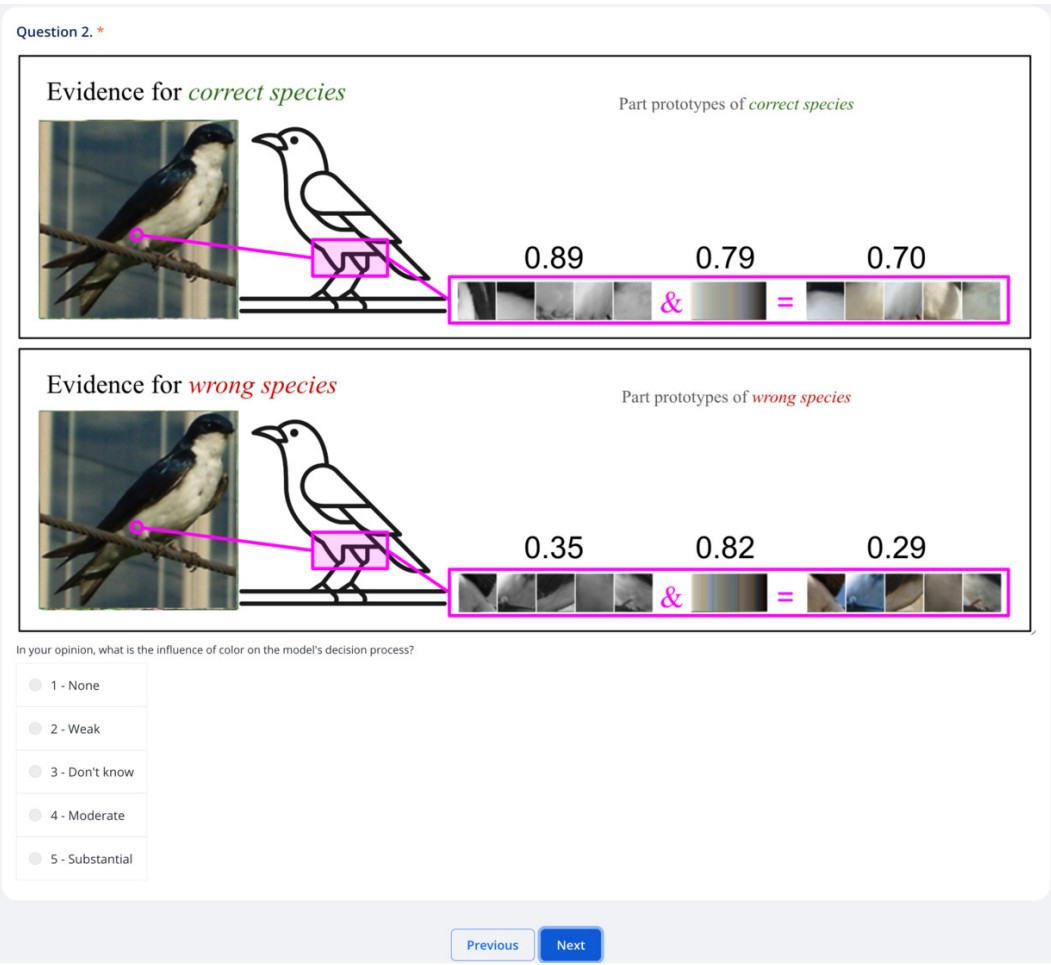

Figure 70: Page 6 of survey for full LucidPPN (with scores)

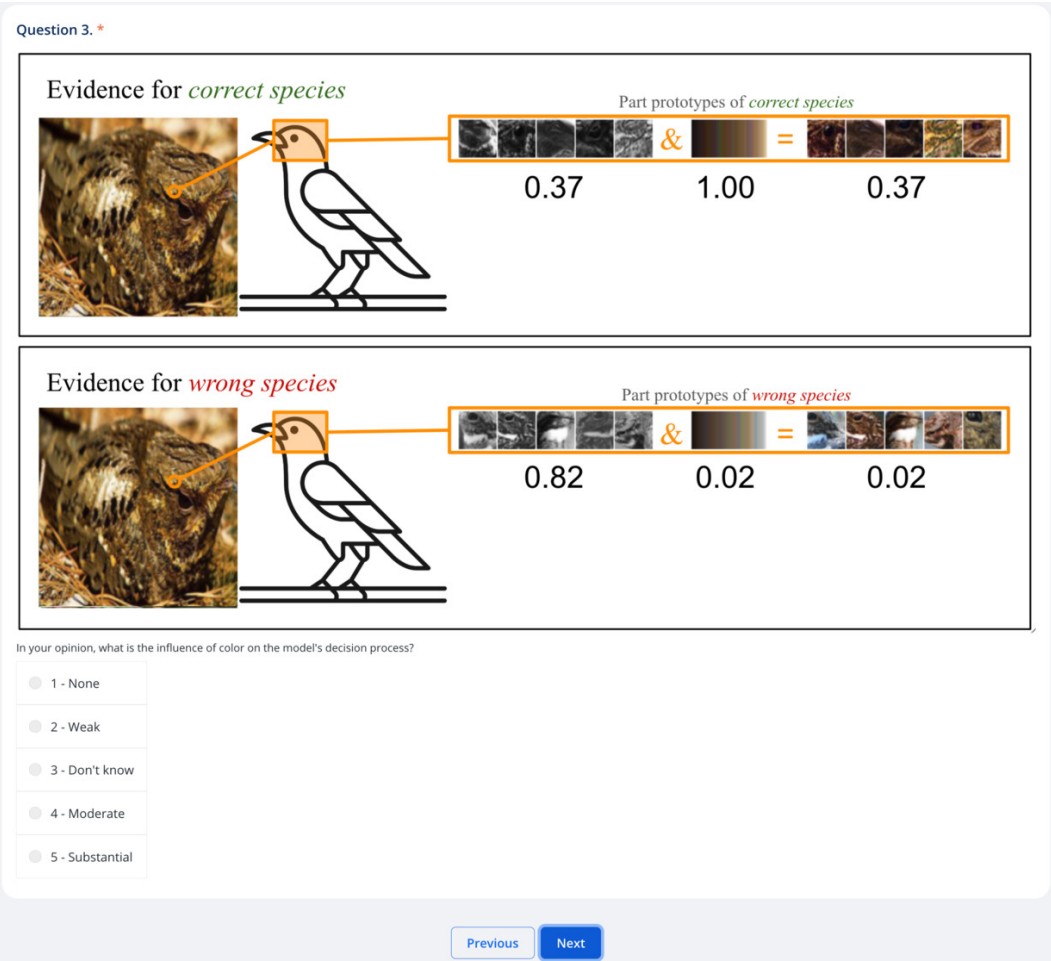

Figure 71: Page 7 of survey for full LucidPPN (with scores)

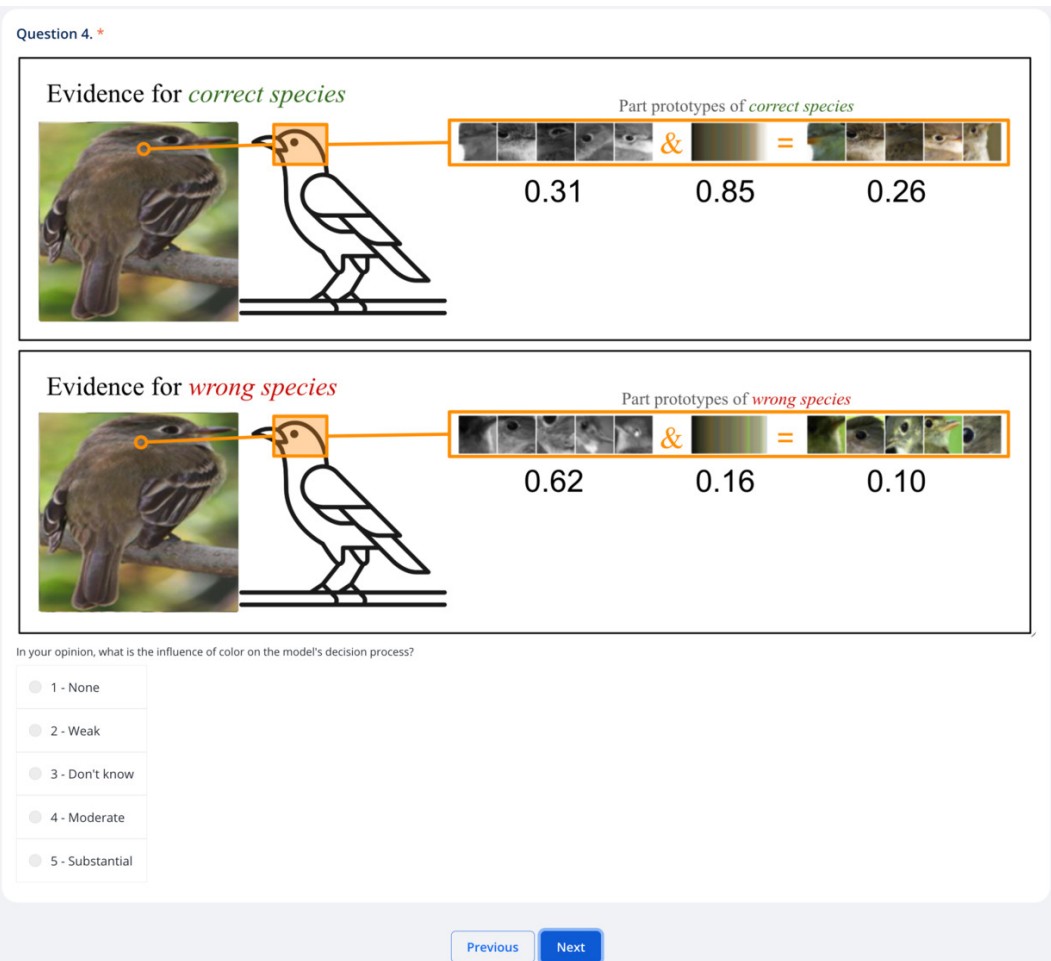

Figure 72: Page 8 of survey for full LucidPPN (with scores)

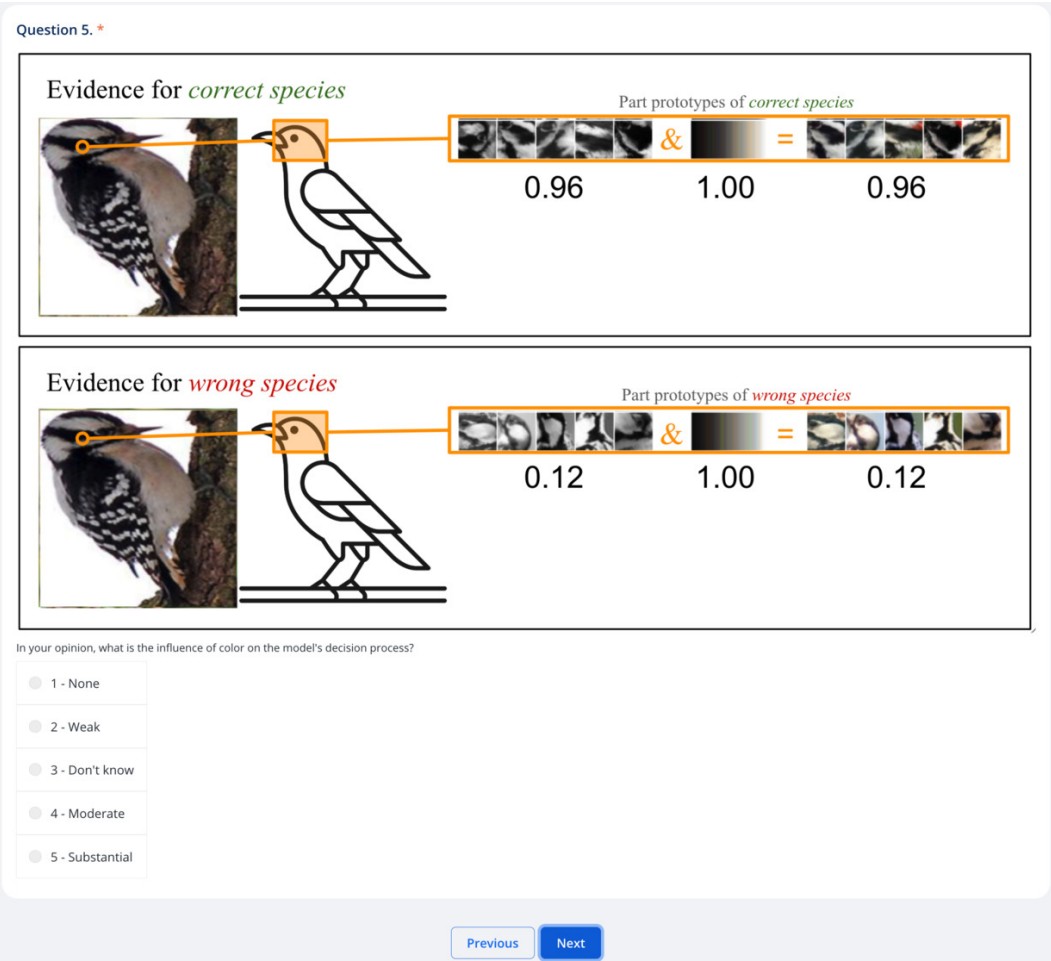

Figure 73: Page 9 of survey for full LucidPPN (with scores)

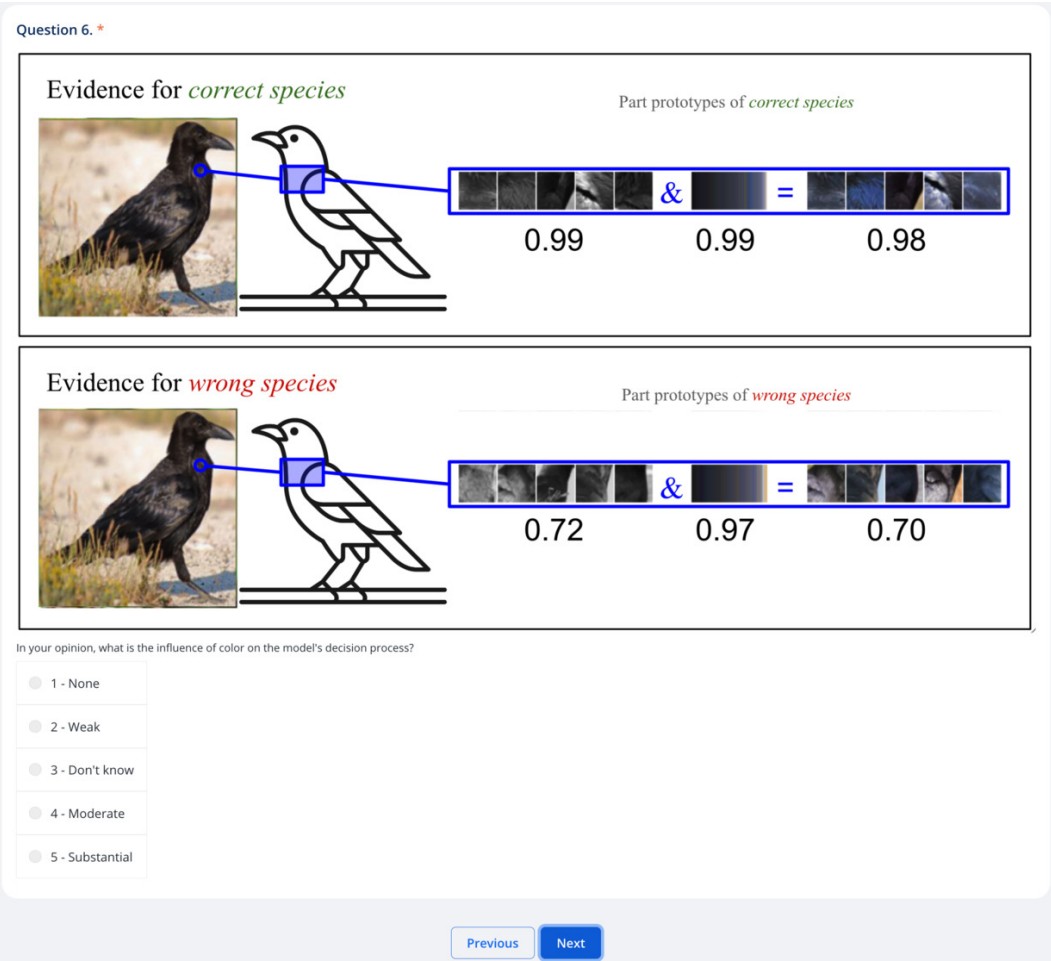

Figure 74: Page 10 of survey for full LucidPPN (with scores)

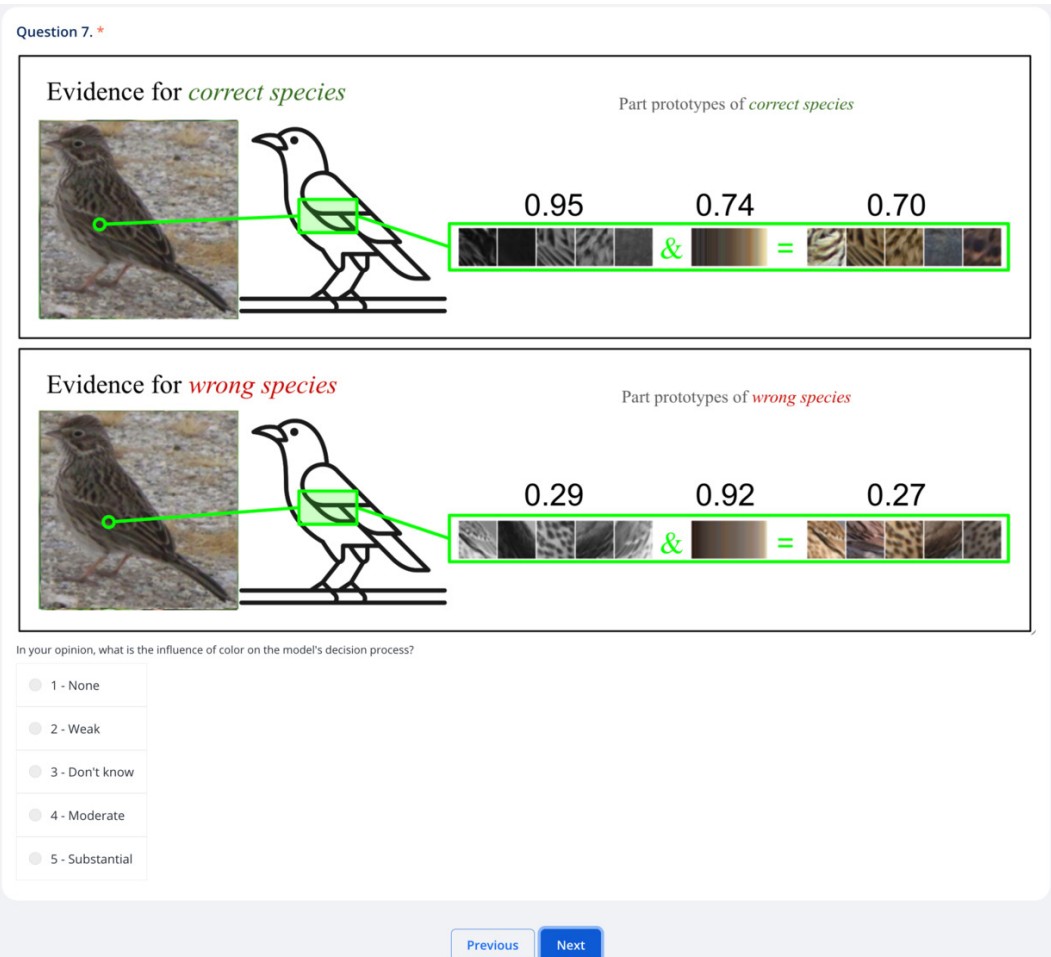

Figure 75: Page 11 of survey for full LucidPPN (with scores)

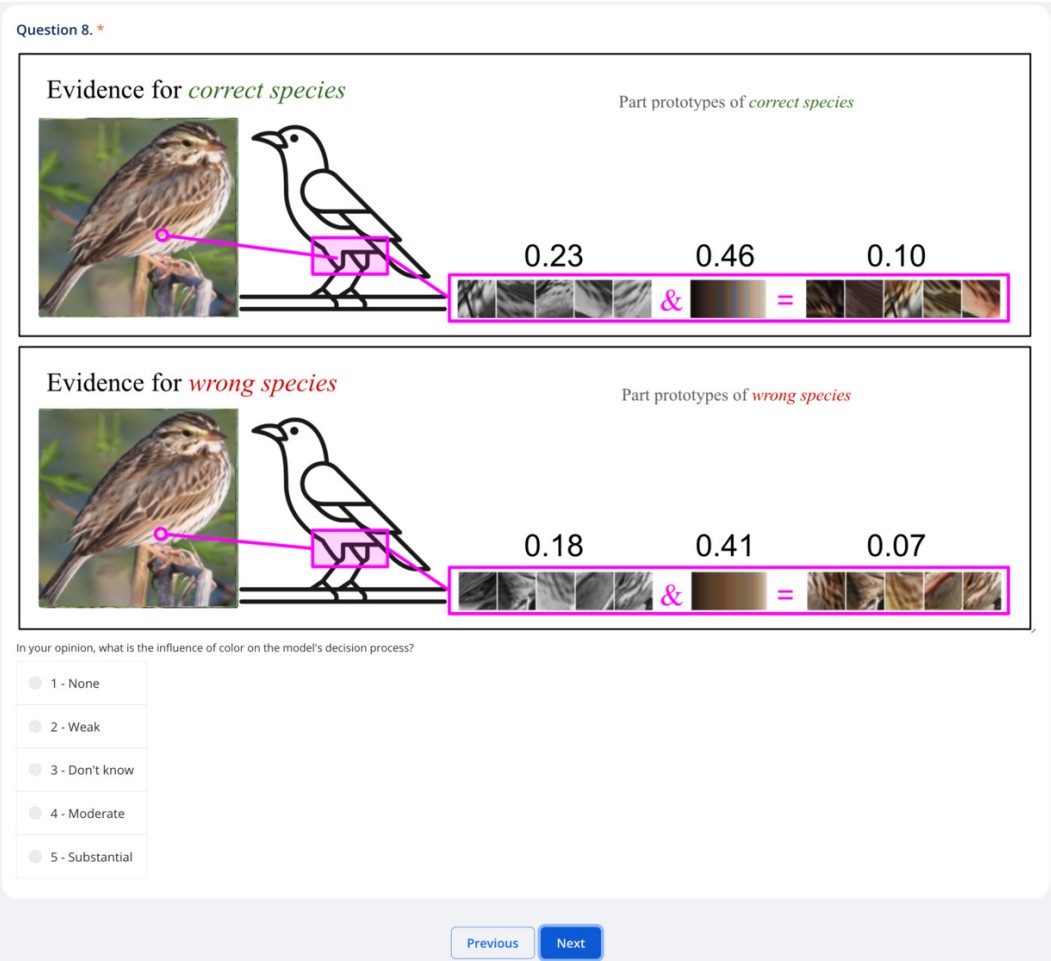

Figure 76: Page 12 of survey for full LucidPPN (with scores)

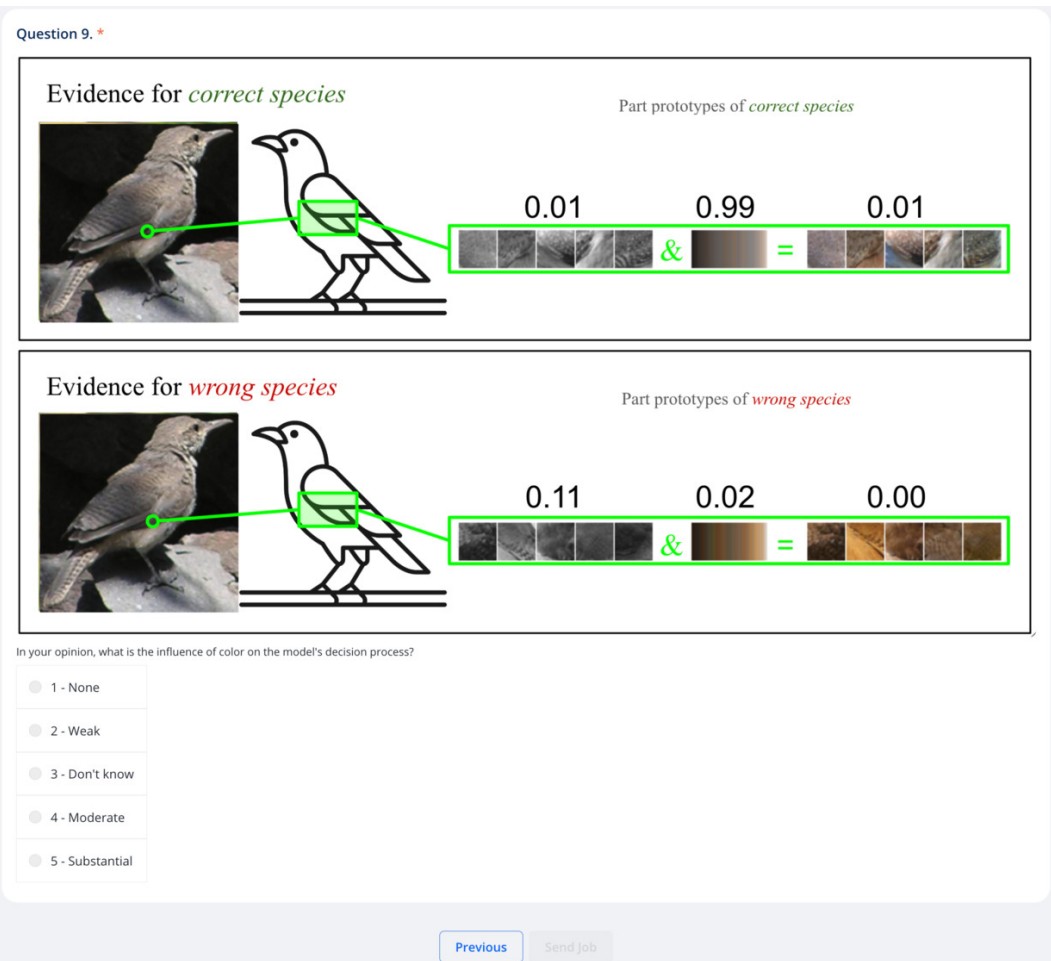

Figure 77: Page 13 of survey for full LucidPPN (with scores)

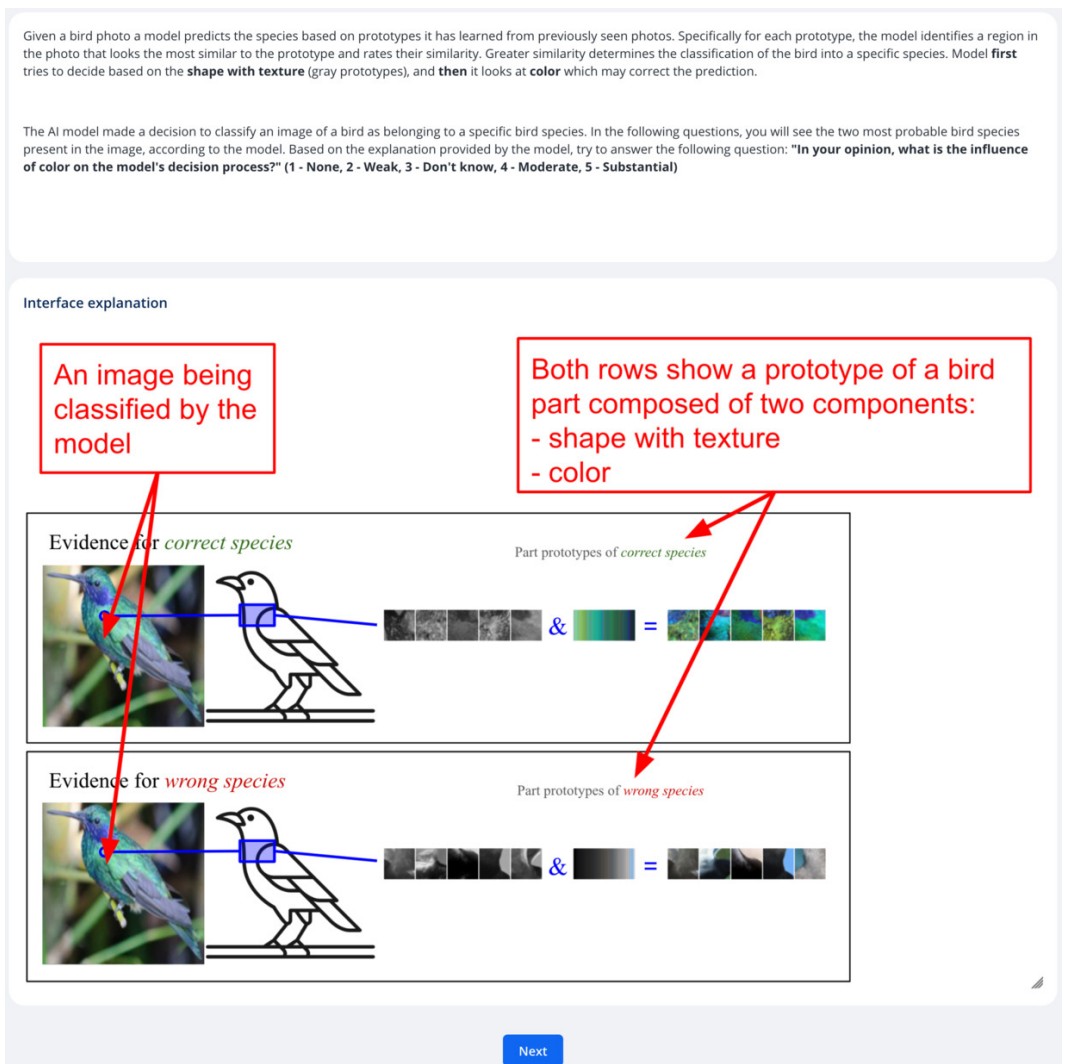

Figure 78: Page 1 of survey for LucidPPN with *no scores*

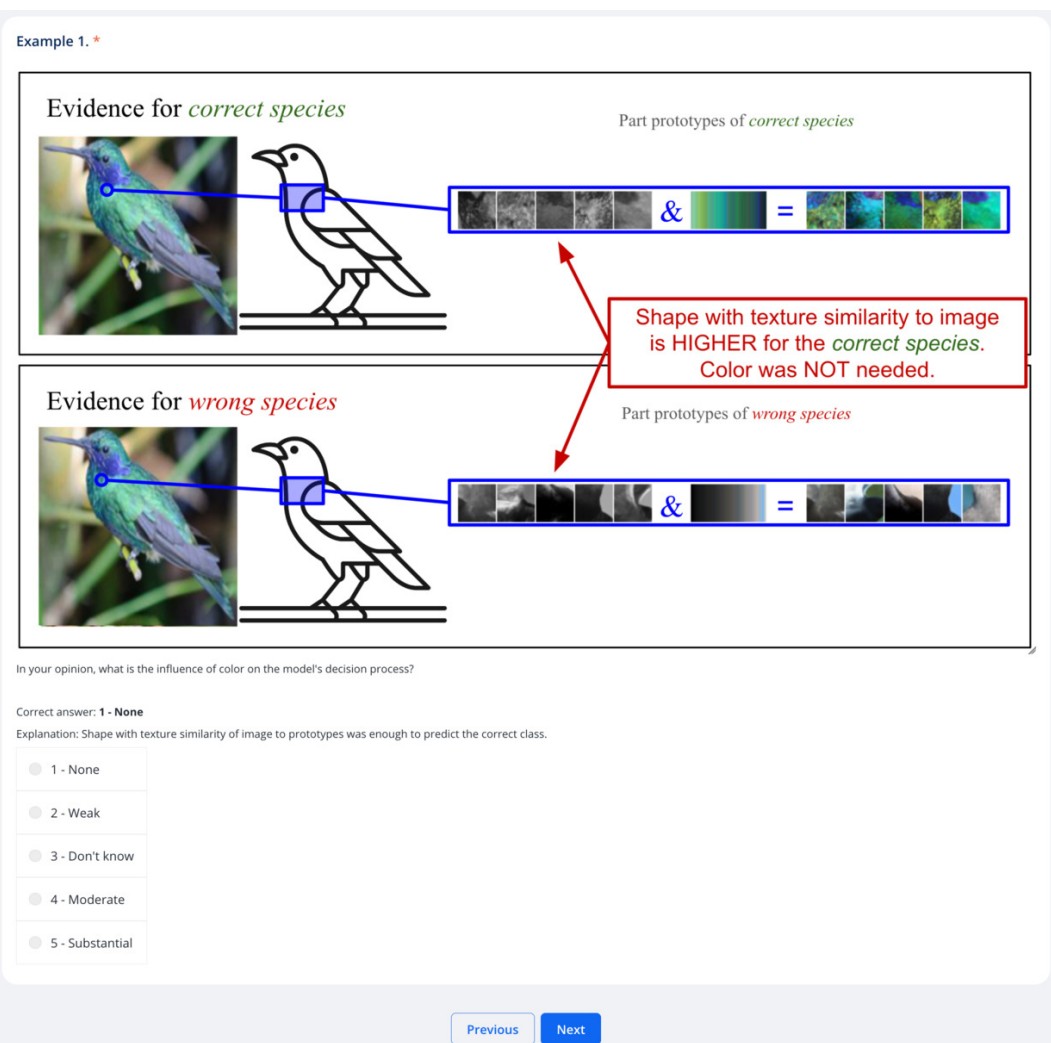

Figure 79: Page 2 of survey for LucidPPN with *no scores*

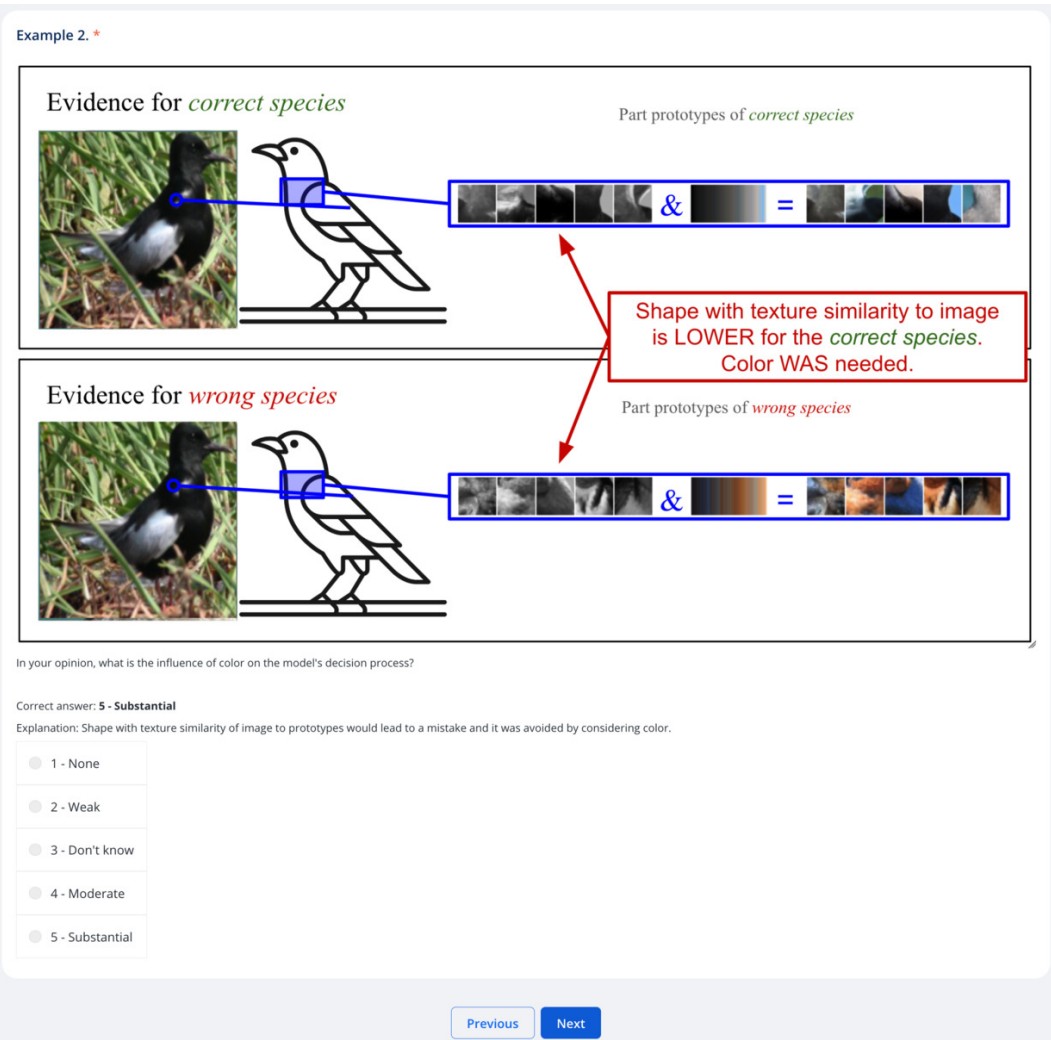

Figure 80: Page 3 of survey for LucidPPN with *no scores*

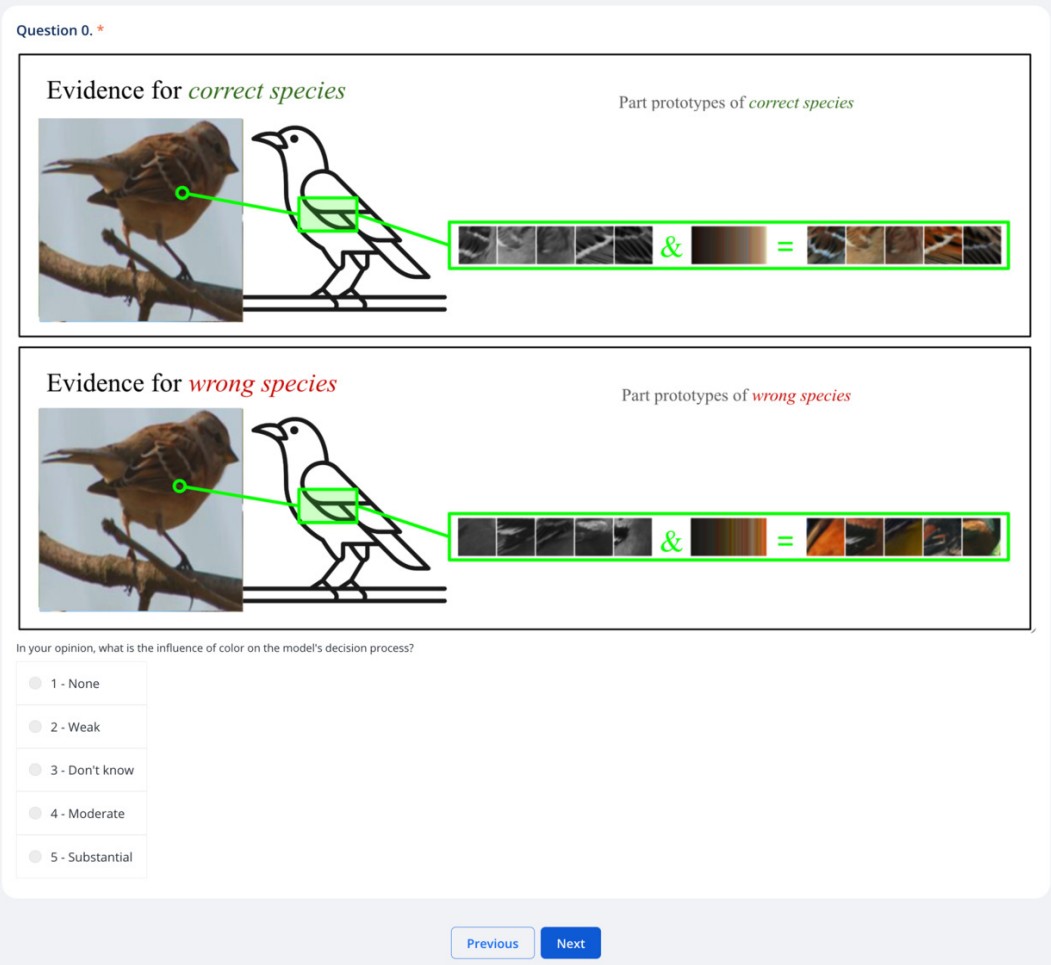

Figure 81: Page 4 of survey for LucidPPN with *no scores*

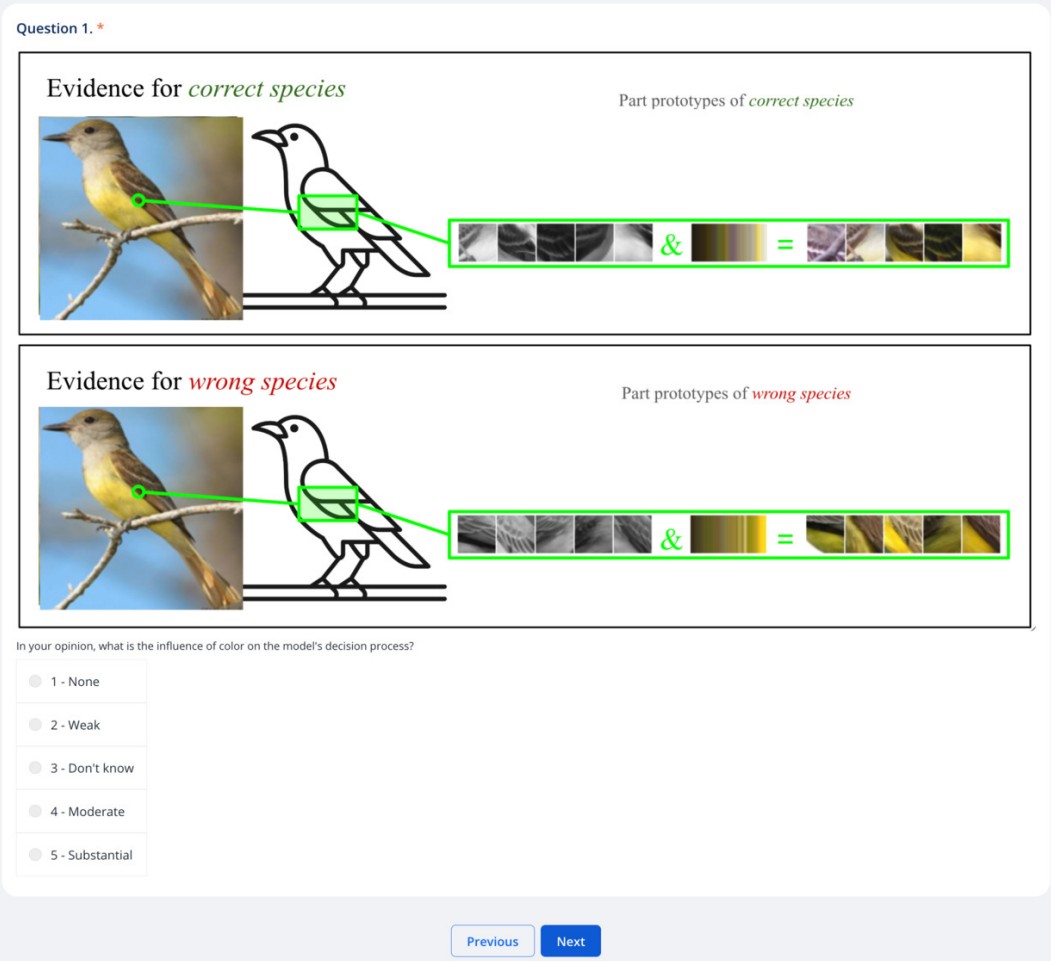

Figure 82: Page 5 of survey for LucidPPN with *no scores*

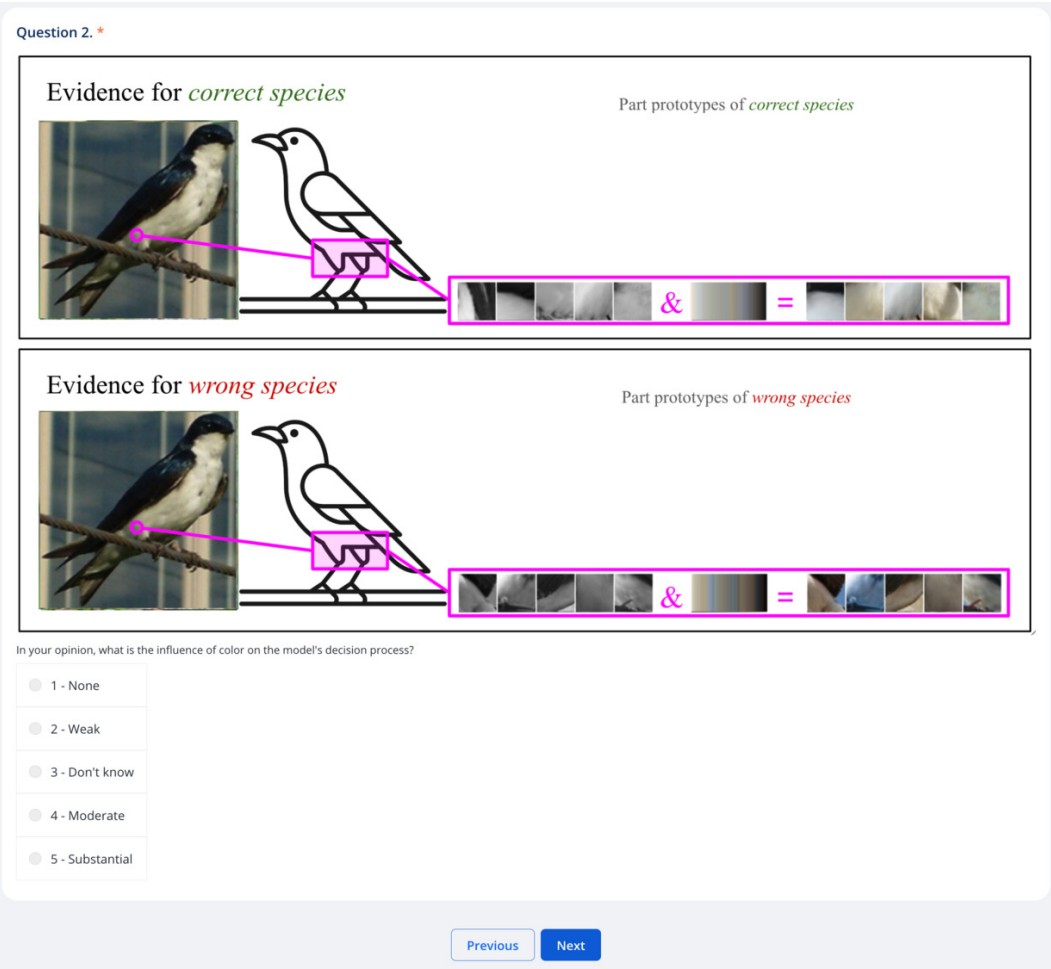

Figure 83: Page 6 of survey for LucidPPN with *no scores*

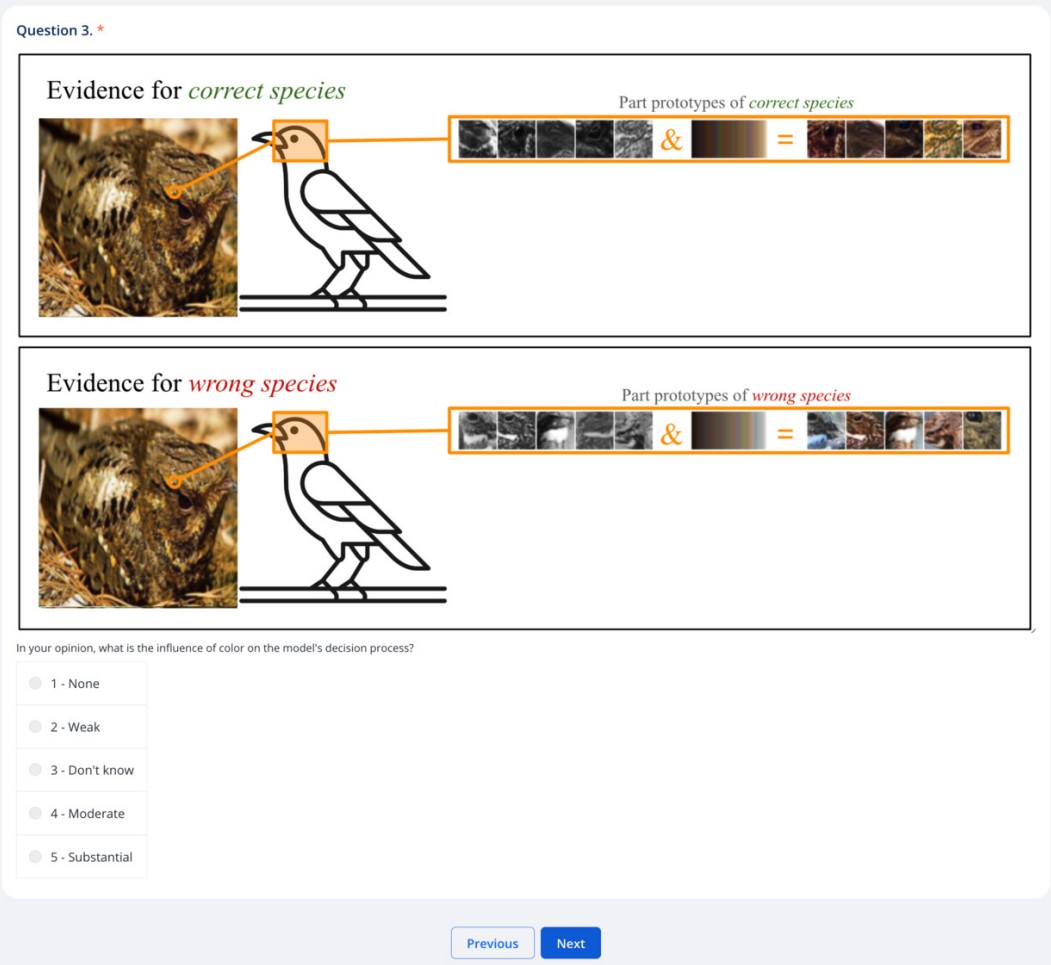

Figure 84: Page 7 of survey for LucidPPN with *no scores*

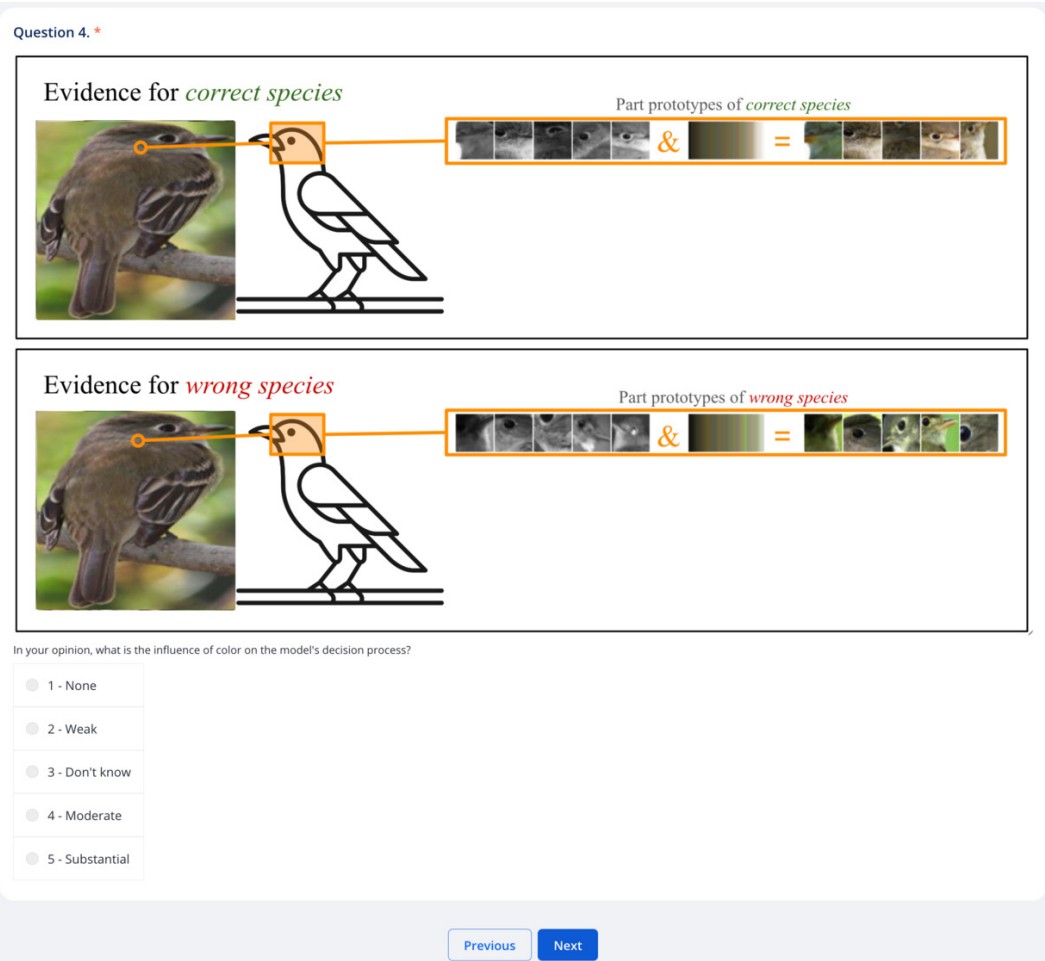

Figure 85: Page 8 of survey for LucidPPN with *no scores*

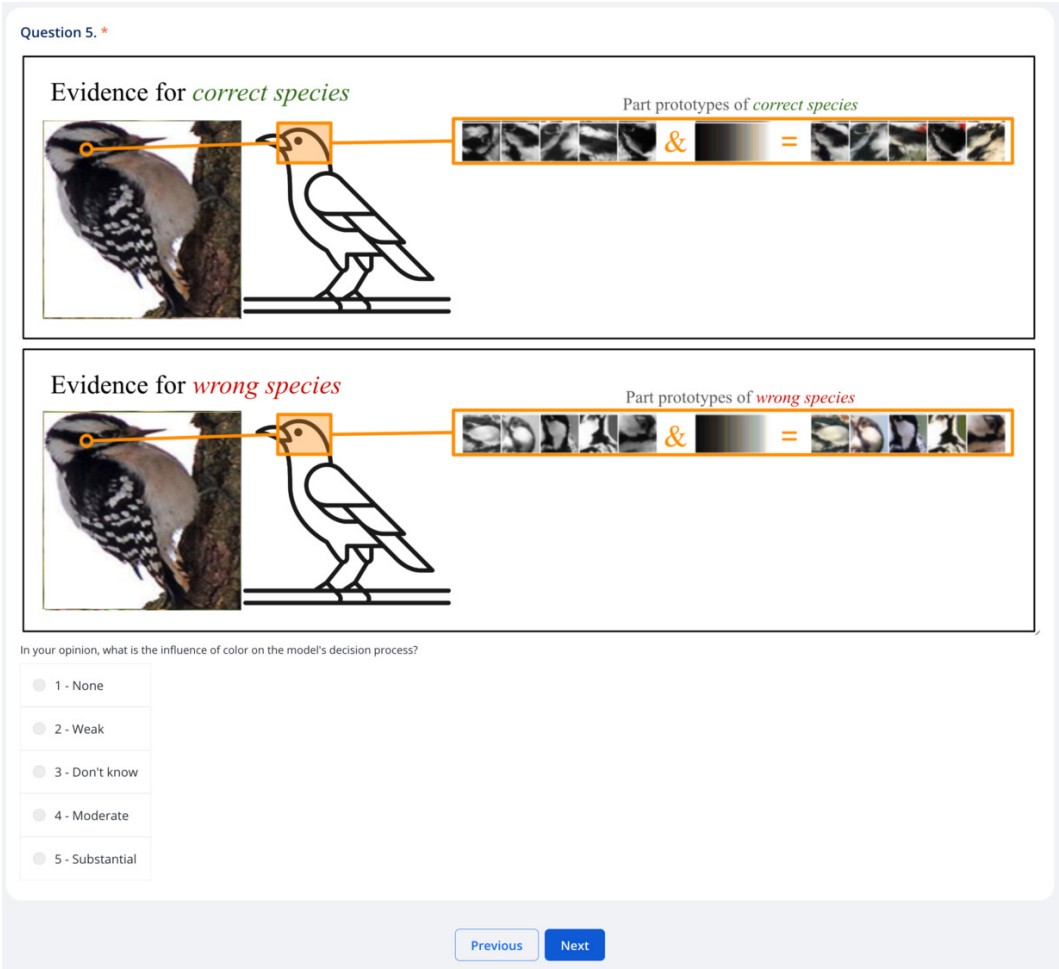

Figure 86: Page 9 of survey for LucidPPN with *no scores*

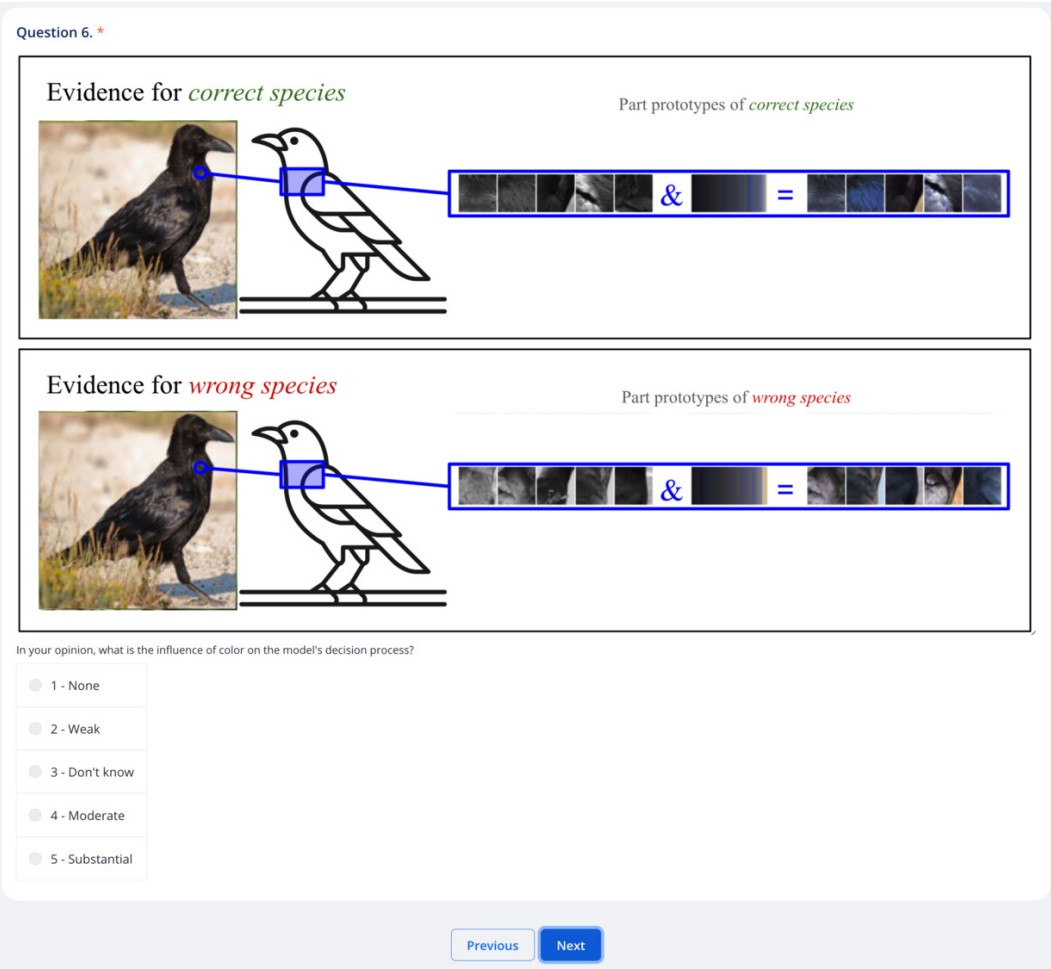

Figure 87: Page 10 of survey for LucidPPN with *no scores*

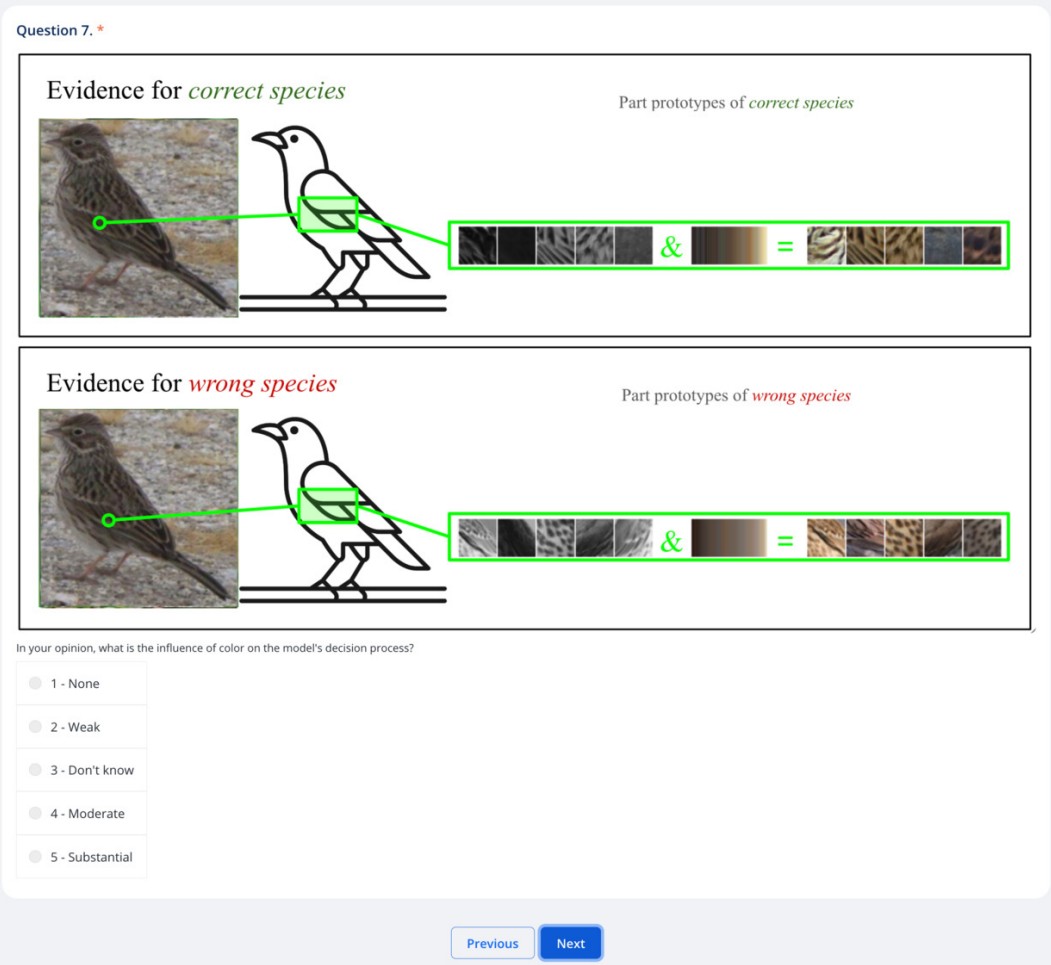

Figure 88: Page 11 of survey for LucidPPN with *no scores*

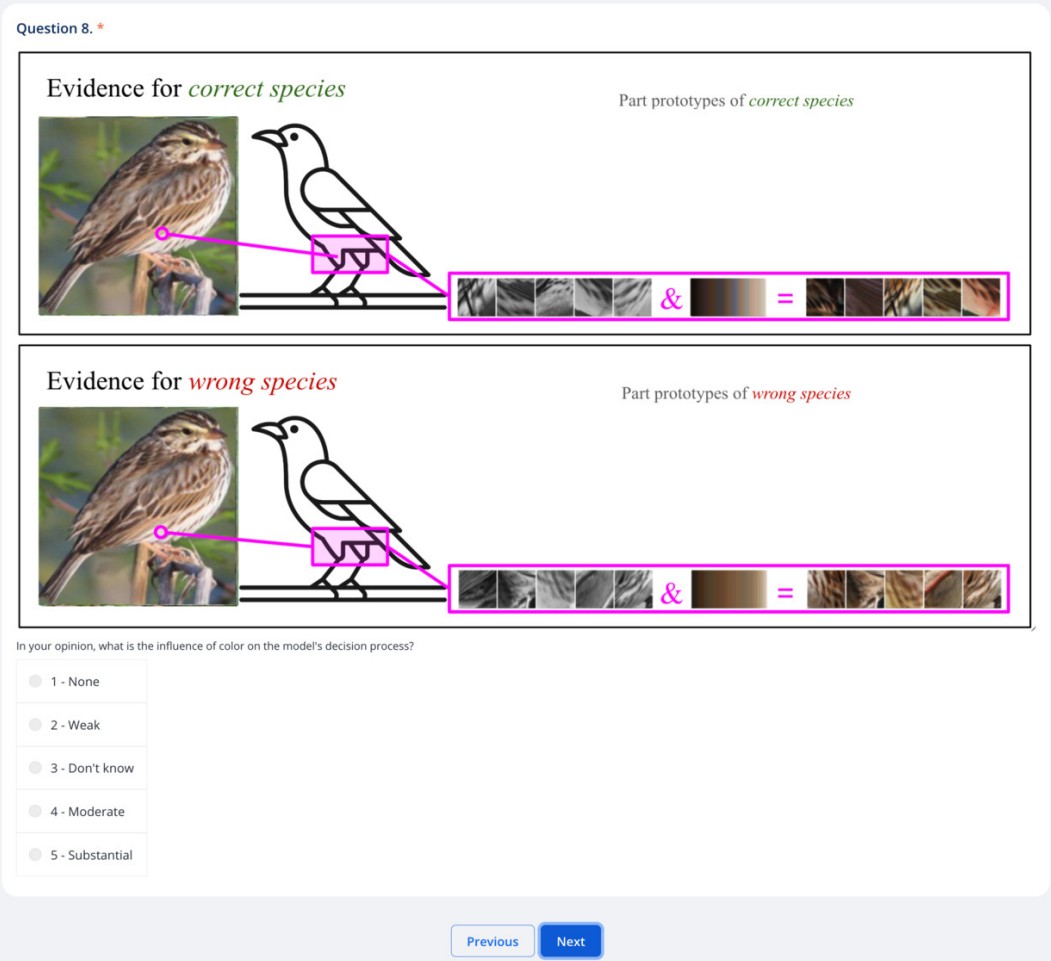

Figure 89: Page 12 of survey for LucidPPN with *no scores*

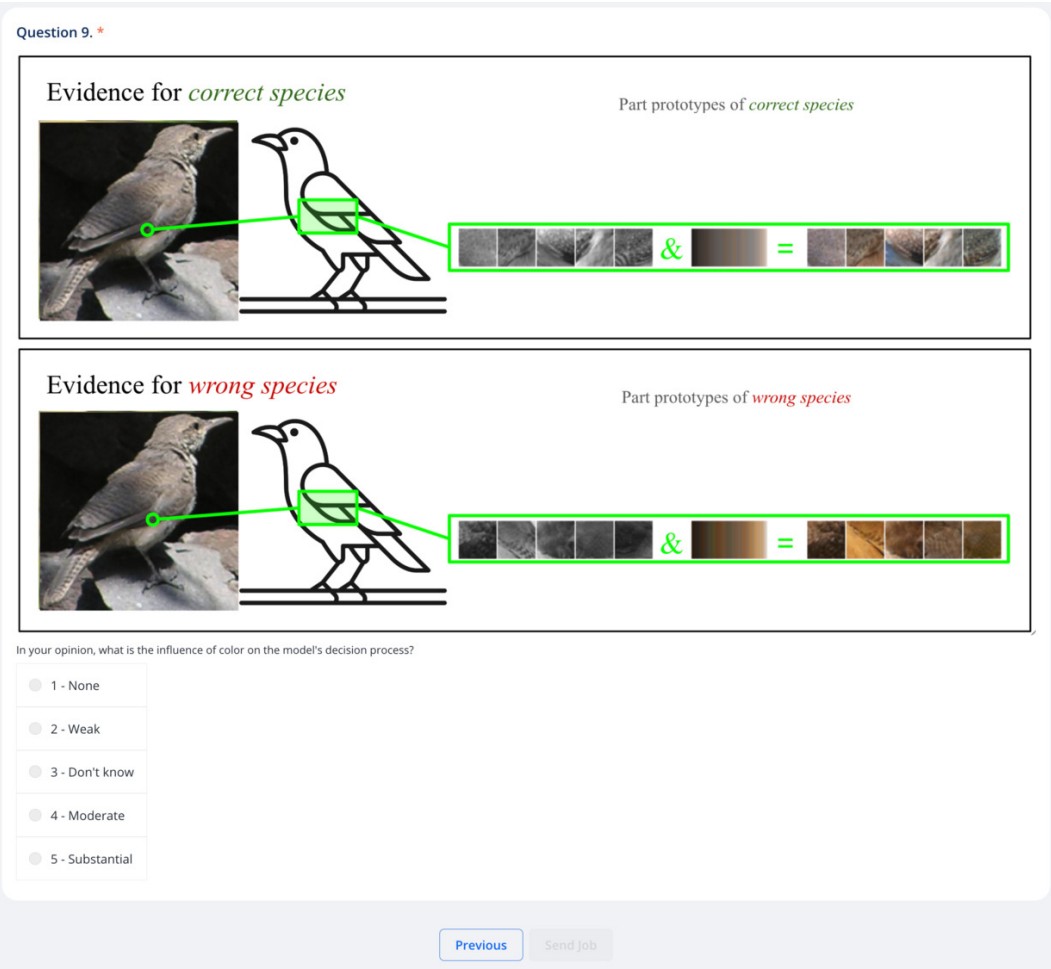

Figure 90: Page 13 of survey for LucidPPN with *no scores*

