# OpenReview forum: "LucidPPN: Unambiguous Prototypical Parts Network for User-centric Interpretable Computer Vision"
_ICLR.cc/2025/Conference — ICLR 2025 Poster_

### Official Review · Reviewer_9WXj · 2024-11-02

**Soundness:** 3
**Presentation:** 3
**Contribution:** 3
**Rating:** 6
**Confidence:** 4

**Summary:**

The manuscript presents the Lucid Prototypical Parts Network (LucidPPN), designed to identify key visual features—specifically color, shape, and texture—based on the prototypical parts networks. The proposed LucidPPN utilizes a non-color branch to process grayscale images alongside a color branch that focuses on color information, thereby clarifying the model's decisions based on these visual attributes. Experimental results demonstrate that the proposed method exhibits advantages over baseline approaches and generates more interpretable prototype parts.

**Strengths:**

(1)	The methodology is well-structured, with intuitive design in the separation of color and non-color network branches, making it accessible and easy to understand.

(2)	The experiments are comprehensive, with a substantial number of visualization results provided in the appendix, enhancing the manuscript's depth.

**Weaknesses:**

(1)	While analyzing "color," "shape," and "texture" offers a valuable perspective, these features have been extensively studied in the field of visual perception. Given that the shallow layers of deep networks are capable of extracting low-level features, the necessity for additional processing and analysis from prototypical parts raises concerns on the novelty and contribution of this work.

(2)	The improvements demonstrated by the proposed method appear to be limited because its performance on some instances is lower than that of the compared methods. For example, in Table 1, the proposed method underperforms other prototypical parts networks on some datasets. While color, shape, and texture are indeed significant visual features in interpretability, they may not be sufficiently critical in this context.

(3)	The organization of the experimental section appears somewhat unbalanced. While the results and visualizations presented are commendable, an excessive amount of content is relegated to the appendix, which may hinder the reader’s ability to grasp key insights and maintain a coherent narrative.

**Questions:**

(1)	The manuscript focuses on interpretability through the lenses of color, shape, and texture. However, other low-level features such as edges, contrast, and spatial frequency are also relevant. Have alternative low-level features also been considered in the analysis?

(2)	The datasets utilized in the experiments are relatively small in size. How will the proposed method perform on larger datasets, such as ImageNet? Some insights into performance scalability would be beneficial.

(3)	The manuscript primarily presents visualization results for the prototypical parts identified by the proposed method. How do these results compare with other prototypical parts-based models? A comparative analysis would enhance the understanding of the method's effectiveness.

(4)	In global feature visualizations, such as Figure 14, the manuscript illustrates the ability of the proposed method to detect shape and color. How does this compare with traditional edge detection operators (e.g., Sobel) for shape extraction and color feature extraction methods (e.g., color histogram)? Additionally, how does it compare with the direct visualizations of shallow layer attention to texture and color using techniques like Grad-CAM?

---

> ### Author Response · Authors · 2024-11-22
>
> **W1. While analyzing ”color,” ”shape,” and ”texture” offers a valuable perspective, these features have been extensively studied in the field of visual perception. Given that the shallow layers of deep networks are capable of extracting low-level features, the necessity for additional processing and analysis from prototypical parts raises concerns on the novelty and contribution of this work.**
>
> It is true that shallow layers of neural networks are capable of extracting low-level features. The goal of LucidPPN, however, is to make this processing more transparent to the user, which aligns with the broader objective of inherently interpretable models Chen et al. (2019); Rudin (2019). Thanks to LucidPPN we can analyze which colors were important for classification. Conventional neural networks often entangle visual features in ways that make it difficult to disentangle and present them in an understandable format for users. This is why we believe our work offers a novel contribution to the field of interpretable AI, particularly in the context of prototypical parts.
>
>
> **W2. The improvements demonstrated by the proposed method appear to be limited because its performance on some instances is lower than that of the compared methods**
>
> We respond to this comment in **shared remarks** in paragraphs *The improvements demonstrated by the proposed method appear to be limited because its performance on some instances is lower than that of the compared methods.* and *There was no noticeable advantage in accuracy. Why?*
>
> **W3. The organization of the experimental section appears somewhat unbalanced. While the results and visualizations presented are commendable, an excessive amount of content is relegated to the appendix, which may hinder the reader’s ability to grasp key insights and maintain a coherent narrative.**
>
> We agree and in the revised version of the manuscript, we have reorganized the experimental section. However, due to space constraints and to adhere to the ICLR template, some content has been moved to the appendix.

---

> ### Author Response · Authors · 2024-11-22
>
> **Q1. The manuscript focuses on interpretability through the lenses of color, shape, and texture. However, other low-level features such as edges, contrast, and spatial frequency are also relevant. Have alternative low-level features also been considered in the analysis?**
>
> This work represents the first step toward disentangling low-level features. As mentioned in the Limitations Section, we plan to explore low-level features in more detail in future work. For now, we focus on extracting color information and correlating prototypical parts with semantic object parts. These contributions already make the work comprehensive. Introducing additional low-level feature extraction and integration at this stage would complicate the model further and make it more difficult to communicate.
>
>
> **Q2. The datasets utilized in the experiments are relatively small in size. How will the proposed method perform on larger datasets, such as ImageNet? Some insights into performance scalability would be beneficial.**
>
> Most benchmarking of prototypical part-based methods has been conducted on fine-grained datasets such as CUB and Stanford Cars Chen et al. (2019); Nauta et al. (2021; 2023b); Rymarczyk et al. (2021; 2022; 2023); Wang et al. (2024). Scaling these architectures to ImageNet-sized datasets is an important but orthogonal research direction that remains unsolved. However, to assess whether LucidPPN generalizes to broader classification tasks (beyond fine-grained datasets), we added results on PartImageNet He et al. (2022) in the Supplementary Materials. For this dataset, LucidPPN achieves an accuracy of 84.1%, outperforming PIPNet, which achieves 82.8%.
>
> **Q3. The manuscript primarily presents visualization results for the prototypical parts identified by the proposed method. How do these results compare with other prototypical parts-based models? A comparative analysis would enhance the understanding of the method’s effectiveness.**
>
> In Supplementary Figures 15-19, we added explanations from various prototypical-part-based methods. Additionally, our user studies demonstrate the effectiveness of our explanations from the user’s perspective.
>
>
> **Q4. In global feature visualizations, such as Figure 14, the manuscript illustrates the ability of the proposed method to detect shape and color. How does this compare with traditional edge detection operators (e.g., Sobel) for shape extraction and color feature extraction methods (e.g., color histogram)? Additionally, how does it compare with the direct visualizations of shallow layer attention to texture and color using techniques like Grad-CAM?**
>
> Edge detectors and color histograms are not trainable and do not represent high-level features, unlike prototypical parts. Even our low-level color prototypical representation captures a higher-level concept, such as a ”red tail,” which is then further decomposed into its components—color (red) and shape/texture (tail). Regarding GradCAM, we want to highlight that post-hoc methods can often be unreliable, as demonstrated in multiple studies Adebayo et al. (2018); Kim et al. (2022); Rudin (2019); Tomsett et al. (2020). This underscores the need for developing inherently interpretable models Rudin (2019); Rudin et al. (2022) such as our LucidPPN.

---

> > ### Comment · Reviewer_9WXj · 2024-11-25
> >
> > Thanks for the detailed response from the authors. While some concerns have been addressed, there remain several points that may require further clarification.
> >
> > 1.	About the contributions:
> > The authors' response on the novelty and contribution of analyzing "color," "shape," and "texture" didn’t seem to fully resolve the concern. Visualizing the shallow layers of the neural network has indeed demonstrated the network's focus on the features of "color," "shape," and "texture," and has visualized "shape" and "texture" more clearly, which contributes to making the network processing more transparent from these perspectives. However, the authors state that "conventional neural networks often entangle visual features in ways that make it difficult to disentangle and present them in an understandable format for users", which may not be entirely consistent with the explanatory results of existing methods that visualize shallow layers of networks. These existing methods also provide a certain level of interpretability.
> >
> > 2.	About performance:
> > We agree with the authors' explanation that the performance decrease may due to the delayed fusion of texture and shape features with color. This delay prevents the network from effectively associating shape and texture with color in the early layers, leading to incomplete detection of certain features. However, there is a lack of experimental results and analysis to support this explanation. For instance, comparisons between shallow fusion and deep fusion under the same conditions would help verify whether the accuracy drop is indeed caused by the location of fusions. Given that the performance decline is likely due to insufficient attention to important features (such as texture and shape), the limitations of the interpretability method itself, and other factors, further experiments related to this topic would be beneficial. A more comprehensive analysis of the limitations and the true advantages of the method may be expected.

---

> > > ### Author Response · Authors · 2024-11-25
> > >
> > > As noted in the Review, we agree that there are techniques to visualize the shallow layers of neural networks, and these methods provide a certain level of interpretability. However, LucidPPN focuses on high-level concepts (represented with prototypical parts) from deeper layers that encode complex information. Our goal is to enhance the transparency of these concepts by disentangling the color from the remaining visual features. While there is a wealth of research on prototypical parts-based interpretability (e.g., Chen et al., 2019; Nauta et al., 2021; 2023b; Rymarczyk et al., 2021; 2022; 2023; Wang et al., 2024), none of these works aim to introduce an inherently interpretable mechanism into the network at the level of low-level visual features.
> > >
> > > Regarding accuracy, we note that the results for the late fusion method (LucidPPN) and the earliest fusion method (single branch) are provided in Table 4. For the CUB dataset, LucidPPN scored 81.5% while single branch 86.6%. To investigate the influence of earlier fusion on the accuracy, we also experimented with fusion applied after the second block of ConvNeXt Tiny. This configuration achieved an accuracy of 84.1%, indicating that earlier fusion can indeed enhance the model's accuracy. However, this comes at the cost of explanation granularity: when fusion occurs earlier, it becomes difficult to disentangle the influence of shape with texture and color on prototypical parts.
> > >
> > > We kindly ask the Reviewer to evaluate whether our responses address their concerns. If not, we would appreciate clarification on two points. First, regarding shallow layer visualization, could you specify the techniques or works you had in mind so we can reference them more precisely? Second, in terms of fusion analysis, what specific types of evaluations or comparisons would you find most informative? If no further concerns remain, we kindly request a reevaluation of your score.

---

> > > > ### Comment · Reviewer_9WXj · 2024-11-26
> > > >
> > > > Thanks for the prompt response.
> > > >
> > > > The concerns regarding accuracy have been largely addressed. We agree with the authors on their explanation that by decoupling color, shape, and texture in the early stages and fusing them later for more accurate interpretation, there is an inherent trade-off in terms of accuracy.
> > > >
> > > > Regarding concerns about the contribution, some doubts still remain. While we acknowledge that the authors have made valuable contributions to the prototype network by introducing inherently interpretable mechanisms at the low-level feature level, similar research [1] on shallow network visualizations has already provided substantial insights into the impact of features such as color, shape, and texture on classification network results. This somewhat limits the scope of the contribution of the proposed method in the context of low-level feature interpretation. The authors are still expected to further clarify the unique advantages of their approach compared to shallow network visualization in the revised version, which would definitely better highlight the contribution and innovation of the manuscript.
> > > >
> > > > But overall, after the above rounds of feedback and discussion, I think this work have adequate technical merits and contributions, and I would be happy to raise the score to 6 (marginally above accept).
> > > >
> > > > 1. Zeiler, M. D. (2014). Visualizing and Understanding Convolutional Networks. In European Conference on Computer Vision.

---

> > > > > ### Author Response · Authors · 2024-11-26
> > > > >
> > > > > Thank you for your valuable feedback and for raising the score of our manuscript. We are pleased to hear that we were able to address most of your concerns.
> > > > >
> > > > > To better clarify the unique advantages of our approach compared to shallow network visualization methods, we have revised the section "Usage of low-level vision features for image classification" in the Related Works.
> > > > >
> > > > > We greatly appreciate your insights, which have helped us improve the manuscript.

---

### Official Review · Reviewer_9gD3 · 2024-11-02

**Soundness:** 4
**Presentation:** 4
**Contribution:** 4
**Rating:** 6
**Confidence:** 5

**Summary:**

This paper propose to disentangle color prototypes from other visual features in ProtoPNets, by introducing a novel network architecture, named LucidPPN. The proposed method clarifies feature importance and aligns prototypical parts with object semantics, enhancing interpretability. Experiments show that LucidPPN achieves competitive accuracy while producing clearer and less ambiguous explanations for users.

**Strengths:**

* This paper explicitly decouple prototypes into specific semantic types, such as color and shape, whereas existing methods have overlooked this aspect of information. And I believe this paper could serve as a significant inspiration for future research.
* This paper provides sufficient cases and visualizations to validate the semantic information of the learned prototypes.
* The paper is well-written and easy to follow.
* The authors provide code for reproducibility check.

**Weaknesses:**

[Major]

1. **Quantitative evaluation of the interpretability:** In previous work, Huang et al. [1] have discussed the inconsistency of traditional ProtoPNets. Does this issue exists within the proposed method? Please provide qualitative or quantitative evaluations.
2. **Experiments:** Please supplement the missing results for baseline methods on datasets like DOGS and FLOWERS in Table 1, as adapting to these datasets, which were not covered in the original papers, seems quite straightforward.
3. **Experiments:** This paper only implement the proposed method on several CNNs. However, vision Transformers are introduced to the realm of CV for several years, and have also been implemented as the backbone of ProtoPNets [2]. Please provide additional experimental results using ViT [3-4] or even CLIP [5] as the backbone.
4. **Related Work:** In XAI, introducing human understandable semantics as evidences for prediction has been explored by concept bottleneck models (CBMs) [6]. What is the relationship between the proposed method and CBMs. Can concepts be introduced into the realm of ProtoPNet for higher interpretability?


> [1] Huang, Qihan, et al. "Evaluation and improvement of interpretability for self-explainable part-prototype networks." Proceedings of the IEEE/CVF International Conference on Computer Vision. 2023.
>
> [2] Xue, Mengqi, et al. "Protopformer: Concentrating on prototypical parts in vision transformers for interpretable image recognition." arXiv preprint arXiv:2208.10431 (2022).
>
> [3] Dosovitskiy, Alexey, et al. "An image is worth 16x16 words: Transformers for image recognition at scale." International Conference on Learning Representations. 2021.
>
> [4] Touvron, Hugo, et al. "Training data-efficient image transformers & distillation through attention." International conference on machine learning. PMLR, 2021.
>
> [5] Radford, Alec, et al. "Learning transferable visual models from natural language supervision." International conference on machine learning. PMLR, 2021.
>
> [6] Koh, Pang Wei, et al. "Concept bottleneck models." International conference on machine learning. PMLR, 2020.

[Minor]

1. **Experiments:** What is the computational cost of inference and training? Please provide a comparison with baseline methods, including metrics such as training time, FLOPs, and memory usage.

**Questions:**

My questions are listed in "Weaknesses" section.

---

> ### Author Response · Authors · 2024-11-22
>
> **W1. In previous work, Huang et al. (2023) have discussed the inconsistency of traditional ProtoPNets. Does this issue exist within the proposed method? Please provide qualitative or quantitative evaluations.**
>
> To answer this question we have calculated consistency and stability of our method and compared it in Supplementary Table 11. One can observe that LucidPPN achieves a consistency score comparable to the method proposed by Huang et al. (2023) while outperforming other prototypical-parts-based methods. Regarding stability, LucidPPN demonstrates results on par with other methods. This improvement is likely due to the correspondence of prototypical parts to semantic parts of the classified objects.
>
> **W2. Please supplement the missing results for baseline methods on datasets like DOGS and FLOWERS in Table 1, as adapting to these datasets, which were not covered in the original papers, seems quite straightforward.**
>
> Thank you for your comment. We have added results for baselines on additional datasets, except for the ProtoTree. This exception is due to the model’s tendency to exhibit instability during training on these datasets. Upon reviewing the issues section of the ProtoTree GitHub repository [https://github.com/M-Nauta/ProtoTree/issues](https://github.com/M-Nauta/ProtoTree/issues), we noticed that others have faced similar challenges in applying this model to different datasets.
>
> **W3. This paper only implement the proposed method on several CNNs. However, vision Transformers are introduced to the realm of CV for several years, and have also been implemented as the backbone of ProtoPNets Xue et al. (2022). Please provide additional experimental results using ViT or even CLIP as the backbone.**
>
> Thank you for your comment. Unfortunately, we are unable to run LucidPPN with a ViT backbone for the following reasons:
> * **Incompatibility with PIPNet-Based Prototypical Part Definition**: ProtoPFormer Xue et al. (2022) is built on the ProtoPNet-based definition of prototypical parts Chen et al. (2019), whereas our method relies on the PIPNet-based definition Nauta et al. (2023b). Currently, there is no adaptation of the PIPNet-based definition for the ViT backbone. This limitation is why we opted for the ConvNeXt backbone, which has demonstrated comparable performance to ViTs.
> * **Challenges with Self-Attention**: Adapting a ViT backbone to the PIPNet-based definition is not straightforward due to the nature of self-attention. Unlike convolutions, self-attention lacks the properties of locality and a direct correspondence between the input and feature map Chen et al. (2019). This discrepancy makes it difficult to visualize prototypical parts faithfully.
> * **Orthogonal Research Direction**: Adapting the ViT backbone to prototypical parts represents a separate research direction. Both ProtoPFormer and recent works in this area Ma et al. (2024a), which were unavailable at the time of submission, highlight that integrating a ViT backbone with prototypical parts is a non-trivial task requiring substantial architectural/training changes. For these reasons, we chose to use the ConvNeXt backbone in our work.
>
>
> **W4. In XAI, introducing human understandable semantics as evidence for prediction has been explored by concept bottleneck models (CBMs) Koh et al. (2020). What is the relationship between the proposed method and CBMs. Can concepts be introduced into the realm of ProtoPNet for higher interpretability?**
>
> Thank you for pointing out the connection between CBMs and our work. I’d like to clarify that both concept bottlenecks and prototypical parts can be considered concept-based models Bontempelli et al. (2022). However, there is a key distinction: concept bottlenecks use predefined intermediate classes (named concepts) that are directly associated with the image. While, prototypical parts, aim to identify relevant classification concepts during model training without any additional labels. A potential future research direction could involve combining concept bottlenecks with prototypical parts.
>
> **w1. What is the computational cost of inference and training? Please provide a comparison with baseline methods, including metrics such as training time, FLOPs, and memory usage.**
>
> In Supplementary Table 12 we provide information about training time, GFLOPs needed, and average memory usage during training for LucidPPN, PIPNet, ProtoPool, and ProtoPNet. One can observe that LucidPPN is faster and uses less memory than PIPNet. However, ProtoPNet and ProtoPool require much less memory to train while having longer training times.

---

> > ### Author Response · Authors · 2024-11-25
> >
> > Dear Reviewer 9gD3,
> >
> > As the deadline for the discussion period is approaching quickly, we would like to kindly remind the reviewer that we are waiting for your response.
> >
> > In particular, we have provided point-by-point responses to all of your questions to address your concerns and provided the revision that reflects such changes. Therefore, your timely feedback and change in the score if applicable would be highly appreciated.
> >
> > Best,
> >
> > Authors

---

### Official Review · Reviewer_na4o · 2024-11-03

**Soundness:** 3
**Presentation:** 3
**Contribution:** 3
**Rating:** 8
**Confidence:** 4

**Summary:**

In this paper, the authors proposed a Lucid Prototypical Parts Network (LucidPPN), a novel prototypical parts network that separates color prototypes from other visual features. A LucidPPN has two branches: a ShapeTexNet and a ColorNet. Given an input image, the ShapeTexNet is a convolutional neural network (CNN) that takes a gray-scale version of the image as input and outputs a set of feature maps, and the ColorNet is another CNN that takes a down-sampled version of the image as input and outputs another set of feature maps. Since the last layer of both the ShapeTexNet and the ColorNet is a 1x1 convolutional layer with KM filters, we can interpret the last convolutional layer as a prototype layer with KM prototypes, where K is the number of prototypes per class and M is the number of classes, and the output of the last layer as prototype activation maps. The output feature maps (aka prototype activation maps) from the ShapeTexNet and the ColorNet are fused using element-wise products, and then max-pooled to yield a prototype similarity score for each prototype. The predicted class score is simply an average of the prototype similarity scores over all prototypes of the class. In a LucidPPN, each of the K prototypes in each of the M classes corresponds to consistent image parts (e.g., the first prototype of each class corresponds to head of a bird, etc.). This is achieved by aligning the fused output feature maps (prototype activation maps) with segmentation masks produced by a pre-trained PDiscoNet (an object part segmentation model) using a prototypical-object part correspondence loss. In addition to a loss function to improve the classification accuracy of the entire model, the authors also introduced a loss function to improve the classification accuracy of the ShapeTexNet alone and to disentangle color from other visual features. The authors evaluated their LucidPPN models on 4 commonly used fine-grained classification benchmarks (CUB-200-2011, Stanford Cars, Stanford Dogs, and Oxford Flowers), and found that their LucidPPN models achieved competitive test accuracy compared to other interpretable models. The authors also did a user study to evaluate the influence of disentangling color from other visual attributes on interpretability.

**Strengths:**

- Originality: The paper introduced a novel idea of disentangling color from shape and texture, so that the visual attribute of each prototype is more clearly defined (compared to prior work).
- Quality: The authors did show that their LucidPPN could maintain a reasonable accuracy while providing less ambiguous prototypes.
- Clarity: The paper is clearly written.
- Significance: Interpretability is a significant area of research in machine learning.

**Weaknesses:**

- Quality: There seems to be no prototype projection in this work. Without prototype projections, it is unclear if the prototypes can be faithfully visualized using training images (because the closest training images to a prototype could still be far away from the prototype in the latent space).
- Clarity: Page 6, Lines 314-315. I am confused as to whether you are aligning the segmentation masks from PDiscoNet with prototype activation maps from the ShapeTexNet or the aggregated feature maps.

**Questions:**

- My main concern is that I did not see prototype projections in this work. Without prototype projections, how could you conclusively visualize prototypes using training images? The closest training images to a prototype could still be far away from the prototype in the latent space.
- During training, are the segmentation masks from PDiscoNet aligned with the ShapeTexNet feature maps or the aggregated feature maps?
- I am also not clear as to why binary cross entropy is used instead of multi-class cross entropy for training?

**Details Of Ethics Concerns:**

N/A.

---

> ### Author Response · Authors · 2024-11-22
>
> **Q1. My main concern is that I did not see prototype projections in this work. Without prototype projections, how could you conclusively visualize prototypes using training images? The closest training images to a prototype could still be far away from the prototype in the latent space.**
>
> Our work builds on PIPNet’s definition of prototypical parts, which is why it lacks projection, which can lead to less faithful visualizations. Despite this drawback, PIPNet-based architectures have been successfully applied in various works De Santi et al. (2024a;b); Nauta et al. (2023a), and further developed, e.g. Wang et al. (2024) improving the interpretability.
>
> Moreover, LucidPPN introduces a key difference in the definition of prototypical parts compared to PIPNet. While PIPNet employs Softmax across channels in the latent feature map, LucidPPN uses the sigmoid activation function. The sigmoid function allows each channel’s activation to be learned independently, not influenced by the relative activations of other channels. While, Softmax normalization can distort activations by emphasizing values that are only relatively high compared to others, even if they are low in absolute terms.
>
> To build an intuition for this statement, let us consider $i$-th pixel of a feature map with activation values $z_i = [−2, 5, −0.2, −0.1]$, and $j$-th pixel with $z_j = [10, 300, 30, 10]$, the Softmax output $\theta$ for both would be $\theta_i = \theta_j = [0, 1, 0, 0]$. This implies that PIPNet would treat both pixels as equally important, despite the activations differing by a factor of 60. In contrast, with sigmoid activation $\sigma$ used in LucidPPN, the outputs would be $\sigma_i = [0.1192, 0.9933, 0.4502, 0.4750]$ and $\sigma_j = [1.0000, 1.0000, 1.0000, 1.0000]$, preserving the distinction in activation magnitudes. As a result, one can easily verify if the image patches selected for visualization are faithful because such patches should have a resemblance score close to 1.
>
>
> **Q2. During training, are the segmentation masks from PDiscoNet aligned with the ShapeTexNet feature maps or the aggregated feature maps?**
>
> Loss $L_D$ is applied only to the ShapeTexNet feature maps as we directly align them with masks from PDiscoNet. Indirectly, it also causes alignment of masks with the aggregated feature maps which are computed from the ShapeTexNet feature maps. To the Supplementary Materials (Figure 14), we added an image illustrating this process more concisely.
>
>
> **Q3. I am also not clear as to why binary cross entropy is used instead of multi-class cross entropy for training?**
>
> The intuition behind BCE usage is rooted from multilabel classification. To some degree ShapeTexNet operates in a multilabel setting from the prototypical parts perspective as they may match multiple classes. Hence, to enable multiple classes having high similarity to the same prototypical parts, we use sigmoid instead of softmax when computing the feature maps. This necessitates a shift from Cross-Entropy (CE) to Binary Cross-Entropy (BCE) because CE would then solely maximize the activation of the correct class while ignoring crucial signals from negative classes. Another reason behind our choice is to make it easier to verify the faithfulness of visualizations (see the answer about prototype projection).

---

> > ### Author Response · Authors · 2024-11-25
> >
> > Dear Reviewer na4o,
> >
> > As the deadline for the discussion period is approaching quickly, we would like to kindly remind the reviewer that we are waiting for your response.
> >
> > In particular, we have provided point-by-point responses to all of your questions to address your concerns and provided the revision that reflects such changes. Therefore, your timely feedback and change in the score if applicable would be highly appreciated.
> >
> > Best,
> >
> > Authors

---

> ### Comment · Reviewer_na4o · 2024-11-25
> **Thank you for your response**
>
> Thank you for your response!
>
> I see why you chose sigmoid (instead of softmax) to be applied to the latent feature maps.
>
> "As a result, one can easily verify if the image patches selected for visualization are faithful because such patches should have a resemblance score close to 1."
>
> In order to establish that your prototype visualizations are faithful, did you verify that every image patch selected for the visualization of a prototype has a self-resemblance score close to 1 with the prototype itself? Is it true? What would you do if you found a prototype whose visualized patch did not have a self-resemblance score close to 1?

---

> > ### Author Response · Authors · 2024-11-26
> >
> > To answer this question, we provide an additional section "Faithfullness of patch visualizations" in the Supplementary Materials. It contains Figure 20 with a distribution of the sigmoid function values obtained for patches used in prototype visualization. For LucidPPN trained on the CUB dataset (blue curve), 61.04% of those patches have values above 0.9, which indicates that prototype visualizations are relatively faithful.
> >
> > Moreover, we show that higher faithfulness can be obtained when training with an additional loss component $L_C$ that punishes the model if the sigmoid function value for a given prototype is smaller than $1$ for all samples in the batch (see green and yellow curves in Figure 20).

---

> ### Comment · Reviewer_na4o · 2024-11-26
> **Thank you for your clarification**
>
> Thank you for your clarification. I have raised my score to 6: marginally above the acceptance threshold.
>
> To further improve the paper, it would be interesting to see what happens if you remove all prototypes whose self-resemblance score is not close to 1.
>
> Also, it would be helpful to add a discussion on why you chose sigmoid (instead of softmax) and binary cross entropy (instead of multi-class cross entropy) for your method.

---

> > ### Author Response · Authors · 2024-11-27
> >
> > Thank you for your valuable feedback and for raising the score.
> >
> > It is indeed interesting to see the effects of pruning the prototypes with less faithful representation (those with resemblance scores < 0.9). Therefore, we investigate it in the newly added section "Pruning prototypes with less faithful visualizations" of the Supplementary Materials.
> > It contains Table 13, which shows that LucidPPN accuracy after pruning drops only by around 2% (from 81.6% to 79.3%). However, interestingly, the accuracy stays the same for $L_C=0.05$.
> > It suggests that combination of $L_C$ and pruning allows to enforce high resemblance scores (>0.9) of visualized patches without sacrificing on the accuracy.
> >
> > When it comes to the discussion on choosing sigmoid and binary cross entropy, we added it as section "Reason behind using the Binary Cross Entropy with Sigmoid instead of the Cross Entropy with Softmax" of the Supplementary Materials.

---

> > > ### Comment · Reviewer_na4o · 2024-11-27
> > > **Thank you for your clarification**
> > >
> > > Thank you for your clarification. I have raised my score to 8: accept, good paper.

---

> > > > ### Author Response · Authors · 2024-11-27
> > > >
> > > > We sincerely appreciate your reassessment and are thrilled that the revisions addressed your concerns. Thank you for your time and constructive feedback!

---

### Official Review · Reviewer_Zw4S · 2024-11-09

**Soundness:** 2
**Presentation:** 2
**Contribution:** 2
**Rating:** 6
**Confidence:** 3

**Summary:**

Summary Of Contributions:
1.Introduction of LucidPPN: This novel architecture separates color features from other visual components during inference, enabling clearer identification of feature importance in the decision-making process.
2.Consistent Object-Part Mapping: A mechanism ensures that prototypes within each class consistently correspond to the same object parts, improving interpretability.
3.Enhanced Visualization Method: A more intuitive visualization type is introduced, optimized for fine-grained classification.
4.Comprehensive Analysis: The paper provides an in-depth examination of LucidPPN's usefulness and limitations, particularly identifying cases where color may or may not be a critical feature in fine-grained classification.

**Strengths:**

1.The LucidPPN in the paper consists of two branches, one for color and the other for shape/texture, which effectively decouples different features. This method can reduce the ambiguity of traditional prototype networks and enable users to better understand the reasons behind the model's decisions.

2.Compared to existing methods, LucidPPN achieves a more detailed analysis of Prototypical Parts, making it easier for users to understand the features that the model is focusing on.

3.Through user studies, it was proven that the explanations provided by LucidPPN are clearer and easier for users to understand than those of other models such as PIP-Net. This empirical result helps to enhance the persuasiveness of the method.

**Weaknesses:**

Weakness

1.The Section 3 has a lot of paragraphs but lacks subheadings, making it difficult to follow the logical flow of the different parts.

2.There was no noticeable advantage in accuracy. The model was compared on four datasets in total, and its accuracy was lower than that of PIP-Net on two of the datasets, especially on the CUB dataset, where its accuracy was lower than that of all three methods, and no explanation was given for this gap.

**Questions:**

Concerns:
1.It is recommended to add subheadings to each key step or method description to make it easier for readers to understand and locate the content.

2.Consider further improving the accuracy of LucidPPN to enhance its explainability while maintaining a minimal loss of performance.

---

> ### Author Response · Authors · 2024-11-22
>
> **1. The Section 3 has a lot of paragraphs but lacks subheadings, making it difficult to follow the logical flow of the different parts.**
>
> We agree that Section 3 was dense. Therefore, we have revised it by introducing subsections that clarify the methodology of LucidPPN more clearly.
>
> **2. There was no noticeable advantage in accuracy. Why?**
>
> We answer this question in **shared remarks** in paragraphs *There was no noticeable advantage in accuracy. Why?* and *The improvements demonstrated by the proposed method appear to be limited because its performance on some instances is lower than that of the compared methods.*

---

> > ### Author Response · Authors · 2024-11-25
> >
> > Dear Reviewer Zw4S,
> >
> > As the deadline for the discussion period is approaching quickly, we would like to kindly remind the reviewer that we are waiting for your response.
> >
> > In particular, we have provided point-by-point responses to all of your questions to address your concerns and provided the revision that reflects such changes. Therefore, your timely feedback and change in the score if applicable would be highly appreciated.
> >
> > Best,
> >
> > Authors

---

> > > ### Comment · Reviewer_Zw4S · 2024-11-26
> > >
> > > Thanks for the response ! I got the explanation for limited improvement. Considering the overall quality, I vote for borderline accept.

---

> > > > ### Author Response · Authors · 2024-11-26
> > > >
> > > > We sincerely thank you for your valuable insights, which have significantly contributed to enhancing our manuscript.

---

### Author Response · Authors · 2024-11-22

We sincerely thank the Reviewers for their positive and encouraging feedback. They have recognized that our work can *serve as a significant inspiration for future research* (R9gD3), introduces *a novel idea of disentangling color from shape and texture* (R-na4o), and addresses a *significant field of research in machine learning* (R-na4o).

The clarity and interpretability of LucidPPN have been particularly appreciated. Reviewers have highlighted that it *can reduce the ambiguity of traditional prototype networks and enable users to better understand the reasons behind the model’s decisions* (R-Zw4S) and that the model makes *it easier for users to understand the features that the model is focusing on* (R-Zw4S). Furthermore, it was emphasized that *the explanations provided by LucidPPN are clearer and easier for users to understand.* (R-Zw4S).

LucidPPN has also been noted for its presentation, described as *clearly written* (R-na4o), and *easy to follow* (R-9gD3). Reviewers appreciated that *this paper provides sufficient cases and visualizations to validate the semantic information of the learned prototypes* (R9gD3) and that *the methodology is well-structured, with intuitive design* (R-9WXj). The experiments were recognized as *comprehensive, with a substantial number of visualization results.* (R-9WXj).

We have carefully addressed the Reviewers’ comments and incorporated their suggestions to strengthen our manuscript. We kindly ask the Reviewers to consider increasing their rating if they find our responses satisfactory. Responses to remarks shared among Reviewers are provided below, and followed by replies to specific comments. Additionally, we have attached a revised version of the work with all changes highlighted in blue.


### **Shared remarks**

**(R-Zw4S, R-jXW9) There was no noticeable advantage in accuracy. Why?**
The primary goal of this work was not to surpass PIPNet in accuracy but to reduce the ambiguity of prototypical parts through color disentanglement and correspondence to semantic parts of classified objects. Multiple works Adebayo et al. (2018); Huang et al. (2023); Kim et al. (2022); Ma et al. (2024b) show that explanations are ambiguous for a user and can cause overconfidence. That is why one should consider user study as the main result that shows that LucidPPN enabled significantly better user scores than PIPNet, even on the CUB dataset where LucidPPN’s accuracy was lower.

**(R-Zw4S, R-jXW9) The improvements demonstrated by the proposed method appear to be limited because its performance on some instances is lower than that of the compared methods.**
The accuracy drop stems from a late-stage fusion of texture and shape features with color. This delay prevents the network from correlating shape and texture with color effectively in earlier layers, causing some features to go undetected. It can be seen as a multimodal scenario, where early fusion (in our case PIPNet) achieves higher accuracy than late fusion (in our case LucidPPN), just like in Nagrani et al. (2021). Nonetheless, increasing the disambiguation of prototypical parts can improve accuracy over PIPNet, like in 3 of the 5 datasets, including PartImageNet added in the rebuttal phase.

---

> ### Author Response · Authors · 2024-11-22
>
> ### **References**
>
> J. Adebayo et al. Sanity checks for saliency maps. NeurIPS 2018.
>
> A. Bontempelli et al. Concept-level debugging of part-prototype networks. arXiv 2022.
>
> C. Chen et al. This looks like that: deep learning for interpretable image recognition. NeurIPS 2019.
>
> L. A. De Santi et al. Patch-based intuitive multimodal prototypes network (pimpnet) for alzheimer’s disease classification. arXiv 2024a.
>
> L. A. De Santi et al. Pipnet3d: Interpretable detection of alzheimer in mri scans. arXiv 2024b.
>
> J. He et al. Partimagenet: A large, high-quality dataset of parts. ECCV 2022.
>
> Q. Huang et al. Evaluation and improvement of interpretability for self-explainable part-prototype networks. ICCV 2023.
>
> S. SY Kim et al. Hive: Evaluating the human interpretability of visual explanations. ECCV 2022.
>
> P. W. Koh et al. Concept bottleneck models. ICML 2020.
>
> C. Ma et al. Interpretable image classification with adaptive prototype-based vision transformers. arXiv 2024a.
>
> C. Ma et al. This looks like those: Illuminating prototypical concepts using multiple visualizations. NeurIPS 2024b.
>
> A. Nagrani et al. Attention bottlenecks for multimodal fusion. NeurIPS 2021.
>
> M. Nauta et al. Neural prototype trees for interpretable fine-grained image recognition. CVPR 2021.
>
> M. Nauta et al. Interpreting and correcting medical image classification with pip-net. ECAI 2023a.
>
> M. Nauta et al. Pip-net: Patch-based intuitive prototypes for interpretable image classification. CVPR 2023b.
>
> C. Rudin. Stop explaining black box machine learning models for high stakes decisions and use interpretable models instead. Nature machine intelligence 2019.
>
> C. Rudin et al. Interpretable machine learning: Fundamental principles and 10 grand challenges. Statistic Surveys 2022.
>
> D. Rymarczyk et al. Protopshare: Prototypical parts sharing for similarity discovery in interpretable image classification. ACM SIGKDD 2021.
>
> D. Rymarczyk et al. Interpretable image classification with differentiable prototypes assignment. ECCV 2022.
>
> D. Rymarczyk et al. Icicle: Interpretable class incremental continual learning. ICCV 2023
>
> R. Tomsett et al. Sanity checks for saliency metrics. AAAI 2020.
>
> BS Wang et al. Mcpnet: An interpretable classifier via multi-level concept prototypes. CVPR 2024.
>
> M. Xue et al. Protopformer: Concentrating on prototypical parts in vision transformers for interpretable image recognition. arXiv 2022.

---

### Meta-Review · Area_Chair_xBkS · 2024-12-20

**Metareview:**

In this paper, a Lucid Prototypical Parts Network (LucidPPN) prototypical parts network is presented,  which has two branches: a ShapeTexNet and a ColorNet. Given an input image, the ShapeTexNet is a convolutional neural network (CNN) that takes a gray-scale version of the image as input and outputs a set of feature maps, and the ColorNet is another CNN that takes a down-sampled version of the image as input and outputs another set of feature maps. Evaluation is carried out on 4 commonly used fine-grained classification benchmarks (CUB-200-2011, Stanford Cars, Stanford Dogs, and Oxford Flowers), and found the LucidPPN models achieved competitive test accuracy compared to other interpretable models.

**Additional Comments On Reviewer Discussion:**

All the reviewers lean to accept the paper. Thanks for the good job.

---

### Decision · Program_Chairs · 2025-01-22

Accept (Poster)